# Globally injective and bijective neural operators

Takashi Furuya[1]    Michael Puthawala[2]    Matti Lassas[3]    Maarten V. de Hoop[4]

[1]Shimane University, `takashi.furuya0101@gmail.com`
[2]South Dakota State University, `Michael.Puthawala@sdstate.edu`
[3]University of Helsinki, `matti.lassas@helsinki.fi`
[4]Rice University, `mdehoop@rice.edu`

## Abstract

Recently there has been great interest in operator learning, where networks learn operators between function spaces from an essentially infinite-dimensional perspective. In this work we present results for when the operators learned by these networks are injective and surjective. As a warmup, we combine prior work in both the finite-dimensional ReLU and operator learning setting by giving sharp conditions under which ReLU layers with linear neural operators are injective. We then consider the case when the activation function is pointwise bijective and obtain sufficient conditions for the layer to be injective. We remark that this question, while trivial in the finite-rank setting, is subtler in the infinite-rank setting and is proven using tools from Fredholm theory. Next, we prove that our supplied injective neural operators are universal approximators and that their implementation, with finite-rank neural networks, are still injective. This ensures that injectivity is not 'lost' in the transcription from analytical operators to their finite-rank implementation with networks. Finally, we conclude with an increase in abstraction and consider general conditions when subnetworks, which may have many layers, are injective and surjective and provide an exact inversion from a 'linearization.' This section uses general arguments from Fredholm theory and Leray-Schauder degree theory for non-linear integral equations to analyze the mapping properties of neural operators in function spaces. These results apply to subnetworks formed from the layers considered in this work, under natural conditions. We believe that our work has applications in Bayesian uncertainty quantification where injectivity enables likelihood estimation and in inverse problems where surjectivity and injectivity corresponds to existence and uniqueness of the solutions, respectively.

## 1  Introduction

In this work, we produce results at the intersection of two fields: neural operators (NO), and injective and bijective networks. Neural operators [Kovachki et al., 2021a,b] are neural networks that take a infinite dimensional perspective on approximation by directly learning an operator between Sobolev spaces. Injectivity and bijectivity on the other hand are fundamental properties of networks that enable likelihood estimation by the change of variables formula, are critical in applications to inverse problems, and are useful properties for downstream applications.

The key contribution of our work is the translation of fundamental notions from the finite-rank setting to the infinite-rank setting. By the 'infinite-dimension setting' we refer to the case when the object of approximation is a mapping between Sobolev spaces. This task, although straight-forward on first inspection, often requires dramatically different arguments and proofs as the topology, analysis and notion of noise are much simpler in the finite-rank case as compared to the infinite-rank case. We see our work as laying the groundwork for the application of neural operators to generative models

37th Conference on Neural Information Processing Systems (NeurIPS 2023).

in function spaces. In the context of operator extensions of traditional VAEs [Kingma and Welling, 2013], injectivity of a decoder forces distinct latent codes to correspond to distinct outputs.

Our work draws parallels between neural operators and pseudodifferential operators Taylor [1981], a class that contains many inverses of linear partial differential operators and integral operators. The connection to pseudodifferential operators provided an algebraic perspective to linear PDE Kohn and Nirenberg [1965]. An important fact in the analysis of pseudodifferential operators, is that the inverses of certain operators, e.g. elliptic pseudodifferential operators, are themselves pseudodifferential operators. By proving an analogous result in section 4.2, that the inverse of invertible NO are themselves given by NO, we draw an important and profound connection between (non)linear partial differential equations and NO.

We also believe that our methods have applications to the solution of inverse problems with neural networks. The desire to use injective neural networks is one of the primary motivations for this work. These infinite dimensional models can then be approximated by a finite dimensional model without losing discretization invariance, see Stuart [2010]. Crucially, discretization must be done 'at the last possible moment,' or else performance degrades as the discretization becomes finer, see Lassas and Siltanen [2004] and also Saksman et al. [2009]. By formulating machine learning problems in infinite dimensional function spaces and then approximating these methods using finite dimensional subspaces, we avoid bespoke ad-hoc methods and instead obtain methods that apply to any discretization.

More details on our motivations for and applications of injectivity & bijectivity of neural operators are given in Appendix A.

## 1.1 Our Contribution

In this paper, we give a rigorous framework for the analysis of the injectivity and bijectivity of neural operators. Our contributions are as follows:

(A) We show an equivalent condition for the layerwise injectivity and bijectivity for linear neural operators in the case of ReLU and bijective activation functions (Section 2). In the particular ReLU case, the equivalent condition is characterized by a directed spanning set (Definition 2).

(B) We prove that injective linear neural operators are universal approximators, and that their implementation by finite rank approximation is still injective (Section 3). We note that universal approximation theorem (Theorem 1) in the infinite dimensional case does not require an increase in dimension, which deviate from the finite dimensional case Puthawala et al. [2022a, Thm. 15].

(C) We zoom out and perform a more abstract global analysis in the case when the input and output dimensions are the same. In this section we 'coarsen' the notion of layer, and provide a sufficient condition for the surjectivity and bijectivity of nonlinear integral neural operators *with nonlinear kernels*. This application arises naturally in the context of subnetworks and transformers. We construct their inverses in the bijective case (Section 4).

## 1.2 Related Works

In the finite-rank setting, injective networks have been well-studied, and shown to be of theoretical and practical interest. See Gomez et al. [2017], Kratsios and Bilokopytov [2020], Teshima et al. [2020], Ishikawa et al. [2022], Puthawala et al. [2022a] for general references establishing the usefulness of injectivity or any of the works on flow networks for the utility of injectivity and bijectivity for downstream applications, [Dinh et al., 2016, Siahkoohi et al., 2020, Chen et al., 2019, Dinh et al., 2014, Kingma et al., 2016], but their study in the infinite-rank setting is comparatively underdeveloped. These works, and others, establish injectivity in the finite-rank setting as a property of theoretical and practical interest. Our work extends Puthawala et al. [2022a] to the infinite-dimensional setting as applied to neural operators, which themselves are a generalization of multilayer perceptrons (MLPs) to function spaces. Moreover, our work includes not only injectivity, but also surjectivity in the non-ReLU activation case, which Puthawala et al. [2022a] has not focused on.

Examples of works in these setting include neural operators Kovachki et al. [2021b,a], DeepONet Lu et al. [2019], Lanthaler et al. [2022], and PCA-Net Bhattacharya et al. [2021], De Hoop et al. [2022]. The authors of Alberti et al. [2022] recently proposed continuous generative neural networks (CGNNs), which are convolution-type architectures for generating $L^2(\mathbb{R})$-functions, and provided the sufficient condition for the global injectivity of their network. Their approach is the wavelet basis expansion, whereas our work relies on an independent choice of basis expansion.

## 1.3 Networks considered and notation

Let $D \subset \mathbb{R}^d$ be an open and connected domain, and $L^2(D; \mathbb{R}^h)$ be the $L^2$ space of $\mathbb{R}^h$-value function on $D$ given by $L^2(D; \mathbb{R}^h) := \underbrace{L^2(D; \mathbb{R}) \times \cdots \times L^2(D; \mathbb{R})}_{h} = L^2(D)^h$.

**Definition 1** (Integral and pointwise neural operators). *We define an integral neural operator $G$ : $L^2(D)^{d_{in}} \to L^2(D)^{d_{out}}$ and layers $\mathcal{L}_\ell : L^2(D)^{d_\ell} \to L^2(D)^{d_{\ell+1}}$ by*

$$G := T_{L+1} \circ \mathcal{L}_L \circ \cdots \mathcal{L}_1 \circ T_0, \quad (\mathcal{L}_\ell v)(x) := \sigma(T_\ell(v)(x) + b_\ell(x)),$$

$$T_\ell(v)(x) = W_\ell(x)u(x) + \int_D k_\ell(x, y)u(y), \quad x \in D,$$

*where $\sigma : \mathbb{R} \to \mathbb{R}$ is a non-linear activation operating element-wise, and $k_\ell(x, y) \in L^2(D \times D; \mathbb{R}^{d_{\ell+1} \times d_\ell})$ are integral kernels, and $W_\ell \in C(\overline{D}; \mathbb{R}^{d_{\ell+1} \times d_\ell})$ are pointwise multiplications with matrices, and $b_\ell \in L^2(D)^{d_{\ell+1}}$ are bias functions ($\ell = 1, ..., L$). Here, $T_0 : L^2(D)^{d_{in}} \to L^2(D)^{d_1}$ and $T_{L+1} : L^2(D)^{d_{L+1}} \to L^2(D)^{d_{out}}$ are mappings (lifting operator) from the input space to the feature space and mappings (projection operator) from the feature spaces to the output space, respectively.*

The layers $T_0$ and $T_{L+1}$ play a special role in the neural operators. They are local linear operators and serve to lift and project the input data from and to finite-dimensional space respectively. These layers may be 'absorbed' into the layers $\mathcal{L}_1$ and $\mathcal{L}_L$ without loss of generality (under some technical conditions), but are not in this text to maintain consistency with prior work. Prior work assumes that $d_{in} < d_1$, we only assume that $d_{in} \leq d_1$ for lifting operator $T_0 : L^2(D)^{d_{in}} \to L^2(D)^{d_1}$. This would seemingly play an important role in the context of injectivity or universality, but we find that our analysis does not require that $d_{in} < d_1$ at all. In fact, as elaborated in Section 3.2, we may take $d_{in} = d_\ell = d_{out}$ for $\ell = 1, \ldots, L$ and our analysis is the same.

## 2 Injective linear neural operator layers with ReLU and bijective activations

In this section we present sharp conditions under which a layer of a neural operator with $\mathrm{ReLU}$ activation is injective. The Directed Spanning Set (DSS) condition, described by Def. 2 is a generalization of the finite-dimensional DSS [Puthawala et al., 2022a] which guarantees layerwise injectivity of $\mathrm{ReLU}$ layers. Extending this condition from finite to infinite dimensions is not automatic. The finite-dimensional DSS will hold with high probability for random weight matrices if they are expansive enough ([Puthawala et al., 2022a, Theorem7]). However, the infinite-dimensional DSS is much more restrictive than the finite dimensional setting. We then present a less restrictive condition that is met when the activation function is bijective, e.g. a leaky-ReLU activation is used.

Although it may appear that the end-to-end result is strictly stronger than the layerwise result, this is not the case. The layerwise result is an exact characterization, whereas the end-to-end result is sufficient for injectivity, but not necessary. The layerwise analysis is also constructive, and so gives a rough guide for the construction of injective networks, whereas the global analysis is less so. Finally, the layerwise condition has different applications, such as network of stochastic depth, see e.g. Huang et al. [2016], Benitez et al. [2023]. End-to-end injectivity by enforcing layerwise injectivity is straightforward, whereas deriving a sufficient condition for networks of any depth is more daunting.

We denote by

$$\sigma(Tv + b)(x) := \begin{pmatrix} \sigma(T_1 v(x) + b_1(x)) \\ \vdots \\ \sigma(T_m v(x) + b_m(x)) \end{pmatrix}, \quad x \in D, \ v \in L^2(D)^n$$

where $\sigma : \mathbb{R} \to \mathbb{R}$ is a non-linear activation function, $T \in \mathcal{L}(L^2(D)^n, L^2(D)^m)$, and $b \in L^2(D)^m$, where $\mathcal{L}(L^2(D)^n, L^2(D)^m)$ is the space of linear bounded operators from $L^2(D)^n$ to $L^2(D)^m$. The aim of this section is to characterize the injectivity condition for the operator $v \mapsto \sigma(Tv+b)$ mapping from $L^2(D)^n$ to $L^2(D)^m$, which corresponds to layer operators $\mathcal{L}_\ell$. Here, $T : L^2(D)^n \to L^2(D)^m$ is linear.

## 2.1 ReLU activation

Let $\mathrm{ReLU} : \mathbb{R} \to \mathbb{R}$ be ReLU activation, defined by $\mathrm{ReLU}(s) = \max\{0, s\}$. With this activation function, we introduce a definition which we will find sharply characterizes layerwise injectivity.

**Definition 2** (Directed Spanning Set). *We say that the operator $T + b$ has a directed spanning set (DSS) with respect to $v \in L^2(D)^n$ if*

$$\mathrm{Ker}\left(T\big|_{S(v,T+b)}\right) \cap X(v, T+b) = \{0\}, \tag{2.1}$$

*where $T|_{S(v,T+b)}(v) = (T_i v)_{i \in S(v,T+b)}$ and*

$$S(v, T+b) := \left\{ i \in [m] \,\Big|\, T_i v + b_i > 0 \text{ in } D \right\}, \tag{2.2}$$

$$X(v, T+b) := \left\{ u \in L^2(D)^n \,\middle|\, \begin{array}{l} \text{for } i \notin S(v, T+b) \text{ and } x \in D, \\ (i)\ T_i v(x) + b_i(x) \leq T_i u(x) \text{ if } T_i v(x) + b_i(x) \leq 0, \\ (ii)\ T_i u(x) = 0 \text{ if } T_i v(x) + b_i(x) > 0 \end{array} \right\}. \tag{2.3}$$

The name directed spanning set arises from the $\ker(T|_{S(v,T+b)})$ term of (2.1). The indices of $S(v, T + b)$ are those that are directed (positive) in the direction of $v$. If $T$ restricted to these indices together span $L^2(D)^n$, then $\ker(T|_{S(v,T+b)})$ is $\{0\}$, and the condition is automatically satisfied. Hence, the DSS condition measures the extent to which the set of indices, which are directed w.r.t. $v$, form a span of the input space.

**Proposition 1.** *Let $T \in \mathcal{L}(L^2(D)^n, L^2(D)^m)$ and $b \in L^2(D)^m$. Then, the operator $\mathrm{ReLU} \circ (T+b) : L^2(D)^n \to L^2(D)^m$ is injective if and only if $T + b$ has a DSS with respect to every $v \in L^2(D)^n$ in the sense of Definition 2.*

See Section B for the proof. Puthawala et al. [2022a] has provided the equivalent condition for the injectivity of ReLU operator in the case of the Euclidean space. However, proving analogous results for operators in function spaces require different techniques. Note that because Def. 2 is a sharp characterization of injectivity, it can not be simplified in any significant way. The condition restrictive Def. 2 is, therefore, difficult to relax while maintaining generality. This is because for each function $v$, multiple components of the function $Tv + b$ are strictly positive in the entire domain $D$, and cardinality $|S(v; T + b)|$ of $S(v; T + b)$ is larger than $n$. This observation prompts us to consider the use of bijective activation functions instead of ReLU, such as leaky ReLU function, defined by $\sigma_a(s) := \mathrm{ReLU}(s) - a\,\mathrm{ReLU}(-s)$ where $a > 0$.

## 2.2 Bijective activation

If $\sigma$ is injective, then injectivity of $\sigma \circ (T + b) : L^2(D)^n \to L^2(D)^m$ is equivalent to the injectivity of $T$. Therefore, we consider the bijectivity in the case of $n = m$. As mentioned in Section 1.3, an significant example is $T = W + K$, where $W \in \mathbb{R}^{n \times n}$ is injective and $K : L^2(D)^n \to L^2(D)^n$ is a linear integral operator with a smooth kernel. This can be generalized to Fredholm operators (see e.g., Jeribi [2015, Section 2.1.4]), which encompasses the property for identity plus a compact operator. It is well known that a Fredholm operator is bijective if and only if it is injective and its Fredholm index is zero. We summarize the above observation as follows:

**Proposition 2.** *Let $\sigma : \mathbb{R} \to \mathbb{R}$ be bijective, and let $T : L^2(D)^n \to L^2(D)^m$ and $b \in L^2(D)^m$. Then, $\sigma \circ (T + b) : L^2(D)^n \to L^2(D)^m$ is injective if and only if $T : L^2(D)^n \to L^2(D)^m$ is injective. Furthermore, if $n = m$ and $T \in \mathcal{L}(L^2(D)^n, L^2(D)^n)$ is the linear Fredholm operator, then, $\sigma \circ (T + b) : L^2(D)^n \to L^2(D)^n$ is bijective if and only if $T : L^2(D)^n \to L^2(D)^n$ is injective with index zero.*

We believe that this characterization of layerwise injectivity is considerably less restrictive than Def. 2, and the characterization of bijectivity in terms of Fredholm theory will be particularly useful in establishing operator generalization of flow networks.

# 3 Global analysis of injectivity and finite-rank implementation

In this section we consider global properties of the injective and bijective networks that constructed in Section 2. First we construct end-to-end injective networks that are not layerwise injective. By doing this, we may avoid the dimension increasing requirement that would be necessary from a layerwise analysis. Next we show that injective neural operators are universal approximators of continuous functions. Although the punchline resembles that of [Puthawala et al., 2022a, Theorem 15], which relied on Whitney's embedding theorem, the arguments are quite different. The finite-rank case has dimensionality restrictions, as required by degree theory, whereas our infinite-rank result does not. Finally, because all implementations of neural operators are ultimately finite-dimensional, we present a theorem that gives conditions under which finite-rank approximations to injective neural operators are also injective.

## 3.1 Global analysis

By using the characterization of layerwise injectivity discussed in Section 2, we can compose injective layers to form $\mathcal{L}_L \circ \cdots \mathcal{L}_1 \circ T_0$, a injective network. Layerwise injectivity, however, prevents us from getting injectivity of $T_{L+1} \circ \mathcal{L}_L \circ \cdots \mathcal{L}_1 \circ T_0$ by a layerwise analysis if $d_{L+1} > d_{out}$, as is common in application [Kovachki et al., 2021b, Pg. 9]. In this section, we consider global analysis and show that $T_{L+1} \circ \mathcal{L}_L \circ \cdots \mathcal{L}_1 \circ T_0$, nevertheless, remains injective. This is summarized in the following lemma.

**Lemma 1.** *Let $\ell \in \mathbb{N}$ with $\ell < m$, and let the operator $T : L^2(D)^n \to L^2(D)^m$ be injective. Assume that there exists an orthogonal sequence $\{\xi_k\}_{k \in \mathbb{N}}$ in $L^2(D)$ and a subspace $S$ in $L^2(D)$ such that $\mathrm{Ran}(\pi_1 T) \subset S$ and*

$$\mathrm{span}\{\xi_k\}_{k \in \mathbb{N}} \cap S = \{0\}. \tag{3.1}$$

*where $\pi_1 : L^2(D)^m \to L^2(D)$ is the restriction operator defined in (C.1). Then, there exists a linear bounded operator $B \in \mathcal{L}(L^2(D)^m, L^2(D)^\ell)$ such that $B \circ T : L^2(D)^n \to L^2(D)^\ell$ is injective.*

See Section C.1 in Appendix C for the proof. $T$ and $B$ correspond to $\mathcal{L}_L \circ \cdots \mathcal{L}_1 \circ T_0$ (from lifting to $L$-th layer) and $T_{L+1}$ (projection), respectively. The assumption (3.1) on the span of $\xi_k$ encodes a subspace distinct from the range of $T$. In Remark 3 of Appendix C, we provide an example that satisfies the assumption (3.1). Moreover, in Remark 4 of Appendix C, we show the exact construction of the operator $B$ by employing projections onto the closed subspace, using the orthogonal sequence $\{\xi_k\}_{k \in \mathbb{N}}$. This construction is given by the combination of "Pairs of projections" discussed in Kato [2013, Section I.4.6] with the idea presented in [Puthawala et al., 2022b, Lemma 29].

## 3.2 Universal approximation

We now show that the injective networks that we consider in this work are universal approximators. We define the set of integral neural operators with $L^2$-integral kernels by

$$\mathrm{NO}_L(\sigma; D, d_{in}, d_{out}) := \Big\{ G : L^2(D)^{d_{in}} \to L^2(D)^{d_{out}} :$$

$$G = K_{L+1} \circ (K_L + b_L) \circ \sigma \cdots \circ (K_2 + b_2) \circ \sigma \circ (K_1 + b_1) \circ (K_0 + b_0),$$

$$K_\ell \in \mathcal{L}(L^2(D)^{d_\ell}, L^2(D)^{d_{\ell+1}}), \ K_\ell : f \mapsto \int_D k_\ell(\cdot, y) f(y) dy \Big|_D, \ k_\ell \in L^2(D \times D; \mathbb{R}^{d_{\ell+1} \times d_\ell}),$$

$$b_\ell \in L^2(D; \mathbb{R}^{d_{\ell+1}}), \ d_\ell \in \mathbb{N}, \ d_0 = d_{in}, \ d_{L+2} = d_{out}, \ \ell = 0, ..., L+2 \Big\}, \tag{3.2}$$

and

$$\mathrm{NO}_L^{inj}(\sigma; D, d_{in}, d_{out}) := \{ G \in \mathrm{NO}_L(\sigma; D, d_{in}, d_{out}) : G \text{ is injective} \}.$$

The following theorem shows that $L^2$-injective neural operators are universal approximators of continuous operators.

**Theorem 1.** *Let $D \subset \mathbb{R}^d$ be a Lipschitz bounded domain, and $G^+ : L^2(D)^{d_{in}} \to L^2(D)^{d_{out}}$ be continuous such that for all $R > 0$ there is $M > 0$ so that*

$$\big\| G^+(a) \big\|_{L^2(D)^{d_{out}}} \leq M, \ \forall a \in L^2(D)^{d_{in}}, \ \|a\|_{L^2(D)^{d_{in}}} \leq R, \tag{3.3}$$

*We assume that either (i) $\sigma \in A_0^L \cap \text{BA}$ is injective, or (ii) $\sigma = \text{ReLU}$. Then, for any compact set $K \subset L^2(D)^{d_{in}}$, $\epsilon \in (0,1)$, there exists $L \in \mathbb{N}$ and $G \in \text{NO}_L^{inj}(\sigma; D, d_{in}, d_{out})$ such that*

$$\sup_{a \in K} \left\| G^+(a) - G(a) \right\|_{L^2(D)^{d_{out}}} \leq \epsilon.$$

See Section C.3 in Appendix C for the proof. For the definitions of $A_0^L$ and BA, see Definition 3 in Appendix C. For example, ReLU and Leaky ReLU functions belong to $A_0^L \cap \text{BA}$ (see Remark 5 (i)).

We briefly remark on the proof of Theorem 1 emphasizing how its proof differs from a straightforward extension of the finite-rank case. In the proof we first employ the standard universal approximation theorem for neural operators ([Kovachki et al., 2021b, Theorem 11]). We denote the approximation of $G^+$ by $\widetilde{G}$, and define the graph of $\widetilde{G}$ as $H : L^2(D)^{d_{in}} \rightarrow L^2(D)^{d_{in}} \times L^2(D)^{d_{out}}$. That is $H(u) = (u, \widetilde{G}(u))$. Next, utilizing Lemma 1, we construct the projection $Q$ such that $Q \circ H$ becomes an injective approximator of $G^+$ and belongs to $\text{NO}_L(\sigma; D, d_{in}, d_{out})$. The proof for universal approximation theorem is constructive. If, in the future, efficient approximation bounds for neural operators are given, such bounds can likely be used directly in our universality proof to generate corresponding efficient approximation bounds for injective neural operators.

This approach resembles the approach in the finite-rank space Puthawala et al. [2022a, Theorem 15], but unlike that theorem we don't have any dimensionality restrictions. More specifically, in the case of Euclidean spaces $\mathbb{R}^d$, Puthawala et al. [2022a, Theorem 15] requires that $2d_{in} + 1 \leq d_{out}$ before all continuous functions $G^+ : \mathbb{R}^{d_{in}} \rightarrow \mathbb{R}^{d_{out}}$ can be uniformly approximated in compact sets by injective neural networks. When $d_{in} = d_{out} = 1$ this result is not true, as is shown in Remark 5 (iii) in Appendix C using topological degree theory [Cho and Chen, 2006]. In contrast, Theorem 1 does not assume any conditions on $d_{in}$ and $d_{out}$. Therefore, we can conclude that infinite dimensional case yields better approximation results than the finite dimensional case.

This surprising improvement in restrictions in infinite-dimensions can be elucidated by an analogy to Hilbert's hotel paradox, see [Burger and Starbird, 2004, Sec 3.2]. In this analogy, the orthonormal bases $\{\varphi_k\}_{k \in \mathbb{N}}$ and $\Psi_{j,k}(x) = (\delta_{ij}\varphi_k(x))_{i=1}^d$ play the part of guests in the hotel with $\mathbb{N}$ floor, each of which as $d$ rooms. A key step in the proof of Theorem 1 is that there is a linear isomorphism $S : L^2(D)^d \rightarrow L^2(D)$ (i.e., a rearrangement of guests) which maps $\Psi_{j,k}$ to $\varphi_{b(j,k)}$, where $b : [d] \times \mathbb{N} \rightarrow \mathbb{N}$ is a bijection.

### 3.3 Injectivity-preserving transfer to Euclidean spaces via finite-rank approximation

In the previous section, we have discussed injective integral neural operators. The conditions are given in terms of integral kernels, but such kernels may not actually be implementable with finite width and depth networks, which have a finite representational power. A natural question to ask, therefore, is how should these formal integral kernels be implemented on actual finite rank networks, the so-called implementable neural operators? In this section we discuss this question.

We consider linear integral operators $K_\ell$ with $L^2$-integral kernels $k_\ell(x,y)$. Let $\{\varphi_k\}_{k \in \mathbb{N}}$ be an orthonormal basis in $L^2(D)$. Since $\{\varphi_k(y)\varphi_p(x)\}_{k,p \in \mathbb{N}}$ is an orthonormal basis of $L^2(D \times D)$, integral kernels $k_\ell \in L^2(D \times D; \mathbb{R}^{d_{\ell+1} \times d_\ell})$ in integral operators $K_\ell \in \mathcal{L}(L^2(D)^{d_\ell}, L^2(D)^{d_{\ell+1}})$ has the expansion

$$k_\ell(x,y) = \sum_{k,p \in \mathbb{N}} C_{k,p}^{(\ell)} \varphi_k(y)\varphi_p(x),$$

where $C_{k,p}^{(\ell)} \in \mathbb{R}^{d_{\ell+1} \times d_\ell}$ whose $(i,j)$-th component $c_{k,p,ij}^{(\ell)}$ is given by

$$c_{k,p,ij}^{(\ell)} = (k_{\ell,ij}, \varphi_k\varphi_p)_{L^2(D \times D)},$$

Here, we denote $(u, \varphi_k) \in \mathbb{R}^{d_\ell}$ by $(u, \varphi_k) = \left((u_1, \varphi_k)_{L^2(D)}, ..., (u_{d_\ell}, \varphi_k)_{L^2(D)}\right)$. By truncating by $N$ finite sums, we approximate $L^2$-integral operators $K_\ell \in \mathcal{L}(L^2(D)^{d_\ell}, L^2(D)^{d_{\ell+1}})$ by finite rank operator $K_{\ell,N}$ with rank $N$, having the form

$$K_{\ell,N}u(x) = \sum_{k,p \in [N]} C_{k,p}^{(\ell)}(u, \varphi_k)\varphi_p(x), \ u \in L^2(D)^{d_\ell}.$$

The choice of orthonormal basis $\{\varphi_k\}_{k\in\mathbb{N}}$ is a hyperparameter. If we choose $\{\varphi_k\}_k$ as Fourier basis and wavelet basis, then network architectures correspond to Fourier Neural Operators (FNOs) [Li et al., 2020b] and Wavelet Neural Operators (WNOs) [Tripura and Chakraborty, 2023], respectively. We show that Propositions 1, 2 (characterization of layerwise injectivity), and Lemma 1 (global injectivity) all have natural analogues for finite rank operator $K_{\ell,N}$ in Proposition 5 and Lemma 2 in Appendix D. These conditions applies out-of-the-box to both FNOs and WNOs.

Lemma 2 and Remark 3 in the appendix D give a 'recipe' to construct the projection $B$ such that the composition $B \circ T$ (interpreted as augmenting finite rank neural operator $T$ with one layer $B$) is injective. The projection $B$ is constructed by using an orthogonal sequence $\{\xi_k\}_{k\in\mathbb{N}}$ subject to the condition (3.1), which does not over-leap the range of $T$. This condition is automatically satisfied for any orthogonal base $\{\varphi_k\}_{k\in\mathbb{N}}$. This could yield practical implications in guiding the choice of the orthogonal basis $\{\varphi_k\}_{k\in\mathbb{N}}$ for the neural operator's design.

We also show the universal approximation in the case of finite rank approximation. We denote $\mathrm{NO}_{L,N}(\sigma; D, d_{in}, d_{out})$ by the set of integral neural operators with $N$ rank, that is the set (3.2) replacing $L^2$-integral kernel operators $K_\ell$ with finite rank operators $K_{\ell,N}$ with rank $N$ (see Definition 4). We define by

$$\mathrm{NO}_{L,N}^{inj}(\sigma; D, d_{in}, d_{out}) := \{G_N \in \mathrm{NO}_{L,N}(\sigma; D, d_{in}, d_{out}) : G_N \text{ is injective}\}.$$

**Theorem 2.** *Let $D \subset \mathbb{R}^d$ be a Lipschitz bounded domain, and $N \in \mathbb{N}$, and $G^+ : L^2(D)^{d_{in}} \to L^2(D)^{d_{out}}$ be continuous with boundedness as in (3.3). Assume that the non-linear activation function $\sigma$ is either ReLU or Leaky ReLU. Then, for any compact set $K \subset L^2(D)^{d_{in}} \cap (\mathrm{span}\{\varphi_k\}_{k \leq N})^{d_{in}}$, $\epsilon \in (0,1)$, there exists $L \in \mathbb{N}$, $N' \in \mathbb{N}$ with*

$$N' d_{out} \geq 2N d_{in} + 1, \tag{3.4}$$

*and $G_{N'} \in \mathrm{NO}_{L,N'}^{inj}(\sigma; D, d_{in}, d_{out})$ such that*

$$\sup_{a\in K} \left\| G^+(a) - G_{N'}(a) \right\|_{L^2(D)^{d_{out}}} \leq \epsilon.$$

See Section D.4 in Appendix D for the proof. In the proof, we make use of Puthawala et al. [2022a, Lemma 29], which gives rise to the assumption (3.4). We do not require any condition on $d_{in}$ and $d_{out}$ as well as Theorem 1.

**Remark 1.** *Observe that in our finite-rank approximation result, we only require that the target function G+ is continuous and bounded, but not smooth. This differs from prior work that requires smoothness of the function to be approximated.*

### 3.4  Limitations

We assume square matrices for the bijectivity and construction of inversions. Weight matrices in actual neural operators are not necessarily square. Lifting and projection are local operators which map the inputs into a higher-dimensional feature space and project back the feature output to the output space. We also haven't addressed possible aliasing effects of our injective operators. We will relax the square assumption and investigate the aliasing of injective operators in future work.

## 4  Subnetworks & nonlinear integral operators: bijectivity and inversion

So far our analysis of injectivity has been restricted to the case where the only source of nonlinearity are the activation functions. In this section we consider a weaker and more abstract problem where nonlinearities can also arise from the integral kernel with surjective activation function, such as leaky ReLU. Specifically, we consider layers of the form

$$F_1(u) = Wu + K(u), \tag{4.1}$$

where $W \in \mathcal{L}(L^2(D)^n, L^2(D)^n)$ is a linear bounded bijective operator, and $K : L^2(D)^n \to L^2(D)^n$ is a non-linear operator. This arises in the non-linear neural operator construction by Kovachki et al. [2021b] or in Ong et al. [2022] to improve performance of integral autoencoders. In this construction, each layer $\mathcal{L}_\ell$ is written as

$$x \in D, \quad (\mathcal{L}_\ell v)(x) = \sigma(W_\ell v(x) + K_\ell(v)(x)), \quad K_\ell(u)(x) = \int_D k_\ell(x, y, u(x), u(y)) u(y) dy,$$

where $W_\ell \in \mathbb{R}^{d_{\ell+1} \times d_\ell}$ independent of $x$, and $K_\ell : L^2(D)^{d_\ell} \to L^2(D)^{d_{\ell+1}}$ is the non-linear integral operator.

This relaxing of assumptions is motivated by a desire to obtain theoretical results for both subnetworks and operator transformers. By subnetworks, we mean compositions of layers within a network. This includes, for example, the encoder or decoder block of a traditional VAE. By neural operator we mean operator generalizations of finite-rank transformers, which can be modeled by letting $K$ be an integral transformation with nonlinear kernel $k$ of the form

$$k(x, y, v(x), v(y)) \equiv \text{softmax} \circ \langle Av(x), Bv(y) \rangle,$$

where $A$ and $B$ are matrices of free parameters, and the (integral) softmax is taken over $x$. This specific choice of $k$ can be understood as a natural generalization of the attention mechanism in transformers, see [Kovachki et al., 2021b, Sec. 5.2] for further details.

## 4.1 Surjectivity and bijectivity

Critical to our analysis is the notion of coercivity. Apart from being a useful theoretical tool, layerwise coercivity of neural networks is a useful property in imaging applications, see e.g.Li et al. [2020a] Recall from Showalter [2010, Sec 2, Chap VII] that a non-linear operator $K : L^2(D)^n \to L^2(D)^n$ is coercive if

$$\lim_{\|u\|_{L^2(D)^n} \to \infty} \langle K(u), \frac{u}{\|u\|_{L^2(D)^n}} \rangle_{L^2(D)^n} = \infty. \tag{4.2}$$

**Proposition 3.** *Let $\sigma : \mathbb{R} \to \mathbb{R}$ be surjective and $W : L^2(D)^n \to L^2(D)^n$ be linear bounded bijective (then the inverse $W^{-1}$ is bounded linear), and let $K : L^2(D)^n \to L^2(D)^n$ be a continuous and compact mapping. Moreover, assume that the map $u \mapsto \alpha u + W^{-1}K(u)$ is coercive with some $0 < \alpha < 1$. Then, the operator $\sigma \circ F_1$ is surjective.*

See Section E.1 in Appendix E for the proof. An example $K$ satisfying the coercivity (4.2) condition is given here.

**Example 1.** *We simply consider the case of $n = 1$, and $D \subset \mathbb{R}^d$ is a bounded interval. We consider the non-linear integral operator*

$$K(u)(x) := \int_D k(x, y, u(x))u(y)dy, \ x \in D.$$

*The operator $u \mapsto \alpha u + W^{-1}K(u)$ with some $0 < \alpha < 1$ is coercive when the non-linear integral kernel $k(x, y, t)$ satisfies certain boundedness conditions. In Examples 2 and 3 in Appendix E, we show that these conditions are met by kernels $k(x, y, t)$ of the form*

$$k(x, y, t) = \sum_{j=1}^{J} c_j(x, y)\sigma(a_j(x, y)t + b_j(x, y)),$$

*where $\sigma : \mathbb{R} \to \mathbb{R}$ is the sigmoid function $\sigma_s : \mathbb{R} \to \mathbb{R}$, and $a, b, c \in C(\overline{D} \times \overline{D})$ or by a wavelet activation function $\sigma_{wire} : \mathbb{R} \to \mathbb{R}$, see Saragadam et al. [2023], where $\sigma_{wire}(t) = Im\left(e^{i\omega t}e^{-t^2}\right)$ and $a, b, c \in C(\overline{D} \times \overline{D})$, and $a_j(x, y) \neq 0$.*

In the proof of Proposition 3, we utilize the Leray-Schauder fix point theorem. By employing the Banach fix point theorem under a contraction mapping condition (4.3), we can obtain bijectivity as follows:

**Proposition 4.** *Let $\sigma : \mathbb{R} \to \mathbb{R}$ be bijective. Let $W : L^2(D)^n \to L^2(D)^n$ be bounded linear bijective, and let $K : L^2(D)^n \to L^2(D)^n$. If $W^{-1}K : L^2(D)^n \to L^2(D)^n$ is a contraction mapping, that is, there exists $\rho \in (0, 1)$ such that*

$$\left\|W^{-1}K(u) - W^{-1}K(v)\right\| \le \rho \left\|u - v\right\|, \ u, v \in L^2(D)^n, \tag{4.3}$$

*then, the operator $\sigma \circ F_1$ is bijective.*

See Section E.3 in Appendix E for the proof. We note that if $K$ is compact linear, then assumption (4.3) implies that $W + K$ is an injective Fredholm operator with index zero, which is equivalent to

$\sigma \circ (W + K)$ being bijective as observed in Proposition 2. That is, Proposition 4 requires stronger assumptions when applied to the linear case.

Assumption (4.3) implies the the injectivity of $K$. An interesting example of injective operators arises when $K$ are Volterra operators. When $D \subset \mathbb{R}^d$ is bounded and $K(u) = \int_D k(x, y, u(y))u(y)dy$, where we denote $x = (x_1, \ldots, x_d)$ and $y = (y_1, \ldots, y_d)$, we recall that $K$ is a Volterra operator if $k(x, y, t) \neq 0$ implies $y_j \leq x_j$ for all $j = 1, 2, \ldots, d$. A well known fact, as discussed in Example 4 in Appendix E, is that if $K(u)$ is a Volterra operator whose kernel $k(x, y, t) \in C(\overline{D} \times \overline{D} \times \mathbb{R}^n)$ is bounded and uniformly Lipschitz in $t$-variable then $F : u \mapsto u + K(u)$ is injective.

**Remark 2.** *A similar analysis (illustrating coercivity) shows that operators of the following two forms are bijective.*

1. *Operators of the form $F(U) := \alpha u + K(u)$ where $K(u) := \int_D a(x, y, u(x), u(y))dy$ where $a(x, y, s_1, s_2)$ is continuous and is such that $\exists R > 0$, $c_1 < \alpha$ so that for all $|(s_1, s_2)| > R$, $\mathrm{sign}(s_1)\mathrm{sign}(s_2)a(x, y, s_1, s_2) \geq -c_1$.*

2. *Layers of the form $(\mathcal{L}_\ell v)(x) := \sigma_1(Wu + \sigma_2(K(u)))$ where $\sigma_1$ is bijective and $\sigma_2$ bounded.*

*Finally we remark that coercivity is bounded preserved by perturbations in a bounded domain. This makes it possible to study non-linear and non-positive perturbations of physical models. For example, in quantum mechanics when a non-negative energy potential $|\phi|^4$ is replaced by a Mexican hat potential $-C|\phi|^2 + |\phi|^4$, as occurs in the study of magnetization, superconductors and the Higgs field.*

## 4.2 Construction of the inverse of a non-linear integral neural operator

The preceding sections clarified sufficient conditions for surjectivity and bijectivity of the non-linear operator $F_1$. We now consider how to construct the inverse of $F_1$ in a compact set $\mathcal{Y}$. We find that constructing inverses is possible in a wide variety of settings and, moreover, that the inverses themselves can be given by neural operators. The proof that neural operators may be inverted with other neural operators provides a theoretical justification for Integral Auto Encoder networks [Ong et al., 2022] where an infinite encoder/decoder pair play a role parallel to those of encoder/decoders in finite-dimensional VAEs [Kingma and Welling, 2013]. This section proves that the decoder half of a IAE-net is provably able to inverse the encoder half. Our analysis also shows that injective differential operators (as arise in PDEs) and integral operator encoders form a formal algebra under operator composition. We prove this in the rest of this section, but first we summarize the main three steps of the proof.

First, by using the Banach fixed point theorem and invertibility of derivatives of $F_1$ we show that, locally, $F_1$ may be inverted by an iteration of a contractive operator near $g_j = F_1(v_j)$. This makes local inversion simple in balls which cover the set $\mathcal{Y}$. Second, we construct partition of unity functions $\Phi_j$ that masks the support of each covering ball and allows us to construct one global inverse that simply passes through to the local inverse on the appropriate ball. Third and finally, we show that each function used in both of the above steps are representable using neural operators with distributional kernels.

As a simple case, let us first consider the case when $n = 1$, and $D \subset \mathbb{R}$ is a bounded interval, and the operator $F_1$ of the form

$$F_1(u)(x) = W(x)u(x) + \int_D k(x, y, u(y))u(y)dy,$$

where $W \in C^1(\overline{D})$ satisfies $0 < c_1 \leq W(x) \leq c_2$ and the function $(x, y, s) \mapsto k(x, y, s)$ is in $C^3(\overline{D} \times \overline{D} \times \mathbb{R})$ and in $\overline{D} \times \overline{D} \times \mathbb{R}$ its three derivatives and the derivatives of $W$ are uniformly bounded by $c_0$, that is,

$$\|k\|_{C^3(\overline{D} \times \overline{D} \times \mathbb{R})} \leq c_0, \quad \|W\|_{C^1(\overline{D})} \leq c_0. \tag{4.4}$$

The condition (4.4) implies that $F_1 : H^1(D) \to H^1(D)$, contains locally Lipschitz smooth functions. Furthermore, $F_1 : H^1(D) \to H^1(D)$ is Fréchet differentiable at $u_0 \in C(\overline{D})$, and we denote Fréchet

derivative of $F_1$ at $u_0$ by $A_{u_0}$, which can be written as the integral operator (F.2). We will assume that for all $u_0 \in C(\overline{D})$, the integral operator

$$A_{u_0} : H^1(D) \to H^1(D) \text{ is an injective operator.} \tag{4.5}$$

This happens for example when $K(u)$ is a Volterra operator, see Examples 4 and 5. As the integral operators $A_{u_0}$ are Fredholm operators having index zero, this implies that the operators (4.5) are bijective. The inverse operator $A_{u_0}^{-1} : H^1(D) \to H^1(D)$ can be written by the integral operator (F.5).

We will consider the inverse function of the map $F_1$ in $\mathcal{Y} \subset \sigma_a(\overline{B}_{C^{1,\alpha}(\overline{D})}(0,R)) = \{\sigma_a \circ g \in C(\overline{D}) : \|g\|_{C^{1,\alpha}(\overline{D})} \le R\}$, which is a set of the image of Hölder spaces $C^{n,\alpha}(\overline{D})$ through (leaky) ReLU-type functions $\sigma_a(s) = \mathrm{ReLU}(s) - a\,\mathrm{ReLU}(-s)$ with $a \ge 0$. We note that $\mathcal{Y}$ is a compact subset the Sobolev space $H^1(D)$, and we use the notations $B_{\mathcal{X}}(g,R)$ and $\overline{B}_{\mathcal{X}}(g,R)$ as open and closed balls with radius $R > 0$ at center $g \in \mathcal{X}$ in Banach space $\mathcal{X}$.

To this end, we will cover the set $\mathcal{Y}$ with small balls $B_{H^1(D)}(g_j, \varepsilon_0)$, $j = 1, 2, \ldots, J$ of $H^1(D)$, centered at $g_j = F_1(v_j)$, where $v_j \in H^1(D)$. As considered in detail in Appendix F, when $g$ is sufficiently near to the function $g_j$ in $H^1(D)$, the inverse map of $F_1$ can be written as a limit $(F_1^{-1}(g), g) = \lim_{m\to\infty} \mathcal{H}_j^{\circ m}(v_j, g)$ in $H^1(D)^2$, where

$$\mathcal{H}_j \begin{pmatrix} u \\ g \end{pmatrix} := \begin{pmatrix} u - A_{v_j}^{-1}(F_1(u) - F_1(v_j)) + A_{v_j}^{-1}(g - g_j) \\ g \end{pmatrix},$$

that is, near $g_j$ we can approximate $F_1^{-1}$ as a composition $\mathcal{H}_j^{\circ m}$ of $2m$ layers of neural operators.

To glue the local inverse maps together, we use a partition of unity $\Phi_{\vec{i}}$, $\vec{i} \in \mathcal{I}$ in the function space $\mathcal{Y}$, where $\mathcal{I} \subset \mathbb{Z}^{\ell_0}$ is a finite index set. The function $\Phi_{\vec{i}}$ are given by neural operators

$$\Phi_{\vec{i}}(v,w) = \pi_1 \circ \phi_{\vec{i},1} \circ \phi_{\vec{i},2} \circ \cdots \circ \phi_{\vec{i},\ell_0}(v,w), \quad \text{where} \quad \phi_{\vec{i},\ell}(v,w) = (F_{y_\ell, s(\vec{i},\ell),\epsilon_1}(v,w), w),$$

where some $\epsilon_1 > 0$, $s(\vec{i},\ell) \in \mathbb{R}$ are some suitable values near $g_{j(\vec{i})}(y_\ell)$, some $y_\ell \in D$ ($\ell = 1, ..., \ell_0$), and $\pi_1$ is the map $\pi_1(v,w) = v$ that maps a pair $(v,w)$ to the first function $v$. Here, $F_{z,s,h}(v,w)$ are integral neural operators with distributional kernels $F_{z,s,h}(v,w)(x) = \int_D k_{z,s,h}(x,y,v(x),w(y))dy$, where $k_{z,s,h}(x,y,v(x),w(y)) = v(x)\mathbf{1}_{[s-\frac{1}{2}h,s+\frac{1}{2}h]}(w(y))\delta(y-z)$, and $\mathbf{1}_A$ is a indicator function of a set $A$ and $y \mapsto \delta(y-z)$ is the Dirac delta distribution at the point $z \in D$. Using these, we can write the inverse of $F_1$ at $g \in \mathcal{Y}$ as

$$F_1^{-1}(g) = \lim_{m\to\infty} \sum_{\vec{i}\in\mathcal{I}} \Phi_{\vec{i}} \mathcal{H}_{j(\vec{i})}^{\circ m} \begin{pmatrix} v_{j(\vec{i})} \\ g \end{pmatrix} \quad \text{in } H^1(D) \tag{4.6}$$

where $j(\vec{i}) \in \{1, 2, \ldots, J\}$.

This result is summarized in following theorem which is proven in Appendix F.

**Theorem 3.** *Assume that $F_1$ satisfies the above assumptions (4.4) and (4.5) and that $F_1 : H^1(D) \to H^1(D)$ is a bijection. Let $\mathcal{Y} \subset \sigma_a(\overline{B}_{C^{1,\alpha}(\overline{D})}(0,R))$ be a compact subset of the Sobolev space $H^1(D)$, where $\alpha > 0$ and $a \ge 0$. Then the inverse of $F_1 : H^1(D) \to H^1(D)$ in $\mathcal{Y}$ can written as a limit (4.6) that is, as a limit of integral neural operators.*

## 5 Discussion and Conclusion

In this paper, we provided a theoretical analysis of injectivity and bijectivity for neural operators. In the future, we will further develop applications of our theory, particularly in the areas of generative models and inverse problems and integral autoencoders. We gave a rigorous framework for the analysis of the injectivity and bijectivity of neural operators including when either the ReLU activation or bijective activation functions are used. We further proved that injective neural operators are universal approximators and their finite-rank implementation are still injective. Finally, we ended by considering the 'coarser' problem of non-linear integral operators, as arises in subnetworks, operator transformers and integral autoencoders.

## Acknowledgments

ML was partially supported by Academy of Finland, grants 273979, 284715, 312110. M.V. de H. was supported by the Simons Foundation under the MATH + X program, the National Science Foundation under grant DMS-2108175, and the corporate members of the Geo-Mathematical Imaging Group at Rice University.

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

# Appendix

## A    Motivation behind our injectivity & bijectivity

**PDEs-based inverse problems:**    We consider the following partial differential equations (PDEs) of the form

$$\mathcal{L}_a u(x) = f(x), \quad x \in D,$$
$$\mathcal{B}u(x) = g(x), \quad x \in \partial D,$$

where $D \subset \mathbb{R}^d$ is a bounded domain, $\mathcal{L}_a$ is partial differential operator with a coefficient $a \in \mathcal{A}(D; \mathbb{R}^{d_a}) := \{a : D \to \mathbb{R}^{d_a}\}$, and $\mathcal{B}$ is some boundary operator, e.g. Dirichlet or Neumann, and function $f$ & $g$ are fixed. When a solution $u \in \mathcal{U}(D; \mathbb{R}^{d_u}) := \{u : D \to \mathbb{R}^{d_a}\}$ is uniquely determined, we may define the solution operator $G : \mathcal{A}(D; \mathbb{R}^{d_a}) \to \mathcal{U}(D; \mathbb{R}^{d_u})$ by putting $G(a) := u$. Note that the operator $G$ is, in general, non-linear even if partial differential Operator $\mathcal{L}_a$ is linear (e.g., $\mathcal{L}_a = \Delta + a$). The aim of the inverse problem is to evaluate $G^{-1}$, the inverse of the solution operator. When $G$ is non-injective, the problem is termed *ill-posed*.

The key link between PDE and our injective neural operators is as follows. Can $G$ be approximated by injective neural operators $N_{inj}$? If so, then $N_{inj}$ can be a surrogate model to solve an ill-posed inverse problem. In general, this is possible, even if $G$ is non-injective, per the results in Section 3.

Moreover, by the results of Section 4, we may conduct the inverse $N_{inj}^{-1}$ of surrogate model $N_{inj}$ with another neural operator.

**Approximation of pushes forward measure:**    Let $\mathcal{M}$ be the submanifold of $X = L^2([0,1]^2)$ or $X = L^2([0,1]^3)$ corresponding to natural images or 3D medical model. Let $K \subset \mathbb{R}^D$ be a manifold with the same topology as $\mathcal{M}$, $\iota : \mathbb{R}^D \to X$ be an embedding, and let $K_1 = \iota(K) \subset \mathcal{M}$. Given $\mu$, a measure supported on $M$, the task is to find a neural operator $f_\theta : X \to X$ that maps (pushes forward) the uniform distribution on the model space $K_1$ to $\mu$ and so thus maps $K_1$ to $\mathcal{M}$. If $f_\theta : X \to X$ is bijective, computing likelihood functions in statistical analysis is made easier via the change of variables formula. Further, we may interpret $f_\theta^{-1}$ as an encoder and $f_\theta$ as the corresponding decoder, which parameterized elements of $\mathcal{M}$. As everything is formulated in infinite dimension function space $X$, we obtain discretization invariant methods.

## B    Proof of Proposition 1 in Section 2

*Proof.* We use the notation $T|_{S(v,T+b)}(v) = (T_i v)_{i \in S(v,T+b)}$. Assume that $T + b$ has a DSS with respect to every $v \in L^2(D)^n$ in the sense of Definition 2, and that

$$\text{ReLU}(Tv^{(1)} + b) = \text{ReLU}(Tv^{(2)} + b) \quad \text{in } D, \tag{B.1}$$

where $v^{(1)}, v^{(2)} \in L^2(D)^n$. Since $T+b$ has a DSS with respect to $v^{(1)}$, we have for $i \in S(v^{(1)}, T+b)$

$$0 < \text{ReLU}(T_i v^{(1)} + b_i) = \text{ReLU}(T_i v^{(2)} + b_i) \text{ in } D,$$

which implies that

$$T_i v^{(1)} + b_i = T_i v^{(2)} + b_i \text{ in } D.$$

Thus,

$$v^{(1)} - v^{(2)} \in \text{Ker}\left(T|_{S(v^{(1)}, T+b)}\right). \tag{B.2}$$

By assuming (B.1), we have for $i \notin S(v^{(1)}, T)$,

$$\{x \in D \mid T_i v^{(1)}(x) + b_i(x) \leq 0\} = \{x \in D \mid T_i v^{(2)}(x) + b_i(x) \leq 0\}.$$

Then, we have

$$T_i(v^{(1)} - (v^{(1)} - v^{(2)}))(x) + b_i(x) = T_i v^{(2)}(x) + b_i(x) \leq 0 \text{ if } T_i v^{(1)}(x) + b_i(x) \leq 0,$$

that is,

$$T_i v^{(1)}(x) + b_i(x) \leq T_i\left(v^{(1)} - v^{(2)}\right)(x) \text{ if } T_i v^{(1)}(x) + b_i(x) \leq 0.$$

In addition,

$$T_i(v^{(1)} - v^{(2)})(x) = T_i v^{(1)}(x) + b_i(x) - \left(Tv^{(2)}(x) + b_i(x)\right) = 0 \text{ if } T_i v^{(1)}(x) + b_i(x) > 0.$$

Thus,

$$v^{(1)} - v^{(2)} \in X(v, T + b). \tag{B.3}$$

Combining (B.2) and (B.3), and (2.1) as $v = v^{(1)}$, we conclude that

$$v^{(1)} - v^{(2)} = 0.$$

Conversely, assume that there exists a $v \in L^2(D)^n$ such that

$$\text{Ker}\left(T\big|_{S(v, T+b)}\right) \cap X(v, T + b) \neq \{0\}.$$

Then there is $u \neq 0$ such that

$$u \in \text{Ker}\left(T\big|_{S(v, T+b)}\right) \cap X(v, T + b).$$

For $i \in S(v, T + b)$, we have by $u \in \text{Ker}(T_i)$,

$$\text{ReLU}\left(T_i(v - u) + b_i(x)\right) = \text{ReLU}\left(T_i v + b_i(x)\right).$$

For $i \notin S(v, T + b)$, we have by $u \in X(v, T + b)$,

$$\text{ReLU}\left(T_i(v - u)(x) + b_i(x)\right) = \begin{cases} 0 & \text{if } T_i v(x) + b_i(x) \leq 0 \\ T_i v(x) + b_i(x) & \text{if } T_i v(x) + b_i(x) > 0 \end{cases}$$
$$= \text{ReLU}\left(T_i v(x) + b_i(x)\right).$$

Therefore, we conclude that

$$\text{ReLU}\left(T(v - u) + b\right) = \text{ReLU}\left(Tv + b\right),$$

where $u \neq 0$, that is, $\text{ReLU} \circ (T + b)$ is not injective. $\qquad\square$

## C  Details in Sections 3.1 and 3.2

### C.1  Proof of Lemma 1

*Proof.* The restriction operator, $\pi_\ell : L^2(D)^m \to L^2(D)^\ell$ ($\ell < m$), acts as follows,

$$\pi_\ell(a, b) := b, \quad (a, b) \in L^2(D)^{m-\ell} \times L^2(D)^\ell. \tag{C.1}$$

Since $L^2(D)$ is a separable Hilbert space, there exists an orthonormal basis $\{\varphi_k\}_{k \in \mathbb{N}}$ in $L^2(D)$. We denote by

$$\varphi_{k,j}^0 := \left(0, ..., 0, \underbrace{\varphi_k}_{j-th}, 0, ..., 0\right) \in L^2(D)^m,$$

for $k \in \mathbb{N}$ and $j \in [m - \ell]$. Then, $\{\varphi_{k,j}^0\}_{k \in \mathbb{N}, j \in [m-\ell]}$ is an orthonormal sequence in $L^2(D)^m$, and

$$V_0 := L^2(D)^{m-\ell} \times \{0\}^\ell$$
$$= \text{span}\left\{\varphi_{k,j}^0 \mid k \in \mathbb{N},\ j \in [m - \ell]\right\}.$$

We define, for $\alpha \in (0, 1)$,

$$\varphi_{k,j}^\alpha := \left(0, ..., 0, \underbrace{\sqrt{(1-\alpha)}\varphi_k}_{j-th}, 0, ..., 0, \sqrt{\alpha}\xi_{(k-1)(m-\ell)+j}\right) \in L^2(D)^m, \tag{C.2}$$

with $k \in \mathbb{N}$ and $j \in [m - \ell]$. We note that $\{\varphi_{k,j}^\alpha\}_{k \in \mathbb{N}, j \in [m-\ell]}$ is an orthonormal sequence in $L^2(D)^m$. We set

$$V_\alpha := \text{span}\left\{\varphi_{k,j}^\alpha \mid k \in \mathbb{N},\ j \in [m - \ell]\right\}. \tag{C.3}$$

It holds for $0 < \alpha < 1/2$ that

$$\left\| P_{V_\alpha^\perp} - P_{V_0^\perp} \right\|_{op} < 1.$$

Indeed, for $u \in L^2(D)^m$ and $0 < \alpha < 1/2$,

$$\left\| P_{V_\alpha^\perp} u - P_{V_0^\perp} u \right\|_{L^2(D)^m}^2 = \| P_{V_\alpha} u - P_{V_0} u \|_{L^2(D)^m}^2$$

$$= \left\| \sum_{k \in \mathbb{N}, j \in [m-\ell]} (u, \varphi_{k,j}^\alpha) \varphi_{k,j}^\alpha - (u, \varphi_{k,j}^0) \varphi_{k,j}^0 \right\|_{L^2(D)^m}^2$$

$$= \left\| \sum_{k \in \mathbb{N}, j \in [m-\ell]} (1 - \alpha)(u_j, \varphi_k) \varphi_k - (u_j, \varphi_k) \varphi_k \right\|_{L^2(D)}^2$$

$$+ \left\| \sum_{k \in \mathbb{N}, j \in [m-\ell]} \alpha(u_m, \xi_{(k-1)(m-\ell)+j}) \xi_{(k-1)(m-\ell)+j} \right\|_{L^2(D)}^2$$

$$\leq \alpha^2 \sum_{j \in [m-\ell]} \sum_{k \in \mathbb{N}} |(u_j, \varphi_k)|^2 + \alpha^2 \sum_{k \in \mathbb{N}} |(u_m, \xi_k)|^2 \leq 4\alpha^2 \|u\|_{L^2(D)^m}^2,$$

which implies that $\left\| P_{V_\alpha^\perp} - P_{V_0^\perp} \right\|_{op} \leq 2\alpha$.

We will show that the operator

$$P_{V_\alpha^\perp} \circ T : L^2(D)^n \to L^2(D)^m,$$

is injective. Assuming that for $a, b \in L^2(D)^n$,

$$P_{V_\alpha^\perp} \circ T(a) = P_{V_\alpha^\perp} \circ T(b),$$

is equivalent to

$$T(a) - T(b) = P_{V_\alpha}(T(a) - T(b)).$$

Denoting by $P_{V_\alpha}(T(a) - T(b)) = \sum_{k \in \mathbb{N}, j \in [m-\ell]} c_{k,j} \varphi_{k,j}^\alpha$,

$$\pi_1(T(a) - T(b)) = \sum_{k \in \mathbb{N}, j \in [m-\ell]} c_{k,j} \xi_{(k-1)(m-\ell)+j}.$$

From (3.1), we obtain that $c_{kj} = 0$ for all $k, j$. By injectivity of $T$, we finally get $a = b$.

We define $Q_\alpha : L^2(D)^m \to L^2(D)^m$ by

$$Q_\alpha := \left( P_{V_0^\perp} P_{V_\alpha^\perp} + (I - P_{V_0^\perp})(I - P_{V_\alpha^\perp}) \right) \left( I - (P_{V_0^\perp} - P_{V_\alpha^\perp})^2 \right)^{-1/2}.$$

By the same argument as in Section I.4.6 Kato [2013], we can show that $Q_\alpha$ is injective and

$$Q_\alpha P_{V_\alpha^\perp} = P_{V_0^\perp} Q_\alpha,$$

that is, $Q_\alpha$ maps from $\mathrm{Ran}(P_{V_\alpha^\perp})$ to

$$\mathrm{Ran}(P_{V_0^\perp}) \subset \{0\}^{m-\ell} \times L^2(D)^\ell.$$

It follows that

$$\pi_\ell \circ Q_\alpha \circ P_{V_\alpha^\perp} \circ T : L^2(D)^n \to L^2(D)^\ell$$

is injective. $\qquad\square$

## C.2 Remarks following Lemma 1

**Remark 3.** *Lemma 1 and Eqn. (3.1) may be interpreted as saying that if* some *orthonormal sequence* $\{\xi_k\}_{k\in\mathbb{N}}$ *exists that doesn't overlap the range of* $T$, *then* $T$ *may be embedded in a small space without losing injectivity. An example that satisfies (3.1) is the neural operator whose* $L$-*th layer operator* $\mathcal{L}_L$ *consists of the integral operator* $K_L$ *with continuous kernel function* $k_L$, *and with continuous activation function* $\sigma$. *Indeed, in this case, we may choose the orthogonal sequence* $\{\xi_k\}_{k\in\mathbb{N}}$ *in* $L^2(D)$ *as a discontinuous functions sequence* [1] *so that* $\mathrm{span}\{\xi_k\}_{k\in\mathbb{N}} \cap C(D) = \{0\}$. *Then, by* $\mathrm{Ran}(\mathcal{L}_L) \subset C(D)^{d_L}$, *the assumption (3.1) holds.*

**Remark 4.** *In the proof of Lemma 1, an operator* $B \in \mathcal{L}(L^2(D)^m, L^2(D)^\ell)$,

$$B = \pi_\ell \circ Q_\alpha \circ P_{V_\alpha^\perp},$$

*appears, where* $P_{V_\alpha^\perp}$ *is the orthogonal projection onto orthogonal complement* $V_\alpha^\perp$ *of* $V_\alpha$ *with*

$$V_\alpha := \mathrm{span}\left\{ \varphi_{k,j}^\alpha \,\Big|\, k \in \mathbb{N}, j \in [m - \ell] \right\} \subset L^2(D)^m,$$

*in which* $\varphi_{k,j}^\alpha$ *is defined for* $\alpha \in (0,1)$, $k \in \mathbb{N}$ *and* $j \in [\ell]$,

$$\varphi_{k,j}^\alpha := \left( 0, ..., 0, \underbrace{\sqrt{(1-\alpha)}\varphi_k}_{j-th}, 0, ..., 0, \sqrt{\alpha}\xi_{(k-1)(m-\ell)+j} \right).$$

*Here,* $\{\varphi_k\}_{k\in\mathbb{N}}$ *is an orthonormal basis in* $L^2(D)$. *Futhermore,* $Q_\alpha : L^2(D)^m \to L^2(D)^m$ *is defined by*

$$Q_\alpha := \left( P_{V_0^\perp} P_{V_\alpha^\perp} + (I - P_{V_0^\perp})(I - P_{V_\alpha^\perp}) \right) \left( I - (P_{V_0^\perp} - P_{V_\alpha^\perp})^2 \right)^{-1/2},$$

*where* $P_{V_0^\perp}$ *is the orthogonal projection onto orthogonal complement* $V_0^\perp$ *of* $V_0$ *with*

$$V_0 := L^2(D)^{m-\ell} \times \{0\}^\ell.$$

*The operator* $Q_\alpha$ *is well-defined for* $0 < \alpha < 1/2$ *because it holds that*

$$\left\| P_{V_\alpha^\perp} - P_{V_0^\perp} \right\|_{\mathrm{op}} < 2\alpha.$$

*This construction is given by the combination of "Pairs of projections" discussed in Kato [2013, Section I.4.6] with the idea presented in [Puthawala et al., 2022b, Lemma 29].*

## C.3 Proof of Theorem 1

We begin with

**Definition 3.** *The set of* $L$-*layer neural networks mapping from* $\mathbb{R}^d$ *to* $\mathbb{R}^{d'}$ *is*

$$\mathrm{N}_L(\sigma; \mathbb{R}^d, \mathbb{R}^{d'}) := \Big\{ f : \mathbb{R}^d \to \mathbb{R}^{d'} \Big| f(x) = W_L\sigma(\cdots W_1\sigma(W_0 x + b_0) + b_1 \cdots) + b_L,$$

$$W_\ell \in \mathbb{R}^{d_{\ell+1} \times d_\ell}, b_\ell \in \mathbb{R}^{d_{\ell+1}}, d_\ell \in \mathbb{N}_0(d_0 = d, \ d_{L+1} = d'), \ell = 0, ..., L \Big\},$$

*where* $\sigma : \mathbb{R} \to \mathbb{R}$ *is an element-wise nonlinear activation function. For the class of nonlinear activation functions,*

$$\mathrm{A}_0 := \Big\{ \sigma \in C(\mathbb{R}) \Big| \exists n \in \mathbb{N}_0 \text{ s.t. } \mathrm{N}_n(\sigma; \mathbb{R}^d, \mathbb{R}) \text{ is dense in } C(K) \text{ for } \forall K \subset \mathbb{R}^d \text{ compact} \Big\}$$

$$\mathrm{A}_0^L := \Big\{ \sigma \in A_0 \Big| \sigma \text{ is Borel measurable s.t. } \sup_{x\in\mathbb{R}} \frac{|\sigma(x)|}{1+|x|} < \infty \Big\}$$

---

[1] e.g., step functions whose supports are disjoint for each sequence.

$$\text{BA} := \Big\{ \sigma \in A_0 \Big| \forall K \subset \mathbb{R}^d \ compact, \forall \epsilon > 0, \ and \ \forall C \geq \text{diam}(K), \exists n \in \mathbb{N}_0,$$

$$\exists f \in \text{N}_n(\sigma; \mathbb{R}^d, \mathbb{R}^d) \ s.t. \ |f(x) - x| \leq \epsilon, \ \forall x \in K, \ and, \ |f(x)| \leq C, \ \forall x \in \mathbb{R}^d \Big\}.$$

*The set of integral neural operators with $L^2$-integral kernels is*

$$\text{NO}_L(\sigma; D, d_{in}, d_{out}) := \Big\{ G : L^2(D)^{d_{in}} \rightarrow L^2(D)^{d_{out}} \Big|$$

$$G = K_{L+1} \circ (K_L + b_L) \circ \sigma \cdots \circ (K_2 + b_2) \circ \sigma \circ (K_1 + b_1) \circ (K_0 + b_0),$$

$$K_\ell \in \mathcal{L}(L^2(D)^{d_\ell}, L^2(D)^{d_{\ell+1}}), \ K_\ell : f \mapsto \int_D k_\ell(\cdot, y) f(y) dy \Big|_D, \tag{C.4}$$

$$k_\ell \in L^2(D \times D; \mathbb{R}^{d_{\ell+1} \times d_\ell}), \ b_\ell \in L^2(D; \mathbb{R}^{d_{\ell+1}}),$$

$$d_\ell \in \mathbb{N}, \ d_0 = d_{in}, \ d_{L+2} = d_{out}, \ \ell = 0, ..., L+2 \Big\}.$$

*Proof.* Let $R > 0$ such that

$$K \subsetneq B_R(0),$$

where $B_R(0) := \{u \in L^2(D)^{d_{in}} \mid \|u\|_{L^2(D)^{d_{in}}} \leq R\}$. By Theorem 11 of Kovachki et al. [2021b], there exists $L \in \mathbb{N}$ and $\widetilde{G} \in \text{NO}_L(\sigma; D, d_{in}, d_{out})$ such that

$$\sup_{a \in K} \left\| G^+(a) - \widetilde{G}(a) \right\|_{L^2(D)^{d_{out}}} \leq \frac{\epsilon}{2}, \tag{C.5}$$

and

$$\left\| \widetilde{G}(a) \right\|_{L^2(D)^{d_{out}}} \leq 4M, \quad \text{for } a \in L^2(D)^{d_{in}}, \quad \|a\|_{L^2(D)^{d_{in}}} \leq R.$$

We write operator $\widetilde{G}$ by

$$\widetilde{G} = \widetilde{K}_{L+1} \circ (\widetilde{K}_L + \widetilde{b}_L) \circ \sigma \cdots \circ (\widetilde{K}_2 + \widetilde{b}_2) \circ \sigma \circ (\widetilde{K}_1 + \widetilde{b}_1) \circ (\widetilde{K}_0 + \widetilde{b}_0),$$

where

$$\widetilde{K}_\ell \in \mathcal{L}(L^2(D)^{d_\ell}, L^2(D)^{d_{\ell+1}}), \ \widetilde{K}_\ell : f \mapsto \int_D \widetilde{k}_\ell(\cdot, y) f(y) dy,$$

$$\widetilde{k}_\ell \in C(D \times D; \mathbb{R}^{d_{\ell+1} \times d_\ell}), \ \widetilde{b}_\ell \in L^2(D; \mathbb{R}^{d_{\ell+1}}),$$

$$d_\ell \in \mathbb{N}, \ d_0 = d_{in}, \ d_{L+2} = d_{out}, \ \ell = 0, ..., L+2.$$

We remark that kernel functions $\widetilde{k}_\ell$ are continuous because neural operators defined in Kovachki et al. [2021b] parameterize the integral kernel function by neural networks, thus,

$$\text{Ran}(\widetilde{G}) \subset C(D)^{d_{out}}. \tag{C.6}$$

We define the neural operator $H : L^2(D)^{d_{in}} \rightarrow L^2(D)^{d_{in}+d_{out}}$ by

$$H = K_{L+1} \circ (K_L + b_L) \circ \sigma \cdots \circ (K_2 + b_2) \circ \sigma \circ (K_1 + b_1) \circ (K_0 + b_0),$$

where $K_\ell$ and $b_\ell$ are defined as follows. First, we choose $K_{inj} \in \mathcal{L}(L^2(D)^{d_{in}}, L^2(D)^{d_{in}})$ as a linear injective integral operator [2].

(i) When $\sigma_1 \in A_0^L \cap \text{BA}$ is injective,

$$K_0 = \begin{pmatrix} K_{inj} \\ \widetilde{K}_0 \end{pmatrix} \in \mathcal{L}(L^2(D)^{d_{in}}, L^2(D)^{d_{in}+d_1}), \quad b_0 = \begin{pmatrix} O \\ \widetilde{b}_0 \end{pmatrix} \in L^2(D)^{d_{in}+d_1},$$

$$\vdots$$

---

[2] For example, if we choose the integral kernel $k_{inj}$ as $k_{inj}(x, y) = \sum_{k=1}^\infty \vec{\varphi}_k(x) \vec{\varphi}_k(y)$, then the integral operator $K_{\text{inj}}$ with the kernel $k_{\text{inj}}$ is injective where $\{\vec{\varphi}\}_k$ is the orthonormal basis in $L^2(D)^{d_{in}}$.

$$K_\ell = \begin{pmatrix} K_{inj} & O \\ O & \widetilde{K}_\ell \end{pmatrix} \in \mathcal{L}(L^2(D)^{d_{in}+d_\ell}, L^2(D)^{d_{in}+d_{\ell+1}}), \quad b_\ell = \begin{pmatrix} O \\ \widetilde{b}_\ell \end{pmatrix} \in L^2(D)^{d_{in}+d_{\ell+1}},$$

$$(1 \le \ell \le L),$$

$$\vdots$$

$$K_{L+1} = \begin{pmatrix} K_{inj} & O \\ O & \widetilde{K}_{L+1} \end{pmatrix} \in \mathcal{L}(L^2(D)^{d_{in}+d_{L+1}}, L^2(D)^{d_{in}+d_{out}}), \quad b_\ell = \begin{pmatrix} O \\ O \end{pmatrix} \in L^2(D)^{d_{in}+d_{out}}.$$

(ii) When $\sigma_1 = \mathrm{ReLU}$,

$$K_0 = \begin{pmatrix} K_{inj} \\ \widetilde{K}_0 \end{pmatrix} \in \mathcal{L}(L^2(D)^{d_{in}}, L^2(D)^{d_{in}+d_1}), \quad b_0 = \begin{pmatrix} O \\ \widetilde{b}_0 \end{pmatrix} \in L^2(D)^{d_{in}+d_1},$$

$$K_1 = \begin{pmatrix} K_{inj} & O \\ -K_{inj} & O \\ O & \widetilde{K}_1 \end{pmatrix} \in \mathcal{L}(L^2(D)^{d_{in}+d_1}, L^2(D)^{2d_{in}+d_2}), \quad b_0 = \begin{pmatrix} O \\ O \\ \widetilde{b}_1 \end{pmatrix} \in L^2(D)^{2d_{in}+d_1},$$

$$\vdots$$

$$K_\ell = \begin{pmatrix} K_{inj} & -K_{inj} & O \\ -K_{inj} & K_{inj} & O \\ O & & \widetilde{K}_\ell \end{pmatrix} \in \mathcal{L}(L^2(D)^{2d_{in}+d_\ell}, L^2(D)^{2d_{in}+d_{\ell+1}}),$$

$$b_\ell = \begin{pmatrix} O \\ O \\ \widetilde{b}_\ell \end{pmatrix} \in L^2(D)^{2d_{in}+d_{\ell+1}}, \quad (2 \le \ell \le L),$$

$$\vdots$$

$$K_L = \begin{pmatrix} K_{inj} & -K_{inj} & O \\ O & & \widetilde{K}_L \end{pmatrix} \in \mathcal{L}(L^2(D)^{2d_{in}+d_L}, L^2(D)^{d_{in}+d_{L+1}}),$$

$$b_L = \begin{pmatrix} O \\ \widetilde{b}_L \end{pmatrix} \in L^2(D)^{d_{in}+d_{L+1}},$$

$$K_{L+1} = \begin{pmatrix} K_{inj} & O \\ O & \widetilde{K}_{L+1} \end{pmatrix} \in \mathcal{L}(L^2(D)^{d_{in}+d_{L+1}}, L^2(D)^{d_{in}+d_{out}}),$$

$$b_{L+1} = \begin{pmatrix} O \\ O \end{pmatrix} \in L^2(D)^{d_{in}+d_{out}}.$$

Then, the operator $H : L^2(D)^{d_{in}} \to L^2(D)^{d_{in}+d_{out}}$ has the form of

$$H := \begin{cases} \begin{pmatrix} K_{inj} \circ K_{inj} \circ \sigma \circ K_{inj} \circ \cdots \circ \sigma \circ K_{inj} \circ K_{inj} \\ \widetilde{G} \end{pmatrix} & \text{in the case of (i).} \\ \begin{pmatrix} K_{inj} \circ \cdots \circ K_{inj} \\ \widetilde{G} \end{pmatrix} & \text{in the case of (ii).} \end{cases}$$

For the case of (ii), we have used the fact

$$\begin{pmatrix} I & -I \end{pmatrix} \circ \mathrm{ReLU} \circ \begin{pmatrix} I \\ -I \end{pmatrix} = I.$$

Thus, in both cases, $H$ is injective.

In the case of (i), as $\sigma \in A_0^L$, we obtain the estimate

$$\|\sigma(f)\|_{L^2(D)^{d_{in}}} \le \sqrt{2|D|d_{in}}C_0 + \|f\|_{L^2(D)^{d_{in}}}, \quad f \in L^2(D)^{d_{in}},$$

where

$$C_0 := \sup_{x \in \mathbb{R}} \frac{|\sigma(x)|}{1+|x|} < \infty.$$

Then we evaluate for $a \in K (\subset B_R(0))$,

$$
\|H(a)\|_{L^2(D)^{d_{in}+d_{out}}}
$$

$$
\leq \left\|\widetilde{G}(a)\right\|_{L^2(D)^{d_{out}}} + \|K_{inj} \circ K_{inj} \circ \sigma \circ K_{inj} \circ \cdots \circ \sigma \circ K_{inj} \circ K_{inj}(a)\|_{L^2(D)^{d_{in}}} \tag{C.7}
$$

$$
\leq 4M + \sqrt{2|D|d_{in}}C_0 \sum_{\ell=1}^{L} \|K_{inj}\|_{\mathrm{op}}^{\ell+1} + \|K_{inj}\|_{\mathrm{op}}^{L+2} R =: C_H.
$$

In the case of (ii), we find the estimate, for $a \in K$,

$$
\|H(a)\|_{L^2(D)^{d_{in}+d_{out}}} \leq 4M + \|K_{inj}\|_{\mathrm{op}}^{L+2} R < C_H. \tag{C.8}
$$

From (C.6) (especially, $\mathrm{Ran}(\pi_1 H) \subset C(D)$) and Remark 3, we can choose an orthogonal sequence $\{\xi_k\}_{k \in \mathbb{N}}$ in $L^2(D)$ such that (3.1) holds. By applying Lemma 1, as $T = H, n = d_{in}, m = d_{in}+d_{out}$, $\ell = d_{out}$, we find that

$$
G := \underbrace{\pi_{d_{out}} \circ Q_\alpha \circ P_{V_\alpha^\perp}}_{=:B} \circ H : L^2(D)^{d_{in}} \to L^2(D)^{d_{out}},
$$

is injective. Here, $P_{V_\alpha^\perp}$ and $Q_\alpha$ are defined as in Remark 4; we choose $0 < \alpha << 1$ such that

$$
\left\|P_{V_\alpha^\perp} - P_{V_0^\perp}\right\|_{\mathrm{op}} < \min\left(\frac{\epsilon}{10C_H}, 1\right) =: \epsilon_0,
$$

where $P_{V_0^\perp}$ is the orthogonal projection onto

$$
V_0^\perp := \{0\}^{d_{in}} \times L^2(D)^{d_{out}}.
$$

By the same argument as in the proof of Theorem 15 in Puthawala et al. [2022a], we can show that

$$
\|I - Q_\alpha\|_{\mathrm{op}} \leq 4\epsilon_0.
$$

Furthermore, since $B$ is a linear operator, $B \circ K_{L+1}$ is also a linear operator with integral kernel $(Bk_{L+1}(\cdot, y))(x)$, where $k_{L+1}(x, y)$ is the kernel of $K_{L+1}$. This implies that

$$
G \in \mathrm{NO}_L(\sigma; D, d_{in}, d_{out}).
$$

We get, for $a \in K$,

$$
\left\|G^+(a) - G(a)\right\|_{L^2(D)^{d_{out}}} \leq \underbrace{\left\|G^+(a) - \widetilde{G}(a)\right\|_{L^2(D)^{d_{out}}}}_{(C.5) \leq \frac{\epsilon}{2}} + \left\|\widetilde{G}(a) - G(a)\right\|_{L^2(D)^{d_{out}}}. \tag{C.9}
$$

Using (C.7) and (C.8), we then obtain

$$
\left\|\widetilde{G}(a) - G(a)\right\|_{L^2(D)^{d_{out}}} = \left\|\pi_{d_{out}} \circ H(a) - \pi_{d_{out}} \circ Q_\alpha \circ P_{V_\alpha^\perp} \circ H(a)\right\|_{L^2(D)^{d_{out}}}
$$

$$
\leq \left\|\pi_{d_{out}} \circ (P_{V_0^\perp} - P_{V_\alpha^\perp} + P_{V_\alpha^\perp}) \circ H(a) - \pi_{d_{out}} \circ Q_\alpha \circ P_{V_\alpha^\perp} \circ H(a)\right\|_{L^2(D)^{d_{out}}}
$$

$$
\leq \left\|\pi_{d_{out}} \circ (P_{V_0^\perp} - P_{V_\alpha^\perp}) \circ H(a)\right\|_{L^2(D)^{d_{out}}} + \left\|\pi_{d_{out}} \circ (I - Q_\alpha) \circ P_{V_\alpha^\perp} \circ H(a)\right\|_{L^2(D)^{d_{out}}}
$$

$$
\leq 5\epsilon_0 \|H(a)\|_{L^2(D)^{d_{in}+d_{out}}} \leq \frac{\epsilon}{2}. \tag{C.10}
$$

Combining (C.9) and (C.10), we conclude that

$$
\sup_{a \in K} \left\|G^+(a) - G(a)\right\|_{L^2(D)^{d_{out}}} \leq \frac{\epsilon}{2} + \frac{\epsilon}{2} = \epsilon.
$$

$\square$

## C.4 Remark following Theorem 1

**Remark 5.** *We make the following observations using Theorem 1:*

(i) *ReLU and Leaky ReLU functions belong to $A_0^L \cap BA$ due to the fact that $\{\sigma \in C(\mathbb{R}) \mid \sigma$ is not a polynomial$\} \subseteq A_0$ (see Pinkus [1999]), and both the ReLU and Leaky ReLU functions belong to $BA$ (see Lemma C.2 in Lanthaler et al. [2022]). We note that Lemma C.2 in Lanthaler et al. [2022] solely established the case for ReLU. However, it holds true for Leaky ReLU as well since the proof relies on the fact that the function $x \mapsto \min(\max(x, R), R)$ can be exactly represented by a two-layer ReLU neural network, and a two-layer Leaky ReLU neural network can also represent this function. Consequently, Leaky ReLU is one of example that satisfies (ii) in Theorem 1.*

(ii) *We emphasize that our infinite-dimensional result, Theorem 1, deviates from the finite-dimensional result. Puthawala et al. [2022a, Theorem 15] assumes that $2d_{in} + 1 \leq d_{out}$ due to the use of Whitney's theorem. In contrast, Theorem 1 does not assume any conditions on $d_{in}$ and $d_{out}$, that is, we are able to avoid invoking Whitney's theorem by employing Lemma 1.*

(iii) *We provide examples that injective universality does not hold when $L^2(D)^{d_{in}}$ and $L^2(D)^{d_{out}}$ are replaced by $\mathbb{R}^{d_{in}}$ and $\mathbb{R}^{d_{out}}$: Consider the case where $d_{in} = d_{out} = 1$ and $G^+ : \mathbb{R} \to \mathbb{R}$ is defined as $G^+(x) = \sin(x)$. We can not approximate $G^+ : \mathbb{R} \to \mathbb{R}$ by an injective function $G : \mathbb{R} \to \mathbb{R}$ in the set $K = [0, 2\pi]$ in the $L^\infty$-norm. According to the topological degree theory (see Cho and Chen [2006, Theorem 1.2.6(iii)]), any continuous function $G : \mathbb{R} \to \mathbb{R}$ which satisfies $\|G - G^+\|_{C([0,2\pi])} < \varepsilon$ satisfies the equation on both intervals $I_1 = [0, \pi]$, $I_2 = [\pi, 2\pi]$ $\deg(G, I_j, s) = \deg(G^+, I_j, s) = 1$ for all $s \in [-1 + \varepsilon, 1 - \varepsilon]$, $j = 1, 2$. This implies that $G : I_j \to \mathbb{R}$ obtains the value $s \in [-1 + \varepsilon, 1 - \varepsilon]$ at least once. Hence, $G$ obtains the values $s \in [-1 + \varepsilon, 1 - \varepsilon]$ at least two times on the interval $[0, 2\pi]$ and is it thus not injective. It is worth noting that the degree theory exhibits significant differences between the infinite-dimensional and finite-dimensional cases [Cho and Chen, 2006].*

# D    Details in Section 3.3

## D.1    Finite rank approximation

We consider linear integral operators $K_\ell$ with $L^2$ kernels $k_\ell(x, y)$. Let $\{\varphi_k\}_{k \in \mathbb{N}}$ be an orthonormal basis in $L^2(D)$. Since $\{\varphi_k(y)\varphi_p(x)\}_{k,p \in \mathbb{N}}$ is an orthonormal basis of $L^2(D \times D)$, integral kernels $k_\ell \in L^2(D \times D; \mathbb{R}^{d_{\ell+1} \times d_\ell})$ in integral operators $K_\ell \in \mathcal{L}(L^2(D)^{d_\ell}, L^2(D)^{d_{\ell+1}})$ has the expansion

$$k_\ell(x, y) = \sum_{k,p \in \mathbb{N}} C_{k,p}^{(\ell)} \varphi_k(y)\varphi_p(x),$$

then integral operators $K_\ell \in \mathcal{L}(L^2(D)^{d_\ell}, L^2(D)^{d_{\ell+1}})$ take the form

$$K_\ell u(x) = \sum_{k,p \in \mathbb{N}} C_{k,p}^{(\ell)}(u, \varphi_k)\varphi_p(x), \ u \in L^2(D)^{d_\ell},$$

where $C_{k,p}^{(\ell)} \in \mathbb{R}^{d_{\ell+1} \times d_\ell}$ whose $(i, j)$-th component $c_{k,p,ij}^{(\ell)}$ is given by

$$c_{k,p,ij}^{(\ell)} = (k_{\ell,ij}, \varphi_k \varphi_p)_{L^2(D \times D)}.$$

Here, we write $(u, \varphi_k) \in \mathbb{R}^{d_\ell}$,

$$(u, \varphi_k) = \left((u_1, \varphi_k)_{L^2(D)}, ..., (u_{d_\ell}, \varphi_k)_{L^2(D)}\right).$$

We define $K_{\ell,N} \in \mathcal{L}(L^2(D)^{d_\ell}, L^2(D)^{d_{\ell+1}})$ as the truncated expansion of $K_\ell$ by $N$ finite sum, that is,

$$K_{\ell,N} u(x) := \sum_{k,p \leq N} C_{k,p}^{(\ell)}(u, \varphi_k)\varphi_p(x).$$

Then $K_{\ell,N} \in \mathcal{L}(L^2(D)^{d_\ell}, L^2(D)^{d_{\ell+1}})$ is a finite rank operator with rank $N$. Furthermore, we have

$$\|K_\ell - K_{\ell,N}\|_{\mathrm{op}} \leq \|K_\ell - K_{\ell,N}\|_{\mathrm{HS}} = \left( \sum_{k,p \geq N} \sum_{i,j} |c_{k,p,ij}^{(\ell)}|^2 \right)^{1/2}, \tag{D.1}$$

which implies that as $N \to \infty$,

$$\|K_\ell - K_{\ell,N}\|_{\mathrm{op}} \to 0.$$

## D.2 Layerwise injectivity

We first revisit layerwise injectivity and bijectivity in the case of the finite rank approximation. Let $K_N : L^2(D)^n \to L^2(D)^m$ be a finite rank operator defined by

$$K_N u(x) := \sum_{k,p \leq N} C_{k,p}(u, \varphi_k) \varphi_p(x), \ u \in L^2(D)^n,$$

where $C_{k,p} \in \mathbb{R}^{m \times n}$ and $(u, \varphi_p) \in \mathbb{R}^n$ is given by

$$(u, \varphi_p) = \left( (u_1, \varphi_p)_{L^2(D)}, ..., (u_n, \varphi_p)_{L^2(D)} \right).$$

Let $b_N \in L^2(D)^n$ be defined by

$$b_N(x) := \sum_{p \leq N} b_p \varphi_p(x),$$

in which $b_p \in \mathbb{R}^m$. As analogues of Propositions 1 and 2, we obtain the following characterization.

**Proposition 5.** *(i) The operator*

$$\mathrm{ReLU} \circ (K_N + b_N) : (\mathrm{span}\{\varphi_k\}_{k \leq N})^n \to L^2(D)^m,$$

*is injective if and only if for every* $v \in (\mathrm{span}\{\varphi_k\}_{k \leq N})^n$,

$$\{u \in L^2(D)^n \mid \vec{u}_N \in \mathrm{Ker}(C_{S,N})\} \cap X(v, K_N + b_N) \cap (\mathrm{span}\{\varphi_k\}_{k \leq N})^n = \{0\}.$$

*where* $S(v, K_N + b_N) \subset [m]$ *and* $X(v, K_N + b_N)$ *are defined in Definition 2, and*

$$\vec{u}_N := ((u, \varphi_p))_{p \leq N} \in \mathbb{R}^{Nn}, \ C_{S,N} := \left( C_{k,q}\big|_{S(v, K_N + b_N)} \right)_{k,q \in [N]} \in \mathbb{R}^{N|S(v, K_N + b_N)| \times Nn}. \tag{D.2}$$

*(ii) Let* $\sigma$ *be injective. Then the operator*

$$\sigma \circ (K_N + b_N) : (\mathrm{span}\{\varphi_k\}_{k \leq N})^n \to L^2(D)^m,$$

*is injective if and only if* $C_N$ *is injective, where*

$$C_N := (C_{k,q})_{k,q \in [N]} \in \mathbb{R}^{Nm \times Nn}. \tag{D.3}$$

*Proof.* The above statements follow from Propositions 1 and 2 by observing that $u \in \mathrm{Ker}(K_N)$ is equivalent to (cf. (D.2) and (D.3))

$$\sum_{k,p \leq N} C_{k,p}(u, \varphi_k) \varphi_p = 0, \iff C_N \vec{u}_N = 0.$$

$\square$

## D.3 Global injectivity

We revisit global injectivity in the case of finite rank approximation. As an analogue of Lemma 1, we have the following

**Lemma 2.** *Let $N, N' \in \mathbb{N}$ and $n, m, \ell \in \mathbb{N}$ with $N'm > N'\ell \geq 2Nn + 1$, and let $T : L^2(D)^n \to L^2(D)^m$ be a finite rank operator with $N'$ rank, that is,*

$$\text{Ran}(T) \subset (\text{span}\{\varphi_k\}_{k \leq N'})^m, \tag{D.4}$$

*and Lipschitz continuous, and*

$$T : (\text{span}\{\varphi_k\}_{k \leq N})^n \to L^2(D)^m,$$

*is injective. Then, there exists a finite rank operator $B \in \mathcal{L}(L^2(D)^m, L^2(D)^\ell)$ with rank $N'$ such that*

$$B \circ T : (\text{span}\{\varphi_k\}_{k \leq N})^n \to (\text{span}\{\varphi_k\}_{k \leq N'})^\ell,$$

*is injective.*

*Proof.* From (D.4), $T : L^2(D)^n \to L^2(D)^m$ has the form of

$$T(a) = \sum_{k \leq N'} (T(a), \varphi_k)\varphi_k,$$

where $(T(a), \varphi_k) \in \mathbb{R}^m$. We define $\mathbf{T} : \mathbb{R}^{Nn} \to \mathbb{R}^{N'm}$ by

$$\mathbf{T}(\mathbf{a}) := ((T(\mathbf{a}), \varphi_k))_{k \in [N']} \in \mathbb{R}^{N'm}, \ \mathbf{a} \in \mathbb{R}^{Nn},$$

where $T(\mathbf{a}) \in L^2(D)^m$ is defined by

$$T(\mathbf{a}) := T\left(\sum_{k \leq N} a_k \varphi_k\right) \in L^2(D)^m,$$

in which $a_k \in \mathbb{R}^n$, $\mathbf{a} = (a_1, ..., a_N) \in \mathbb{R}^{Nn}$.

Since $T : L^2(D)^n \to L^2(D)^m$ is Lipschitz continuous, $\mathbf{T} : \mathbb{R}^{Nn} \to \mathbb{R}^{N'm}$ is also Lipschitz continuous. As $N'm > N'\ell \geq 2Nn + 1$, we can apply Lemma 29 from Puthawala et al. [2022a] with $D = N'm$, $m = N'\ell$, $n = Nn$. According to this lemma, there exists a $N'\ell$-dimensional linear subspace $\mathbf{V}^\perp$ in $\mathbb{R}^{N'm}$ such that

$$\left\|P_{\mathbf{V}^\perp} - P_{\mathbf{V}_0^\perp}\right\|_{\text{op}} < 1,$$

and

$$P_{\mathbf{V}^\perp} \circ \mathbf{T} : \mathbb{R}^{Nn} \to \mathbb{R}^{N'm},$$

is injective, where $\mathbf{V}_0^\perp = \{0\}^{N'(m-\ell)} \times \mathbb{R}^{N'\ell}$. Furthermore, in the proof of Theorem 15 of Puthawala et al. [2022a], denoting

$$\mathbf{B} := \pi_{N'\ell} \circ \mathbf{Q} \circ P_{\mathbf{V}^\perp} \in \mathbb{R}^{N'\ell \times N'm},$$

we are able to show that

$$\mathbf{B} \circ \mathbf{T} : \mathbb{R}^{Nn} \to \mathbb{R}^{N'\ell}$$

is injective. Here, $\pi_{N'\ell} : \mathbb{R}^{N'm} \to \mathbb{R}^{N'\ell}$

$$\pi_{N'\ell}(a, b) := b, \quad (a, b) \in \mathbb{R}^{N'(m-\ell)} \times \mathbb{R}^{N'\ell},$$

where $\mathbf{Q} : \mathbb{R}^{N'm} \to \mathbb{R}^{N'm}$ is defined by

$$\mathbf{Q} := \left(P_{\mathbf{V}_0^\perp}P_{\mathbf{V}^\perp} + (I - P_{\mathbf{V}_0^\perp})(I - P_{\mathbf{V}^\perp})\right)\left(I - (P_{\mathbf{V}_0^\perp} - P_{\mathbf{V}^\perp})^2\right)^{-1/2}.$$

We define $B : L^2(D)^m \to L^2(D)^\ell$ by

$$Bu = \sum_{k,p \leq N'} \mathbf{B}_{k,p}(u, \varphi_k)\varphi_p,$$

where $\mathbf{B}_{k,p} \in \mathbb{R}^{\ell \times m}$, $\mathbf{B} = (\mathbf{B}_{k,p})_{k,p \in [N']}$. Then $B : L^2(D)^m \to L^2(D)^\ell$ is a linear finite rank operator with $N'$ rank, and

$$B \circ T : L^2(D)^n \to L^2(D)^\ell$$

is injective because, by the construction, it is equivalent to

$$\mathbf{B} \circ \mathbf{T} : \mathbb{R}^{Nn} \to \mathbb{R}^{N'\ell},$$

is injective. $\qquad\square$

## D.4 Proof of Theorem 2

**Definition 4.** *We define the set of integral neural operators with $N$ rank by*

$$\mathrm{NO}_{L,N}(\sigma; D, d_{in}, d_{out}) := \Big\{ G_N : L^2(D)^{d_{in}} \to L^2(D)^{d_{out}} :$$

$$G_N = K_{L+1,N} \circ (K_{L,N} + b_{L,N}) \circ \sigma \cdots \circ (K_{2,N} + b_{2,N}) \circ \sigma \circ (K_{1,N} + b_{1,N}) \circ (K_{0,N} + b_{0,N}),$$

$$K_{\ell,N} \in \mathcal{L}(L^2(D)^{d_\ell}, L^2(D)^{d_{\ell+1}}), \ K_{\ell,N} : f \mapsto \sum_{k,p \leq N} C_{k,p}^{(\ell)}(f, \varphi_k)\varphi_p,$$

$$b_{\ell,N} \in L^2(D; \mathbb{R}^{d_{\ell+1}}), \ b_{\ell,N} = \sum_{p \leq N} b_p^{(\ell)}\varphi_m,$$

$$C_{k,p}^{(\ell)} \in \mathbb{R}^{d_{\ell+1} \times d_\ell}, \ b_p^{(\ell)} \in \mathbb{R}^{d_{\ell+1}}, \ k, p \leq N,$$

$$d_\ell \in \mathbb{N}, \ d_0 = d_{in}, \ d_{L+2} = d_{out}, \ \ell = 0, ..., L+2 \Big\}. \tag{D.5}$$

*Proof.* Let $R > 0$ such that
$$K \subsetneq B_R(0),$$
where $B_R(0) := \{u \in L^2(D)^{d_{in}} \mid \|u\|_{L^2(D)^{d_{in}}} \leq R\}$. As ReLU and Leaky ReLU function belongs to $\mathrm{A}_0^L \cap \mathrm{BA}$, by Theorem 11 of Kovachki et al. [2021b], there exists $L \in \mathbb{N}$ and $\widetilde{G} \in \mathrm{NO}_L(\sigma; D, d_{in}, d_{out})$ such that

$$\sup_{a \in K} \left\| G^+(a) - \widetilde{G}(a) \right\|_{L^2(D)^{d_{out}}} \leq \frac{\epsilon}{3}. \tag{D.6}$$

and

$$\left\| \widetilde{G}(a) \right\|_{L^2(D)^{d_{out}}} \leq 4M, \quad \text{for } a \in L^2(D)^{d_{in}}, \quad \|a\|_{L^2(D)^{d_{in}}} \leq R.$$

We write operator $\widetilde{G}$ by

$$\widetilde{G} = \widetilde{K}_{L+1} \circ (\widetilde{K}_L + \widetilde{b}_L) \circ \sigma \cdots \circ (\widetilde{K}_2 + \widetilde{b}_2) \circ \sigma \circ (\widetilde{K}_1 + \widetilde{b}_1) \circ (\widetilde{K}_0 + \widetilde{b}_0),$$

where

$$\widetilde{K}_\ell \in \mathcal{L}(L^2(D)^{d_\ell}, L^2(D)^{d_{\ell+1}}), \ \widetilde{K}_\ell : f \mapsto \int_D \widetilde{k}_\ell(\cdot, y) f(y) dy,$$

$$\widetilde{k}_\ell \in L^2(D \times D; \mathbb{R}^{d_{\ell+1} \times d_\ell}), \ \widetilde{b}_\ell \in L^2(D; \mathbb{R}^{d_{\ell+1}}),$$

$$d_\ell \in \mathbb{N}, \ d_0 = d_{in}, \ d_{L+2} = d_{out}, \ \ell = 0, ..., L+2.$$

We set $\widetilde{G}_{N'} \in \mathrm{NO}_{L,N'}(\sigma; D, d_{in}, d_{out})$ such that

$$\widetilde{G}_{N'} = \widetilde{K}_{L+1,N'} \circ (\widetilde{K}_{L,N'} + \widetilde{b}_{L,N'}) \circ \sigma \cdots \circ (\widetilde{K}_{2,N'} + \widetilde{b}_{2,N'}) \circ \sigma \circ (\widetilde{K}_{1,N'} + \widetilde{b}_{1,N'}) \circ (\widetilde{K}_{0,N'} + \widetilde{b}_{0,N'}),$$

where $\widetilde{K}_{\ell,N'} : L^2(D)^{d_\ell} \to L^2(D)^{d_{\ell+1}}$ is defined by

$$\widetilde{K}_{\ell,N'}u(x) = \sum_{k,p \leq N'} C_{k,p}^{(\ell)}(u, \varphi_k)\varphi_p(x),$$

where $C_{k,p}^{(\ell)} \in \mathbb{R}^{d_{\ell+1} \times d_\ell}$ whose $(i,j)$-th component $c_{k,p,ij}^{(\ell)}$ is given by

$$c_{k,p,ij}^{(\ell)} = (\widetilde{k}_{\ell,ij}, \varphi_k\varphi_p)_{L^2(D \times D)}.$$

Since

$$\left\| \widetilde{K}_\ell - \widetilde{K}_{\ell,N'} \right\|_{\mathrm{op}}^2 \leq \left\| \widetilde{K}_\ell - \widetilde{K}_{\ell,N'} \right\|_{\mathrm{HS}}^2 = \sum_{k,p \geq N'+1} \sum_{i,j} |c_{k,p,ij}^{(\ell)}|^2 \to 0 \text{ as } N' \to \infty,$$

there is a large $N' \in \mathbb{N}$ such that

$$\sup_{a \in K} \left\| \widetilde{G}(a) - \widetilde{G}_{N'}(a) \right\|_{L^2(D)^{d_{out}}} \leq \frac{\epsilon}{3}. \tag{D.7}$$

Then, we have

$$\sup_{a \in K} \left\| \widetilde{G}_{N'}(a) \right\|_{L^2(D)^{d_{out}}} \leq \sup_{a \in K} \left\| \widetilde{G}_{N'}(a) - \widetilde{G}(a) \right\|_{L^2(D)^{d_{out}}} + \sup_{a \in K} \left\| \widetilde{G}(a) \right\|_{L^2(D)^{d_{out}}}$$
$$\leq 1 + 4M.$$

We define the operator $H_{N'} : L^2(D)^{d_{in}} \to L^2(D)^{d_{in}+d_{out}}$ by

$$H_{N'}(a) = \begin{pmatrix} H_{N'}(a)_1 \\ H_{N'}(a)_2 \end{pmatrix} := \begin{pmatrix} K_{inj,N} \circ \cdots \circ K_{inj,N}(a) \\ \widetilde{G}_{N'}(a) \end{pmatrix},$$

where $K_{inj,N} : L^2(D)^{d_{in}} \to L^2(D)^{d_{in}}$ is defined by

$$K_{inj,N} u = \sum_{k \leq N} (u, \varphi_k) \varphi_k.$$

As $K_{inj,N} : (\mathrm{span}\{\varphi_k\}_{k \leq N})^{d_{in}} \to L^2(D)^{d_{in}}$ is injective,

$$H_{N'} : (\mathrm{span}\{\varphi_k\}_{k \leq N})^{d_{in}} \to (\mathrm{span}\{\varphi_k\}_{k \leq N})^{d_{in}} \times (\mathrm{span}\{\varphi_k\}_{k \leq N'})^{d_{out}},$$

is injective. Furthermore, by the same argument (ii) (construction of $H$) in the proof of Theorem 1,

$$H_{N'} \in NO_{L,N'}(\sigma; D, d_{in}, d_{out}),$$

because both of two-layer ReLU and Leaky ReLU neural networks can represent the identity map. Note that above $K_{inj,N}$ is an orthogonal projection, so that $K_{inj,N} \circ \cdots \circ K_{inj,N} = K_{inj,N}$. However, we write above $H_{N'}(a)_1$ as $K_{inj,N} \circ \cdots \circ K_{inj,N}(a)$ so that it can be considered as combination of $(L+2)$ layers of neural networks.

We estimate that for $a \in L^2(D)^{d_{in}}$, $\|a\|_{L^2(D)^{d_{in}}} \leq R$,

$$\|H_{N'}(a)\|_{L^2(D)^{d_{in}+d_{out}}} \leq 1 + 4M + \|K_{inj}\|_{op}^{L+2} R =: C_H.$$

Here, we repeat an argument similar to the one in the proof of Lemma 2: $H_{N'} : L^2(D)^{d_{in}} \to L^2(D)^{d_{in}+d_{out}}$ has the form of

$$H_{N'}(a) = \left( \sum_{k \leq N} (H_{N'}(a)_1, \varphi_k) \varphi_k, \sum_{k \leq N'} (H_{N'}(a)_2, \varphi_k) \varphi_k \right).$$

where $(H_{N'}(a)_1, \varphi_k) \in \mathbb{R}^{d_{in}}$, $(H_{N'}(a)_2, \varphi_k) \in \mathbb{R}^{d_{out}}$. We define $\mathbf{H}_{N'} : \mathbb{R}^{Nd_{in}} \to \mathbb{R}^{Nd_{in}+N'd_{out}}$ by

$$\mathbf{H}_{N'}(\mathbf{a}) := \left[ \left( (H_{N'}(\mathbf{a})_1, \varphi_k) \right)_{k \in [N]}, \left( (H_{N'}(\mathbf{a})_2, \varphi_k) \right)_{k \in [N']} \right] \in \mathbb{R}^{Nd_{in}+N'd_{out}}, \quad \mathbf{a} \in \mathbb{R}^{Nd_{in}},$$

where $H_{N'}(\mathbf{a}) = (H_{N'}(\mathbf{a})_1, H_{N'}(\mathbf{a})_2) \in L^2(D)^{d_{in}+d_{out}}$ is defined by

$$H_{N'}(\mathbf{a})_1 := H_{N'} \left( \sum_{k \leq N} a_k \varphi_k \right)_1 \in L^2(D)^{d_{in}},$$

$$H_{N'}(\mathbf{a})_2 := H_{N'} \left( \sum_{k \leq N'} a_k \varphi_k \right)_2 \in L^2(D)^{d_{out}},$$

where $a_k \in \mathbb{R}^{d_{in}}, \mathbf{a} = (a_1, ..., a_N) \in \mathbb{R}^{Nd_{in}}$. Since $H_{N'} : L^2(D)^{d_{in}} \to L^2(D)^{d_{in}+d_{out}}$ is Lipschitz continuous, $\mathbf{H}_{N'} : \mathbb{R}^{Nd_{in}} \to \mathbb{R}^{N'd_{out}}$ is also Lipschitz continuous. As

$$Nd_{in} + N'd_{out} > N'd_{out} \geq 2Nd_{in} + 1,$$

we can apply Lemma 29 of Puthawala et al. [2022a] with $D = Nd_{in} + N'd_{out}$, $m = N'd_{out}$, $n = Nd_{in}$. According to this lemma, there exists a $N'd_{out}$-dimensional linear subspace $\mathbf{V}^\perp$ in $\mathbb{R}^{Nd_{in}+N'd_{out}}$ such that

$$\left\| P_{\mathbf{V}^\perp} - P_{\mathbf{V}_0^\perp} \right\|_{op} < \min \left( \frac{\epsilon}{15 C_{H_N}}, 1 \right) =: \epsilon_0$$

and
$$P_{\mathbf{V}^\perp} \circ \mathbf{H}_{N'} : \mathbb{R}^{Nd_{in}} \to \mathbb{R}^{Nd_{in}+N'd_{out}},$$
is injective, where $\mathbf{V}_0^\perp = \{0\}^{Nd_{in}} \times \mathbb{R}^{N'd_{out}}$. Furthermore, in the proof of Theorem 15 of Puthawala et al. [2022a], denoting by
$$\mathbf{B} := \pi_{N'd_{out}} \circ \mathbf{Q} \circ P_{\mathbf{V}^\perp},$$
we can show that
$$\mathbf{B} \circ \mathbf{H}_{N'} : \mathbb{R}^{Nd_{in}} \to \mathbb{R}^{N'd_{out}},$$
is injective, where $\pi_{N'd_{out}} : \mathbb{R}^{Nd_{in}+N'd_{out}} \to \mathbb{R}^{N'd_{out}}$
$$\pi_{N'd_{out}}(a,b) := b, \quad (a,b) \in \mathbb{R}^{Nd_{in}} \times \mathbb{R}^{N'd_{out}},$$
and $\mathbf{Q} : \mathbb{R}^{Nd_{in}+N'd_{out}} \to \mathbb{R}^{Nd_{in}+N'd_{out}}$ is defined by
$$\mathbf{Q} := \left( P_{\mathbf{V}_0^\perp} P_{\mathbf{V}^\perp} + (I - P_{\mathbf{V}_0^\perp})(I - P_{\mathbf{V}^\perp}) \right) \left( I - (P_{\mathbf{V}_0^\perp} - P_{\mathbf{V}^\perp})^2 \right)^{-1/2}.$$
By the same argument in proof of Theorem 15 in Puthawala et al. [2022a], we can show that
$$\|I - \mathbf{Q}\|_{\mathrm{op}} \le 4\epsilon_0.$$
We define $B : L^2(D)^{d_{in}+d_{out}} \to L^2(D)^{d_{out}}$
$$Bu = \sum_{k,p \le N'} \mathbf{B}_{k,p}(u, \varphi_k)\varphi_p,$$
$\mathbf{B}_{k,p} \in \mathbb{R}^{d_{out} \times (d_{in}+d_{out})}$, $\mathbf{B} = (\mathbf{B}_{k,p})_{k,p \in [N']}$, then $B : L^2(D)^{d_{in}+d_{out}} \to L^2(D)^{d_{out}}$ is a linear finite rank operator with $N'$ rank. Then,
$$G_{N'} := B \circ H_{N'} : L^2(D)^{d_{in}} \to L^2(D)^{d_{out}},$$
is injective because by the construction, it is equivalent to
$$\mathbf{B} \circ \mathbf{H}_{N'} : \mathbb{R}^{Nd_{in}} \to \mathbb{R}^{N'd_{out}},$$
is injective. Furthermore, we have
$$G_{N'} \in NO_{L,N'}(\sigma; D, d_{in}, d_{out}).$$
Indeed, $H_{N'} \in NO_{L,N'}(\sigma; D, d_{in}, d_{out})$, $B$ is the linear finite rank operator with $N'$ rank, and multiplication of two linear finite rank operators with $N'$ rank is also a linear finite rank operator with $N'$ rank.

Finally, we estimate for $a \in K$,
$$\left\| G^+(a) - G_{N'}(a) \right\|_{L^2(D)^{d_{out}}}$$
$$= \underbrace{\left\| G^+(a) - \widetilde{G}(a) \right\|_{L^2(D)^{d_{out}}}}_{(D.6) \le \frac{\epsilon}{3}} + \underbrace{\left\| \widetilde{G}(a) - \widetilde{G}_{N'}(a) \right\|_{L^2(D)^{d_{out}}}}_{(D.7) \le \frac{\epsilon}{3}} + \left\| \widetilde{G}_{N'}(a) - G_{N'}(a) \right\|_{L^2(D)^{d_{out}}}.$$
$$\tag{D.8}$$
Using notation $(a, \varphi_k) \in \mathbb{R}^{d_{in}}$, and $\mathbf{a} = ((a, \varphi_k))_{k \in [N]} \in \mathbb{R}^{Nd_{in}}$, we further estimate for $a \in K$,
$$\left\| \widetilde{G}_{N'}(a) - G_{N'}(a) \right\|_{L^2(Q)^{d_{out}}} = \| \pi_{d_{out}} H_{N'}(a) - B \circ H_{N'}(a) \|_{L^2(Q)^{d_{out}}}$$
$$= \| \pi_{N'd_{out}} \mathbf{H}_{N'}(\mathbf{a}) - \mathbf{B} \circ \mathbf{H}_{N'}(\mathbf{a}) \|_2$$
$$= \| \pi_{N'd_{out}} \circ \mathbf{H}_{N'}(\mathbf{a}) - \pi_{N'd_{out}} \circ \mathbf{Q} \circ P_{\mathbf{V}^\perp} \circ \mathbf{H}_{N'}(\mathbf{a}) \|_2$$
$$\le \left\| \pi_{N'd_{out}} \circ (P_{\mathbf{V}_0^\perp} - P_{\mathbf{V}^\perp} + P_{\mathbf{V}^\perp}) \circ \mathbf{H}_{N'}(\mathbf{a}) - \pi_{N'd_{out}} \circ \mathbf{Q} \circ P_{\mathbf{V}^\perp} \circ \mathbf{H}_{N'}(\mathbf{a}) \right\|_2 \tag{D.9}$$
$$\le \left\| \pi_{N'd_{out}} \circ (P_{\mathbf{V}_0^\perp} - P_{\mathbf{V}^\perp}) \circ \mathbf{H}_{N'}(\mathbf{a}) \right\|_2 + \| \pi_{N'd_{out}} \circ (I - \mathbf{Q}) \circ P_{\mathbf{V}^\perp} \circ \mathbf{H}_{N'}(\mathbf{a}) \|_2$$
$$\le 5\epsilon_0 \underbrace{\| \mathbf{H}_{N'}(\mathbf{a}) \|_2}_{= \|H_{N'}(a)\|_{L^2(D)^{d_{out}}} < C_H} \le \frac{\epsilon}{3},$$
where $\|\cdot\|_2$ is the Euclidean norm. Combining (D.8) and (D.9), we conclude that
$$\sup_{a \in K} \left\| G^+(a) - G_{N'}(a) \right\|_{L^2(D)^{d_{out}}} \le \frac{\epsilon}{3} + \frac{\epsilon}{3} + \frac{\epsilon}{3} = \epsilon.$$

$\square$

# E  Details in Section 4.1

## E.1  Proof of Proposition 3

*Proof.* Since $W$ is bijective, and $\sigma$ is surjective, it is enough to show that $u \mapsto Wu + K(u)$ is surjective. We observe that for $z \in L^2(D)^n$,

$$Wu + K(u) = z,$$

is equivalent to

$$H_z(u) := -W^{-1}K(u) + W^{-1}z = u.$$

We will show that $H_z : L^2(D)^n \to L^2(D)^n$ has a fixed point for each $z \in L^2(D)^n$. By the Leray-Schauder theorem, see Gilbarg and Trudinger [2001, Theorem 11.3], $H : L^2(D) \to L^2(D)$ has a fixed point if the union $\bigcup_{0 < \lambda \leq 1} V_\lambda$ is bounded, where the sets

$$
\begin{aligned}
V_\lambda &:= \{u \in L^2(D) : \ u = \lambda H_z(u)\} \\
&= \{u \in L^2(D) : \ \lambda^{-1}u = H_z(u)\} \\
&= \{u \in L^2(D) : \ -\lambda^{-1}u = W^{-1}K(u) - W^{-1}z\},
\end{aligned}
$$

are parametrized by $0 < \lambda \leq 1$.

As the map $u \mapsto \alpha u + W^{-1}K(u)$ is coercive, there is an $r > 0$ such that for $\|u\|_{L^2(D)^n} > r$,

$$\frac{\left\langle \alpha u + W^{-1}K(u), u \right\rangle_{L^2(D)^n}}{\|u\|_{L^2(D)^n}} \geq \|W^{-1}z\|_{L^2(D)^n}.$$

Thus, we have that for $\|u\|_{L^2(D)^n} > r$

$$
\begin{aligned}
&\frac{\left\langle W^{-1}K(u) - W^{-1}z, u \right\rangle_{L^2(D)^n}}{\|u\|^2_{L^2(D)^n}} \\
&\geq \frac{\left\langle \alpha u + W^{-1}K(u), u \right\rangle_{L^2(D)^n} - \left\langle \alpha u + W^{-1}z, u \right\rangle_{L^2(D)^n}}{\|u\|^2_{L^2(D)^n}} \\
&\geq \frac{\|W^{-1}z\|_{L^2(D)^n}}{\|u\|_{L^2(D)^n}} - \frac{\left\langle W^{-1}z, u \right\rangle_{L^2(D)^n}}{\|u\|^2_{L^2(D)^n}} - \alpha \geq -\alpha > -1,
\end{aligned}
$$

and, hence, for all $\|u\|_{L^2(D)} > r_0$ and $\lambda \in (0, 1]$ we have $u \notin V_\lambda$. Thus

$$\bigcup_{\lambda \in (0,1]} V_\lambda \subset B(0, r_0).$$

Again, by the Leray-Schauder theorem (see Gilbarg and Trudinger [2001, Theorem 11.3]), $H_z$ has a fixed point. $\qquad \square$

## E.2  Examples for Proposition 3

**Example 2.** *We consider the case where $n = 1$ and $D \subset \mathbb{R}^d$ is a bounded interval. We consider the non-linear integral operator,*

$$K(u)(x) := \int_D k(x, y, u(x))u(y)dy, \ x \in D,$$

*and $k(x, y, t)$ is bounded, that is, there is $C_K > 0$ such that*

$$|k(x, y, t)| \leq C_K, \ x, y \in D, \ t \in \mathbb{R}.$$

*If $\left\|W^{-1}\right\|_{\mathrm{op}}$ is small enough such that*

$$1 > \left\|W^{-1}\right\|_{\mathrm{op}} C_K |D|,$$

*then, for* $\alpha \in \left( \left\| W^{-1} \right\|_{\mathrm{op}} C_K |D|, 1 \right)$, $u \mapsto \alpha u + W^{-1} K(u)$ *is coercive. Indeed, we have for* $u \in L^2(D)$,

$$\frac{\left\langle \alpha u + W^{-1} K(u), u \right\rangle_{L^2(D)}}{\|u\|_{L^2(D)}}$$

$$\geq \alpha \|u\|_{L^2(D)} - \left\| W^{-1} \right\|_{\mathrm{op}} \|K(u)\|_{L^2(D)} \geq \underbrace{\left( \alpha - \left\| W^{-1} \right\|_{\mathrm{op}} C_K |D| \right)}_{>0} \|u\|_{L^2(D)}.$$

*For example, we can consider a kernel*

$$k(x, y, t) = \sum_{j=1}^{J} c_j(x, y) \sigma_s(a_j(x, y)t + b_j(x, y)),$$

*where* $\sigma_s : \mathbb{R} \to \mathbb{R}$ *is the sigmoid function defined by*

$$\sigma_s(t) = \frac{1}{1 + e^{-t}}.$$

*T are functions* $a, b, c \in C(\overline{D} \times \overline{D})$ *such that*

$$\sum_{j=1}^{J} \|c_j\|_{L^\infty(D \times D)} < \left\| W^{-1} \right\|_{\mathrm{op}}^{-1} |D|^{-1}.$$

**Example 3.** *Again, we consider the case where* $n = 1$ *and* $D \subset \mathbb{R}^d$ *a is a bounded set. We assume that* $W \in C^1(\overline{D})$ *satisfies* $0 < c_1 \leq W(x) \leq c_2$. *For simplicity, we assume that* $|D| = 1$. *We consider the non-linear integral operator*

$$K(u)(x) := \int_D k(x, y, u(x)) u(y) dy, \ x \in D, \tag{E.1}$$

*where*

$$k(x, y, t) = \sum_{j=1}^{J} c_j(x, y) \sigma_{wire}(a_j(x, y)t + b_j(x, y)), \tag{E.2}$$

*in which* $\sigma_{wire} : \mathbb{R} \to \mathbb{R}$ *is the wavelet function defined by*

$$\sigma_{wire}(t) = \boldsymbol{Im} \left( e^{i\omega t} e^{-t^2} \right),$$

*and* $a_j, b_j, c_j \in C(\overline{D} \times \overline{D})$ *are such that the* $a_j(x, y)$ *are nowhere vanishing functions, that is,* $a_j(x, y) \neq 0$ *for all* $x, y \in \overline{D} \times \overline{D}$. *the and its generalizations (e.g. activation functions which do decay only exponentially).*

*The next lemma holds for any activation function with exponential decay, including the activation function* $\sigma_{wire}$ *and settles the key condition for Proposition 3 to hold.*

**Lemma 3.** *Assume that* $|D| = 1$ *and the activation function* $\sigma : \mathbb{R} \to \mathbb{R}$ *be continuous. Assume that there exists* $M_1, m_0 > 0$ *such that*

$$|\sigma(t)| \leq M_1 e^{-m_0 |t|}, \quad t \in \mathbb{R}.$$

*Let* $a_j, b_j, c_j \in C(\overline{D} \times \overline{D})$ *be such that* $a_j(x, y)$ *are nowhere vanishing functions, that is,* $a_j(x, y) \neq 0$ *for all* $x, y \in \overline{D} \times \overline{D}$. *Moreover, let* $K : L^2(D) \to L^2(D)$ *be a non-linear integral operator given in (E.1) with a kernel satisfying (E.2). Let* $\alpha > 0$ *and* $0 < c_0 \leq W(x) \leq c_1$. *Then function* $F : L^2(D) \to L^2(D)$, $F(u) = \alpha u + W^{-1} K(u)$ *is coercive.*

*Proof.* As $\overline{D}$ is compact, there is $a_0 > 0$ such that for all $j = 1, 2, \ldots, J$ we have $|a_j(x, y)| \geq a_0$ a.e. and $|b_j(x, y)| \leq b_0$ a.e. We point out that $|\sigma(t)| \leq M_1$. Next,

$$\left( \sum_{j=1}^{J} \|W^{-1} c_j\|_{L^\infty(D \times D)} \right) M_1 \varepsilon < \frac{\alpha}{4}, \tag{E.3}$$

we consider $\lambda > 0$ and $u \in L^2(D)$ and the sets

$$D_1(\lambda) = \{x \in D : |u(x)| \geq \varepsilon\lambda\},$$
$$D_2(\lambda) = \{x \in D : |u(x)| < \varepsilon\lambda\}.$$

Let $\varepsilon > 0$ be such that

$$(\sum_{j=1}^{J} \|W^{-1}c_j\|_{L^\infty(D\times D)}) \cdot M_1\varepsilon < \frac{\alpha}{4}.$$

Then, for $x \in D_2(\lambda)$,

$$\sum_{j=1}^{J} \|W^{-1}c_j\|_{L^\infty(D\times D)}|\sigma(a_j(x,y)u(x) + b_j(x,y))u(x)|$$

$$\leq \sum_{j=1}^{J} \|W^{-1}c_j\|_{L^\infty(D\times D)}M_1\epsilon\lambda \underset{(E.3)}{\leq} \frac{\alpha}{4}\lambda.$$

After $\varepsilon$ is chosen as in the above, we choose $\lambda_0 \geq \max(1, b_0/(a_0\varepsilon))$ to be sufficiently large so that for all $|t| \geq \varepsilon\lambda_0$ it holds

$$(\sum_{j=1}^{J} \|W^{-1}c_j\|_{L^\infty(D\times D)})M_1\exp(-m_0|a_0t - b_0|)t < \frac{\alpha}{4}.$$

Here, we observe that, as $\lambda_0 \geq b_0/(a_0\varepsilon)$, we have that for all $|t| \geq \varepsilon\lambda_0$, $a_0|t| - b_0 > 0$. Then, when $\lambda \geq \lambda_0$, we have for $x \in D_1(\lambda)$,

$$(\sum_{j=1}^{J} \|W^{-1}c_j\|_{L^\infty(D\times D)}\left|\sigma\left(a_j(x,y)u(x) + b_j(x,y)\right)u(x)\right| \leq \frac{\alpha}{4}.$$

When $u \in L^2(D)$ has the norm $\|u\|_{L^2(D)} = \lambda \geq \lambda_0 \geq 1$, we have

$$\left|\int_D \int_D W(x)^{-1}k(x,y,u(x))u(x)u(y)dxdy\right|$$

$$\leq \int_D \left(\int_{D_1} (\sum_{j=1}^{J} \|W^{-1}c_j\|_{L^\infty(D\times D)})M_1\exp\left(-m_0|a_0|u(x)| - b_0|\right)|u(x)|dx\right)|u(y)|dy$$

$$+ \int_D \left(\int_{D_2} \sum_{j=1}^{J} \|W^{-1}c_j\|_{L^\infty(D\times D)}|\sigma(a_j(x,y)u(x) + b_j(x,y))||u(x)|dx\right)|u(y)|dy$$

$$\leq \frac{\alpha}{4}\|u\|_{L^2(D)} + \frac{\alpha}{4}\lambda\|u\|_{L^2(D)}$$

$$\leq \frac{\alpha}{2}\|u\|_{L^2(D)}^2.$$

Hence,

$$\frac{\langle \alpha u + W^{-1}K(u), u \rangle_{L^2(D)}}{\|u\|_{L^2(D)}} \geq \frac{\alpha}{2}\|u\|_{L^2(D)},$$

and the function $u \to \alpha u + W^{-1}K(u)$ is coercive. $\qquad\square$

### E.3   Proof of Proposition 4

*Proof.* (Injectivity) Assume that

$$\sigma(Wu_1 + K(u_1) + b) = \sigma(Wu_2 + K(u_2) + b).$$

where $u_1, u_2 \in L^2(D)^n$. Since $\sigma$ is injective and $W : L^2(D)^n \to L^2(D)^n$ is bounded linear bijective, we have

$$u_1 + W^{-1}K(u_1) = u_2 + W^{-1}K(u_2) =: z.$$

Since the mapping $u \mapsto z - W^{-1}K(u)$ is contraction (because $W^{-1}K$ is contraction), by the Banach fixed-point theorem, the mapping $u \mapsto z - W^{-1}K(u)$ admit a unique fixed-point in $L^2(D)^n$, which implies that $u_1 = u_2$.

(Surjectivity) Since $\sigma$ is surjective, it is enough to show that $u \mapsto Wu + K(u) + b$ is surjective. Let $z \in L^2(D)^n$. Since the mapping $u \mapsto W^{-1}z - W^{-1}b - W^{-1}K(u)$ is contraction, by Banach fixed-point theorem, there is $u^* \in L^2(D)^n$ such that

$$u^* = W^{-1}z - W^{-1}b - W^{-1}K(u^*) \iff Wu^* + K(u^*) + b = z.$$

$\square$

### E.4 Examples for Proposition 4

**Example 4.** *We consider the case of $n = 1$, and $D \subset [0, \ell]^d$. We consider Volterra operators*

$$K(u)(x) = \int_D k(x, y, u(x), u(y))u(y)dy,$$

*where $x = (x_1, \ldots, x_d)$ and $y = (y_1, \ldots, y_d)$. We recall that $K$ is a Volterra operator if*

$$k(x, y, t, s) \neq 0 \implies y_j \leq x_j \quad \text{for all } j = 1, 2, \ldots, d. \tag{E.4}$$

*In particular, when $D = (a, b) \subset \mathbb{R}$ is an interval, the Volterra operators are of the form*

$$K(u)(x) = \int_a^x k(x, y, u(x), u(y))u(y)dy,$$

*and if $x$ is considered as a time variable, the Volterra operators are causal in the sense that the value of $K(u)(x)$ at the time $x$ depends only on $u(y)$ at the times $y \leq x$.*

*Assume that $k(x, y, t, s) \in C(\overline{D} \times \overline{D} \times \mathbb{R} \times \mathbb{R})$ is bounded and uniformly Lipschitz smooth in the $t$ and $s$ variables, that is, $k \in C(\overline{D} \times \overline{D}; C^{0,1}(\mathbb{R} \times \mathbb{R}))$.*

*Next, we consider the non-linear operator $F : L^2(D) \to L^2(D)$,*

$$F(u) = u + K(u). \tag{E.5}$$

*Assume that $u, w \in L^2(D)$ are such that $u + K(u) = w + K(w)$, so that $w - u = K(u) - K(w)$. Next, we will show that then $u = w$. We denote and $D(z_1) = D \cap ([0, z_1] \times [0, \ell]^{d-1})$,. Then for $x \in D(z_1)$ the Volterra property of the kernel implies that*

$$|u(x) - w(x)| \leq \int_D |k(x, y, u(x), u(y))u(y) - k(x, y, w(x), w(y))w(y)|dy$$

$$\leq \int_{D(z_1)} |k(x, y, u(x), u(y))u(y) - k(x, y, w(x), u(y))u(y)|dy$$

$$+ \int_{D(z_1)} |k(x, y, w(x), u(y))u(y) - k(x, y, w(x), w(y))u(y)|dy$$

$$+ \int_{D(z_1)} |k(x, y, w(x), w(y))u(y) - k(x, y, w(x), w(y))w(y)|dy$$

$$\leq 2\|k\|_{C(\overline{D} \times \overline{D}; C^{0,1}(\mathbb{R} \times \mathbb{R}))}\|u - w\|_{L^2(D(z_1))}\|u\|_{L^2(D(z_1))}$$

$$+ \|k\|_{L^\infty(D \times D \times \mathbb{R} \times \mathbb{R})}\|u - w\|_{L^2(D(z_1))}\sqrt{|D(z_1)|},$$

*so that for all $0 < z_1 < \ell$,*

$$\|u - w\|^2_{L^2(D(z_1))}$$

$$= \int_0^{z_1} \left( \int_0^\ell \cdots \int_0^\ell \mathbf{1}_D(x)|u(x) - w(x)|^2 dx_d dx_{d-1} \ldots dx_2 \right) dx_1$$

$$\leq z_1 \ell^{d-1} \bigg( 2\|k\|_{C(\overline{D} \times \overline{D}; C^{0,1}(\mathbb{R} \times \mathbb{R}))} \|u - w\|_{L^2(D(z_1))} \|u\|_{L^2(D(z_1))}$$

$$+ \|k\|_{L^\infty(D \times D \times \mathbb{R} \times \mathbb{R})} \|u - w\|_{L^2(D(z_1))} \sqrt{|D(z_1)|} \bigg)^2$$

$$\leq z_1 \ell^{d-1} \bigg( \|k\|_{C(\overline{D} \times \overline{D}; C^{0,1}(\mathbb{R} \times \mathbb{R}))} \|u\|_{L^2(D)} + \|k\|_{L^\infty(D \times D \times \mathbb{R} \times \mathbb{R})} \sqrt{|D|} \bigg)^2 \|u - w\|^2_{L^2(D(z_1))}.$$

*Thus, when $z_1$ is so small that*

$$z_1 \ell^{d-1} \bigg( \|k\|_{C(\overline{D} \times \overline{D}; C^{0,1}(\mathbb{R}^n))} \|u\|_{L^2(D)} + \|k\|_{L^\infty(D \times D \times \mathbb{R} \times \mathbb{R})} \sqrt{|D|} \bigg)^2 < 1,$$

*we find that $\|u - w\|_{L^2(D(z_1))} = 0$, that is, $u(x) - w(x) = 0$ for $x \in D(z_1)$. Using the same arguments as above, we see for all $k \in \mathbb{N}$ that that if $u = w$ in $D(kz_1)$ then $u = w$ in $D((k+1)z_1)$. Using induction, we see that $u = w$ in $D$. Hence, the operator $u \mapsto F(u)$ is injective in $L^2(D)$.*

**Example 5.** *We consider derivatives of Volterra operators in the domain $D \subset [0, \ell]^d$. Let $K : L^2(D) \to L^2(D)$ be a non-linear operator*

$$K(u) = \int_D k(x, y, u(y))u(y)dy, \tag{E.6}$$

*where $k(x, y, t)$ satisfies (E.4), is bounded, and $k \in C(\overline{D} \times \overline{D}; C^{0,1}(\mathbb{R} \times \mathbb{R}))$. Let $F_1 : L^2(D) \to L^2(D)$ be*

$$F_1(u) = u + K(u). \tag{E.7}$$

*Then the Fréchet derivative of $K$ at $u_0 \in L^2(D)$ to the direction $w \in L^2(D)$ is*

$$DF_1|_{u_0}(w) = w(x) + \int_D k_1(x, y, u_0(y))w(y)dy, \tag{E.8}$$

*where*

$$k_1(x, y, u_0(y)) = u_0(y)\frac{\partial}{\partial t}k(x, y, t)\bigg|_{t=u_0(x)} + k(x, y, u_0(y)) \tag{E.9}$$

*is a Volterra operator satisfying*

$$k_1(x, y, t) \neq 0 \implies y_j \leq x_j \quad \text{for all } j = 1, 2, \ldots, d. \tag{E.10}$$

*As seen in Example 4, the operator $DF_1|_{u_0} : L^2(D) \to L^2(D)$ is injective.*

## F   Details in Section 4.2

In this appendix, we prove Theorem 3. We recall that in the theorem, we consider the case when $n = 1$, and $D \subset \mathbb{R}$ is a bounded interval, and the operator $F_1$ is of the form

$$F_1(u)(x) = W(x)u(x) + \int_D k(x, y, u(y))u(y)dy,$$

where $W \in C^1(\overline{D})$ satisfies $0 < c_1 \leq W(x) \leq c_2$ and the function $(x, y, s) \mapsto k(x, y, s)$ is in $C^3(\overline{D} \times \overline{D} \times \mathbb{R})$ and in $\overline{D} \times \overline{D} \times \mathbb{R}$ its three derivatives and the derivatives of $W$ are uniformly bounded by $c_0$, that is,

$$\|k\|_{C^3(\overline{D} \times \overline{D} \times \mathbb{R})} \leq c_0, \quad \|W\|_{C^1(\overline{D})} \leq c_0. \tag{F.1}$$

We recall that the identical embedding $H^1(D) \to L^\infty(D)$ is bounded and compact by Sobolev's embedding theorem.

As we will consider kernels $k(x, y, u_0(y))$, we will consider the non-linear operator $F_1$ mainly as an operator in a Sobolev space $H^1(D)$.

The Frechet derivative of $F_1$ at $u_0$ to direction $w$, denoted $A_{u_0} w = DF_1|_{u_0}(w)$ is given by

$$A_{u_0} w = W(x) w(x) + \int_D k(x, y, u_0(y)) w(y) dy + \int_D u_0(y) \frac{\partial k}{\partial u}(x, y, u_0(y)) w(y) dy. \qquad \text{(F.2)}$$

The condition (F.1) implies that

$$F_1 : H^1(D) \to H^1(D), \qquad \text{(F.3)}$$

is a locally Lipsichitz smooth function and the operator

$$A_{u_0} : H^1(D) \to H^1(D),$$

given in (F.2), is defined for all $u_0 \in C(\overline{D})$ as a bounded linear operator.

When $\mathcal{X}$ is a Banach space, let $B_{\mathcal{X}}(0, R) = \{v \in \mathcal{X} : \|v\|_{\mathcal{X}} < R\}$ and $\overline{B}_{\mathcal{X}}(0, R) = \{v \in \mathcal{X} : \|v\|_{\mathcal{X}} \leq R\}$ be the open and closed balls in $\mathcal{X}$, respectively.

We consider the Hölder spaces $C^{n,\alpha}(\overline{D})$ and their image in (leaky) ReLU-type functions. Let $a \geq 0$ and $\sigma_a(s) = \text{ReLU}(s) - a\,\text{ReLU}(-s)$. We will consider the image of the closed ball of $C^{1,\alpha}(\overline{D})$ in the map $\sigma_a$, that is $\sigma_a(\overline{B}_{C^{1,\alpha}(\overline{D})}(0, R)) = \{\sigma_a \circ g \in C(\overline{D}) : \|g\|_{C^{1,\alpha}(\overline{D})} \leq R\}$.

We will below assume that for all $u_0 \in C(\overline{D})$ the integral operator

$$A_{u_0} : H^1(D) \to H^1(D) \text{ is an injective operator.} \qquad \text{(F.4)}$$

This condition is valid when $K(u)$ is a Volterra operator, see Examples 4 and 5. As the integral operators $A_{u_0}$ are Fredholm operators having index zero, this implies that the operators (F.4) are bijective.

The inverse operator $A_{u_0}^{-1} : H^1(D) \to H^1(D)$ can be written as

$$A_{u_0}^{-1} v(x) = \widetilde{W}(x) v(x) - \int_D \widetilde{k}_{u_0}(x, y) v(y) dy, \qquad \text{(F.5)}$$

where $\widetilde{k}_{u_0}, \partial_x \widetilde{k}_{u_0} \in C(\overline{D} \times \overline{D})$ and $\widetilde{W} \in C^1(\overline{D})$.

We will consider the inverse function of the map $F_1$ in a set $\mathcal{Y} \subset \sigma_a(\overline{B}_{C^{1,\alpha}(\overline{D})}(0, R))$ that is a compact subset of the Sobolev space $H^1(D)$. To this end, we will cover the set $\mathcal{Y}$ with small balls $B_{H^1(D)}(g_j, \varepsilon_0)$, $j = 1, 2, \ldots, J$ of $H^1(D)$, centred at $g_j = F_1(v_j)$, where $v_j \in H^1(D)$. We will show below that when $g \in B_{H^1(D)}(g_j, 2\varepsilon_0)$, that is, $g$ is $2\varepsilon_1$-close to the function $g_j$ in $H^1(D)$, the inverse map of $F_1$ can be written as a limit $(F_1^{-1}(g), g) = \lim_{m \to \infty} \mathcal{H}_j^{\circ m}(v_j, g)$ in $H^1(D)^2$, where

$$\mathcal{H}_j \begin{pmatrix} u \\ g \end{pmatrix} = \begin{pmatrix} u - A_{v_j}^{-1}(F_1(u) - F_1(v_j)) + A_{v_j}^{-1}(g - g_j) \\ g \end{pmatrix},$$

that is, near $g_j$ we can approximate $F_1^{-1}$ as a composition $\mathcal{H}_j^{\circ m}$ of $2m$ layers of neural operators.

To glue the local inverse maps together, we use a partition of unity in the function space $\mathcal{Y}$ that is given by integral neural operators

$$\Phi_{\vec{i}}(v, w) = \pi_1 \circ \phi_{\vec{i},1} \circ \phi_{\vec{i},2} \circ \cdots \circ \phi_{\vec{i},\ell_0}(v, w), \quad \text{where} \quad \phi_{\vec{i},\ell}(v, w) = (F_{y_\ell, s(\vec{i},\ell), \epsilon_1}(v, w), w),$$

where $\vec{i}$ belongs in some finite index set $\mathcal{I} \subset \mathbb{Z}^{\ell_0}$, some $\epsilon_1 > 0$, some $y_\ell \in D$ ($\ell = 1, \ldots, \ell_0$), $s(\vec{i}, \ell) = i_\ell \epsilon_1 . \pi_1(v, w) = v$ that maps a pair $(v, w)$ to the first function $v$ Here, $F_{z,s,h}(v, w)$ are integral neural operators with distributional kernels

$$F_{z,s,h}(v, w)(x) = \int_D k_{z,s,h}(x, y, v(x), w(y)) dy,$$

where $k_{z,s,h}(x, y, v(x), w(y)) = v(x)\mathbf{1}_{[s-\frac{1}{2}h, s+\frac{1}{2}h)}(w(y))\delta(y - z)$, and $\mathbf{1}_A$ is the indicator function of a set $A$ and $y \mapsto \delta(y - z)$ is the Dirac delta distribution at the point $z \in D$. Using these, we can write the inverse of $F_1$ at $g \in \mathcal{Y}$ as

$$F_1^{-1}(g) = \lim_{m\to\infty} \sum_{\vec{i}\in\mathcal{I}} \Phi_{\vec{i}} \mathcal{H}_{j(\vec{i})}^{\circ m} \begin{pmatrix} v_{j(\vec{i})} \\ g \end{pmatrix}, \tag{F.6}$$

where $j(\vec{i}) \in \{1, 2, \ldots, J\}$ are suitably chosen and the limit is taken in the norm topology of $H^1(D)$. This result is summarized in the following theorem, that is a modified version of Theorem 3 for the inverse operator $F_1^{-1}$ in (F.6) where we have refined the partition of unity $\Phi_{\vec{i}}$ so that we use indexes $\vec{i} \in \mathcal{I} \subset \mathbb{Z}^{\ell_0}$ instead of $j \in \{1, \ldots, J\}$.

This result is summarized in following theorem:

**Theorem 4.** *Assume that $F_1$ satisfies the above assumptions (F.1) and (F.4) and that $F_1 : H^1(D) \to H^1(D)$ is a bijection. Let $\mathcal{Y} \subset \sigma_a(\overline{B}_{C^{1,\alpha}(\overline{D})}(0, R))$ be a compact subset of the Sobolev space $H^1(D)$, where $\alpha > 0$ and $a \geq 0$. Then the inverse of $F_1 : H^1(D) \to H^1(D)$ in $\mathcal{Y}$ can written as a limit (F.6) that is, as a limit of integral neural operators.*

Observe that Theorem 4 includes the case where $a = 1$, in which case $\sigma_a = Id$ and $\mathcal{Y} \subset \sigma_a(\overline{B}_{C^{1,\alpha}(\overline{D})}(0, R)) = \overline{B}_{C^{1,\alpha}(\overline{D})}(0, R)$. We note that when $\sigma_a$ is a leaky ReLU-function with parameter $a > 0$, Theorem 4 can be applied to compute the inverse of $\sigma_a \circ F_1$ that is given by $F_1^{-1} \circ \sigma_a^{-1}$, where $\sigma_a^{-1} = \sigma_{1/a}$. Note that the assumption that $\mathcal{Y} \subset \sigma_a(\overline{B}_{C^{1,\alpha}(\overline{D})}(0, R))$ makes it possible to apply Theorem 4 in the case when one trains deep neural networks having layers $\sigma_a \circ F_1$ and the parameter $a$ of the leaky ReLU-function is a free parameter which is also trained.

*Proof.* As the operator $F_1$ can be multiplied by function $W(x)^{-1}$, it is sufficient to consider the case when $W(x) = 1$.

Below, we use the fact that as $D \subset \mathbb{R}$, Sobolev's embedding theorem yields that the embedding $H^1(D) \to C(\overline{D})$ is bounded and there is $C_S > 0$ such that

$$\|u\|_{C(\overline{D})} \leq C_S \|u\|_{H^1(D)}. \tag{F.7}$$

For clarity, we denote below the norm of $C(\overline{D})$ by $\|u\|_{L^\infty(D)}$.

Next we considre the Frechet derivatives of $F_1$. We recall that the 1st Frechet derivative of $F_1$ at $u_0$ is the operator $A_{u_0}$. The 2nd Frechet derivative of $F_1$ at $u_0$ to directions $w_1$ and $w_2$ is

$$\begin{aligned} D^2 F_1|_{u_0}(w_1, w_2) &= \int_D 2\frac{\partial k}{\partial u}(x, y, u_0(y))w_1(y)w_2(y)dy + \int_D u_0(y)\frac{\partial k^2}{\partial u^2}(x, y, u_0(y))w_1(y)w_2(y)dy \\ &= \int_D p(x, y)w_1(y)w_2(y)dy, \end{aligned}$$

where

$$p(x, y) = 2\frac{\partial k}{\partial u}(x, y, u_0(y)) + u_0(y)\frac{\partial k^2}{\partial u^2}(x, y, u_0(y)), \tag{F.8}$$

and

$$\frac{\partial}{\partial x}p(x, y) = 2\frac{\partial^2 k}{\partial u\partial x}(x, y, u_0(y)) + u_0(y)\frac{\partial k^3}{\partial u^2\partial x}(x, y, u_0(y)). \tag{F.9}$$

Thus,

$$\tag{F.10}$$

$$\|D^2 F_1|_{u_0}(w_1, w_2)\|_{H^1(D)} \leq 3|D|^{1/2}\|k\|_{C^3(D\times D\times\mathbb{R})}(1 + \|u_0\|_{L^\infty(D)})\|w_1\|_{L^\infty(D)}\|w_2\|_{L^\infty(D)}.$$

When we freeze the function $u$ in kernel $k$ to be $u_0$, we denote

$$K_{u_0}v(x) = \int_D k(x, y, u_0(y))v(y)dy,$$

**Lemma 4.** *For $u_0, u_1 \in C(\overline{D})$ we have*

$$\|K_{u_1} - K_{u_0}\|_{L^2(D) \to H^1(D)} \leq \|k\|_{C^2(D \times D \times \mathbb{R})}|D|\|u_1 - u_0\|_{L^\infty(D)}.$$

*and*

$$\|A_{u_1} - A_{u_0}\|_{L^2(D) \to H^1(D)} \leq 2\|k\|_{C^2(D \times D \times \mathbb{R})}|D|(1 + \|u_0\|_{L^\infty(D)})\|u_1 - u_0\|_{L^\infty(D)}. \quad \text{(F.11)}$$

*Proof.* Denote

$$
\begin{aligned}
M_{u_0}v(x) &= \int_D u_0(y)\frac{\partial k}{\partial u}(x, y, u_0(y))v(y)dy, \\
N_{u_1, u_2}v(x) &= \int_D u_1(y)\frac{\partial k}{\partial u}(x, y, u_2(y))v(y)dy.
\end{aligned}
$$

We have

$$M_{u_2}v - M_{u_1}v = (N_{u_2, u_2}v - N_{u_2, u_1}v) + (N_{u_2, u_1}v - N_{u_1, u_1}v).$$

By Schur's test for continuity of integral operators,

$$
\begin{aligned}
\|K_{u_0}\|_{L^2(D) \to L^2(D)} &\leq \left(\sup_{x \in D}\int_D |k(x, y, u_0(y))|dy\right)^{1/2}\left(\sup_{y \in D}\int_D |k(x, y, u_0(y))|dx\right)^{1/2} \\
&\leq \|k\|_{C^0(D \times D \times \mathbb{R})},
\end{aligned}
$$

and

$$
\begin{aligned}
&\|M_{u_0}\|_{L^2(D) \to L^2(D)} \\
&\leq \left(\sup_{x \in D}\int_D |u_0(y)\frac{\partial k}{\partial u}(x, y, u_0(y))|dy\right)^{1/2}\left(\sup_{y \in D}\int_D |u_0(y)\frac{\partial k}{\partial u}(x, y, u_0(y))|dx\right)^{1/2} \\
&\leq \|k\|_{C^1(D \times D \times \mathbb{R})}\|u\|_{C(\overline{D})},
\end{aligned}
$$

and

$$
\begin{aligned}
&\|K_{u_2} - K_{u_1}\|_{L^2(D) \to L^2(D)} \\
&\leq \left(\sup_{x \in D}\int_D |k(x, y, u_2(y)) - k(x, y, u_1(y))|dy\right)^{1/2} \\
&\qquad\qquad \times \left(\sup_{y \in D}\int_D |k(x, y, u_2(y)) - k(x, y, u_1(y))|dx\right)^{1/2} \\
&\leq \left(\|k\|_{C^1(D \times D \times \mathbb{R})}\int_D |u_2(y) - u_1(y))|dy\right)^{1/2} \\
&\qquad\qquad \times \left(\|k\|_{C^1(D \times D \times \mathbb{R})}\sup_{y \in D}\int_D |u_2(y) - u_1(y))|dx\right)^{1/2} \\
&\leq \|k\|_{C^1(D \times D \times \mathbb{R})}\left(\int_D |u_2(y) - u_1(y))|dy\right)^{1/2}\left(\sup_{y \in D}\int_D |u_2(y) - u_1(y))|dx\right)^{1/2} \\
&\leq \|k\|_{C^1(D \times D \times \mathbb{R})}\left(|D|^{1/2}\|u_2 - u_1\|_{L^2(D)}\right)^{1/2}\left(|D|\sup_{y \in D}|u_2(y) - u_1(y))|\right)^{1/2} \\
&\leq \|k\|_{C^1(D \times D \times \mathbb{R})}|D|^{3/4}\|u_2 - u_1\|_{L^2(D)}^{1/2}\|u_2 - u_1\|_{L^\infty(D)}^{1/2} \\
&\leq \|k\|_{C^1(D \times D \times \mathbb{R})}|D|\|u_2 - u_1\|_{L^\infty(D)},
\end{aligned}
$$

and

$$\|N_{u_2,u_2} - N_{u_2,u_1}\|_{L^2(D)\to L^2(D)}$$

$$\leq \left(\sup_{x\in D}\int_D |u_2(y)k(x,y,u_2(y)) - u_2(y)k(x,y,u_1(y))|dy\right)^{1/2}$$

$$\times\left(\sup_{y\in D}\int_D |u_2(y)k(x,y,u_2(y)) - u_2(y)k(x,y,u_1(y))|dx\right)^{1/2}$$

$$\leq \|k\|_{C^1(D\times D\times\mathbb{R})}|D|^{3/4}\|u_2\|_{C^0(D)}\|u_2 - u_1\|_{L^2(D)}^{1/2}\|u_2 - u_1\|_{L^\infty(D)}^{1/2}$$

$$\leq \|k\|_{C^1(D\times D\times\mathbb{R})}|D|\|u_2\|_{C^0(D)}\|u_2 - u_1\|_{L^\infty(D)},$$

and

$$\|N_{u_2,u_1} - N_{u_1,u_1}\|_{L^2(D)\to L^2(D)}$$

$$\leq \left(\sup_{x\in D}\int_D |(u_2(y) - u_1(y))k(x,y,u_1(y))|dy\right)^{1/2}$$

$$\times\left(\sup_{y\in D}\int_D |(u_2(y) - u_1(y))k(x,y,u_1(y))|dx\right)^{1/2}$$

$$\leq \|k\|_{C^0(D\times D\times\mathbb{R})}|D|\|u_2 - u_1\|_{L^\infty(D)},$$

so that

$$\|M_{u_2} - M_{u_1}\|_{L^2(D)\to L^2(D)}$$

$$\leq \|k\|_{C^1(D\times D\times\mathbb{R})}|D|(1 + \|u_2\|_{C^0(D)})\|u_2 - u_1\|_{L^\infty(D)}.$$

Also, when $D_x v = \frac{dv}{dx}$,

$$\|D_x \circ K_{u_0}\|_{L^2(D)\to L^2(D)}$$

$$\leq \left(\sup_{x\in D}\int_D |D_x k(x,y,u_0(y))|dy\right)^{1/2}\left(\sup_{y\in D}\int_D |D_x k(x,y,u_0(y))|dx\right)^{1/2}$$

$$\leq \|k\|_{C^1(D\times D\times\mathbb{R})},$$

and

$$\|D_x \circ K_{u_1} - D_x \circ K_{u_0}\|_{L^2(D)\to L^2(D)}$$

$$\leq \left(\sup_{x\in D}\int_D |D_x k(x,y,u_1(y)) - D_x k(x,y,u_0(y))|dy\right)^{1/2}$$

$$\times\left(\sup_{y\in D}\int_D |D_x k(x,y,u_1(y)) - D_x k(x,y,u_0(y))|dx\right)^{1/2}$$

$$\leq \left(\|k\|_{C^2(D\times D\times\mathbb{R})}\int_D |u_1(y) - u_0(y))|dy\right)^{1/2}$$

$$\times\left(\|k\|_{C^2(D\times D\times\mathbb{R})}\sup_{y\in D}\int_D |u_1(y) - u_0(y))|dx\right)^{1/2}$$

$$\leq \|k\|_{C^2(D\times D\times\mathbb{R})}\left(\int_D |u_1(y) - u_0(y))|dy\right)^{1/2}\left(\sup_{y\in D}\int_D |u_1(y) - u_0(y))|dx\right)^{1/2}$$

$$\leq \|k\|_{C^2(D\times D\times\mathbb{R})}\left(|D|^{1/2}\|u_1 - u_0\|_{L^2(D)}\right)^{1/2}\left(|D|\sup_{y\in D}|u_1(y) - u_0(y))|\right)^{1/2}$$

$$\leq \|k\|_{C^2(D\times D\times\mathbb{R})}|D|^{3/4}\|u_1 - u_0\|_{L^2(D)}^{1/2}\|u_1 - u_0\|_{L^\infty(D)}^{1/2}$$

$$\leq \|k\|_{C^2(D\times D\times\mathbb{R})}|D|\|u_1 - u_0\|_{L^\infty(D)}.$$

Thus,

$$\|K_{u_0}\|_{L^2(D)\to H^1(D)} \leq \|k\|_{C^1(D\times D\times\mathbb{R})},$$

and

$$\|M_{u_0}\|_{L^2(D)\to H^1(D)} \quad \leq \quad \|u_0\|_{C^0(D)}\|k\|_{C^1(D\times D\times\mathbb{R})},$$

and

$$\|K_{u_1} - K_{u_0}\|_{L^2(D)\to H^1(D)} \quad \leq \quad \|k\|_{C^2(D\times D\times\mathbb{R})}|D|\|u_1 - u_0\|_{L^\infty(D)}.$$

Similarly,

$$\|M_{u_1} - M_{u_0}\|_{L^2(D)\to H^1(D)} \quad \leq \quad \|k\|_{C^2(D\times D\times\mathbb{R})}|D|(1 + \|u_2\|_{C^0(D)})\|u_1 - u_0\|_{L^\infty(D)}.$$

As $A_{u_1} = K_{u_1} + M_{u_1}$, the claim follows. $\qquad\qquad\square$

As the embedding $H^1(D) \to C(\overline{D})$ is bounded and has norm $C_S$, Lemma 4 implies that for all $R > 0$ there is

$$C_L(R) = 2\|k\|_{C^2(D\times D\times\mathbb{R})}|D|(1 + C_S R),$$

such that the map,

$$u_0 \mapsto DF_1|_{u_0}, \quad u_0 \in \overline{B}_{H^1}(0, R), \tag{F.12}$$

is Lipschitz map $\overline{B}_{H^1}(0, R) \to \mathcal{L}(H^1(D), H^1(D))$ with a Lipschitz constant $C_L(R)$, that is,

$$\|DF_1|_{u_1} - DF_1|_{u_2}\|_{H^1(D)\to H^1(D)} \leq C_L(R)\|u_1 - u_2\|_{H^1(D)}. \tag{F.13}$$

As $u_0 \mapsto A_{u_0} = DF_1|_{u_0}$ is continuous, the inverse $A_{u_0}^{-1} : H^1(D) \to H^1(D)$ exists for all $u_0 \in C(\overline{D})$, and the embedding $H^1(D) \to C(\overline{D})$ is compact, we have that for all $R > 0$ there is $C_B(R) > 0$ such that

$$\|A_{u_0}^{-1}\|_{H^1(D)\to H^1(D)} \leq C_B(R), \quad \text{for all } u_0 \in \overline{B}_{H^1}(0, R). \tag{F.14}$$

Let $R_1, R_2 > 0$ be such that $\mathcal{Y} \subset \overline{B}_{H^1}(0, R_1)$ and $X = F_1^{-1}(\mathcal{Y}) \subset \overline{B}_{H^1}(0, R_2)$. Below, we denote $C_L = C_L(2R_2)$ and $C_B = C_B(R_2)$.

Next we consider inverse of $F_1$ in $\mathcal{Y}$. To this end, let us consider $\varepsilon_0 > 0$, which we choose later to be small enough. As $\mathcal{Y} \subset \overline{B}_{H^1}(0, R)$ is compact there are finite number of elements $g_j = F_1(v_j) \in \mathcal{Y}$, where $v_j \in X, j = 1, 2, \ldots, J$ such that

$$\mathcal{Y} \subset \bigcup_{j=1}^{J} B_{H^1(D)}(g_j, \varepsilon_0).$$

We observe that for $u_0, u_1 \in X$,

$$A_{u_1}^{-1} - A_{u_0}^{-1} = A_{u_1}^{-1}(A_{u_1} - A_{u_0})A_{u_0}^{-1},$$

and hence the Lipschitz constant of $A^{-1} : u \mapsto A_u^{-1}, X \to \mathcal{L}(H^1(D), H^1(D))$ satisfies

$$Lip(A_\cdot^{-1}) \leq C_A = C_B^2 C_L, \tag{F.15}$$

see (F.11).

Let us consider a fixed $j$ and $g_j \in \mathcal{Y}$. When $g$ satisfies

$$\|g - g_j\|_{H^1(D)} < 2\varepsilon_0, \tag{F.16}$$

the equation

$$F_1(u) = g, \quad u \in X,$$

is equivalent to the fixed point equation

$$u = u - A_{v_j}^{-1}(F_1(u) - F_1(v_j)) + A_{v_j}^{-1}(g - g_j),$$

that is equivalent to the fixed point equation

$$H_j(u) = u,$$

for the function $H_j : H^1(D) \to H^1(D)$,

$$H_j(u) = u - A_{v_j}^{-1}(F_1(u) - F_1(v_j)) + A_{v_j}^{-1}(g - g_j).$$

Note that $H_j$ depends on $g$, and thus we later denote $H_j = H_j^g$. We observe that

$$H_j(v_j) = v_j + A_{v_j}^{-1}(g - g_j). \tag{F.17}$$

Let $u, v \in \overline{B}_{H^1}(0, 2R_2)$. We have

$$F_1(u) = F_1(v) + A_v(u - v) + B_v(u - v), \quad \|B_v(u - v)\| \leq C_0\|u - v\|^2,$$

where, see (F.10),

$$C_0 = 3|D|^{1/2}\|k\|_{C^3(D \times D \times \mathbb{R})}(1 + 2C_S R_2)C_S^2,$$

so that for $u_1, u_2 \in \overline{B}_{H^1}(0, 2R_2)$,

$$u_1 - u_2 - A_{v_j}^{-1}(F_1(u_1) - F_1(u_2))$$
$$= u_1 - u_2 - A_{u_2}^{-1}(F_1(u_1) - F_1(u_2)) - (A_{u_2}^{-1} - A_{v_j}^{-1})(F_1(u_1) - F_1(u_2)),$$

and

$$\|u_1 - u_2 - A_{u_2}^{-1}(F_1(u_1) - F_1(u_2))\|_{H^1(D)}$$
$$= \|A_{u_2}^{-1}(B_{u_2}(u_1 - u_2))\|_{H^1(D)}$$
$$\leq \|A_{u_2}^{-1}\|_{H^1(D) \to H^1(D)}\|B_{u_2}(u_1 - u_2)\|_{H^1(D)}$$
$$\leq \|A_{u_2}^{-1}\|_{H^1(D) \to H^1(D)}C_0\|u_1 - u_2\|_{H^1(D)}^2,$$
$$\leq C_B C_0\|u_1 - u_2\|_{H^1(D)}^2,$$

and

$$\|(A_{u_2}^{-1} - A_{v_j}^{-1})(F_1(u_1) - F_1(u_2))\|_{H^1(D)}$$
$$\leq \|A_{u_2}^{-1} - A_{v_j}^{-1}\|_{H^1(D) \to H^1(D)}\|F_1(u_1) - F_1(u_2)\|_{H^1(D)}$$
$$\leq Lip_{\overline{B}_{H^1}(0, 2R_2) \to H^1(D)}(A_\cdot^{-1})\|u_2 - v_j\|Lip_{\overline{B}_{H^1}(0, 2R_2) \to H^1(D)}(F_1)\|u_2 - u_1\|_{H^1(D)}$$
$$\leq C_A\|u_2 - v_j\|(C_B + 4C_0 R_2)\|u_2 - u_1\|_{H^1(D)},$$

see (F.2), and hence, when $\|u - v_j\| \leq r \leq R_2$,

$$\|H_j(u_1) - H_j(u_2)\|_{H^1(D)}$$
$$\leq \|u_1 - u_2 - A_{v_j}^{-1}(F_1(u_1) - F_1(u_2))\|_{H^1(D)}$$
$$\leq \|u_1 - u_2 - A_{u_2}^{-1}(F_1(u_1) - F_1(u_2))\|_{H^1(D)} + \|(A_{u_2}^{-1} - A_{v_j}^{-1})(F_1(u_1) - F_1(u_2))\|_{H^1(D)}$$
$$\leq \left(C_B C_0(\|u_1 - v_j\|_{H^1(D)} + \|u_2 - v_j\|_{H^1(D)}) + C_A(C_B + 4C_0 R_2)\|u_2 - v_j\|\right)\|u_2 - u_1\|_{H^1(D)}$$
$$\leq C_H r\|u_2 - u_1\|_{H^1(D)},$$

where

$$C_H = 2C_B C_0 + C_A(C_B + 4C_0 R_2).$$

We now choose

$$r = \min(\frac{1}{2C_H}, R_2).$$

We consider

$$\varepsilon_0 \leq \frac{1}{8C_B}\frac{1}{2C_H}.$$

Then, we have

$$r \geq 2C_B\varepsilon_0/(1 - C_H r).$$

Then, we have that $\mathrm{Lip}_{\overline{B}_{H^1}(0,2R_2)\to H^1(D)}(H_j) \leq a = C_H r < \frac{1}{2}$, and

$$r \geq \|A_{v_j}^{-1}\|_{H^1(D)\to H^1(D)}\|g - g_j\|_{H^1(D)}/(1-a),$$

and for all $u \in \overline{B}_{H^1}(0, R_2)$ such that $\|u - v_j\| \leq r$, we have $\|A_{v_j}^{-1}(g - g_j)\|_{H^1(D)} \leq (1-a)r$. Then,

$$\begin{aligned}
\|H_j(u) - v_j\|_{H^1(D)} &\leq \|H_j(u) - H_j(v_j)\|_{H^1(D)} + \|H_j(v_j) - v_j\|_{H^1(D)} \\
&\leq a\|u - v_j\|_{H^1(D)} + \|v_j + A_{v_j}^{-1}(g - g_j) - v_j\|_{H^1(D)} \\
&\leq ar + \|A_{v_j}^{-1}(g - g_j)\|_{H^1(D)} \leq r,
\end{aligned}$$

that is, $H_j$ maps $\overline{B}_{H^1(D)}(v_j, r)$ to itself. By Banach fixed point theorem, $H_j : \overline{B}_{H^1(D)}(v_j, r) \to \overline{B}_{H^1(D)}(v_j, r)$ has a fixed point.

Let us denote

$$\mathcal{H}_j\begin{pmatrix} u \\ g \end{pmatrix} = \begin{pmatrix} H_j^g(u) \\ g \end{pmatrix} = \begin{pmatrix} u - A_{v_j}^{-1}(F_1(u) - F_1(v_j)) + A_{v_j}^{-1}(g - g_j) \\ g \end{pmatrix}.$$

By the above, when we choose $\varepsilon_0$ to have a value

$$\varepsilon_0 < \frac{1}{8C_B}\frac{1}{2C_H},$$

the map $F_1$ has a right inverse map $\mathcal{R}_j$ in $B_{H^1}(g_j, 2\varepsilon_0)$, that is,

$$F_1(\mathcal{R}_j(g)) = g, \quad \text{for } g \in B_{H^1}(g_j, 2\varepsilon_0), \tag{F.18}$$

it holds that $\mathcal{R}_j : B_{H^1}(g_j, 2\varepsilon_0) \to \overline{B}_{H^1(D)}(v_j, r)$, and by Banach fixed point theorem it is given by the limit

$$\mathcal{R}_j(g) = \lim_{m\to\infty} w_{j,m}, \quad g \in B_{H^1}(g_j, 2\varepsilon_0), \tag{F.19}$$

in $H^1(D)$, where

$$w_{j,0} = v_j, \tag{F.20}$$
$$w_{j,m+1} = H_j^g(w_{j,m}). \tag{F.21}$$

We can write for $g \in B_{H^1}(g_j, 2\varepsilon_0)$,

$$\begin{pmatrix} \mathcal{R}_j(g) \\ g \end{pmatrix} = \lim_{m\to\infty} \mathcal{H}_j^{\circ m}\begin{pmatrix} v_j \\ g \end{pmatrix},$$

where the limit takes space in $H^1(D)^2$ and

$$\mathcal{H}_j^{\circ m} = \mathcal{H}_j \circ \mathcal{H}_j \circ \cdots \circ \mathcal{H}_j, \tag{F.22}$$

is the composition of $m$ operators $\mathcal{H}_j$. This implies that $\mathcal{R}_j$ can be written as a limit of finite iterations of neural operators $H_j$ (we will consider how the operator $A_{v_j}^{-1}$ can be written as a neural operator below).

As $\mathcal{Y} \subset \sigma_a(\overline{B}_{C^{1,\alpha}(\overline{D})}(0, R))$, there are finite number of points $y_\ell \in D, \ell = 1, 2, \ldots, \ell_0$ and $\varepsilon_1 > 0$ such that the sets

$$Z(i_1, i_2, \ldots, i_{\ell_0}) = \{g \in \mathcal{Y} : (i_\ell - \frac{1}{2})\varepsilon_1 \leq g(y_\ell) < (i_\ell + \frac{1}{2})\varepsilon_1, \text{ for all } \ell\},$$

where $i_1, i_2, \ldots, i_{\ell_0} \in \mathbb{Z}$, satisfy the condition

$$\tag{F.23}$$

If $(Z(i_1, i_2, \ldots, i_{\ell_0}) \cap \mathcal{Y}) \cap B_{H^1(D)}(g_j, \varepsilon_0) \neq \emptyset$ then $Z(i_1, i_2, \ldots, i_{\ell_0}) \cap \mathcal{Y} \subset B_{H^1(D)}(g_j, 2\varepsilon_0)$.

To show (F.23), we will below use the mean value theorem for function $g = \sigma_a \circ v \in \mathcal{Y}$, where $v \in C^{1,\alpha}(\overline{D})$. First, let us consider the case when the parameter $a$ of the leaky ReLU function $\sigma_a$ is strictly positive. Without loss of generality, we can assume that $D = [0, 1]$ and $y_\ell = h\ell$, where

$h = 1/\ell_0$ and $\ell = 0, 1, \ldots, \ell_0$. We consider $g \in \mathcal{Y} \cap Z(i_1, i_2, \ldots, i_{\ell_0}) \subset \sigma_a(\overline{B}_{C^{1,\alpha}(\overline{D})}(0, R))$ of the form $g = \sigma_a \circ v$. As $a$ is non-zero, the inequality $(i_\ell - \frac{1}{2})\varepsilon \leq g(y_\ell) < (i_\ell + \frac{1}{2})\varepsilon$ is equivalent to $\sigma_{1/a}((i_\ell - \frac{1}{2})\varepsilon) \leq v(y_\ell) < \sigma_{1/a}((i_\ell + \frac{1}{2})\varepsilon)$, and thus

$$\sigma_{1/a}(i_\ell \varepsilon) - A\varepsilon \leq v(y_\ell) < \sigma_{1/a}(i_\ell \varepsilon) + A\varepsilon, \tag{F.24}$$

where $A = \frac{1}{2}\max(1, a, 1/a)$, that is, for $g = \sigma_a(v) \in Z(i_1, i_2, \ldots, i_{\ell_0})$ the values $v(y_\ell)$ are known within small errors. By applying mean value theorem on the interval $[(\ell - 1)h, \ell h]$ for function $v$ we see that there is $x' \in [(\ell - 1)h, \ell h]$ such that

$$\frac{dv}{dx}(x') = \frac{v(\ell h) - v((\ell - 1)h)}{h},$$

and thus by (F.24),

$$\left|\frac{dv}{dx}(x') - d_{\ell, \vec{i}}\right| \leq 2A\frac{\varepsilon_1}{h}, \tag{F.25}$$

where

$$d_{\ell, \vec{i}} = \frac{1}{h}(\sigma_{1/a}(i_\ell \varepsilon_1) - \sigma_{1/a}((i_\ell - 1)\varepsilon_1)), \tag{F.26}$$

Observe that these estimates are useful when $\varepsilon_1$ is much smaller that $h$. As $g = \sigma_a \circ v \in \mathcal{Y} \subset \sigma_a(\overline{B}_{C^{1,\alpha}(\overline{D})}(0, R))$, we have $v \in \overline{B}_{C^{1,\alpha}(\overline{D})}(0, R)$, so that $\frac{dv}{dx} \in \overline{B}_{C^{0,\alpha}(\overline{D})}(0, R)$ satisfies (F.25) implies that

$$\left|\frac{dv}{dx}(x) - d_{\ell, \vec{i}}\right| \leq 2A\frac{\varepsilon_1}{h} + Rh^\alpha, \quad \text{for all } x \in [(\ell - 1)h, \ell h]. \tag{F.27}$$

Moreover, (F.24) and $v \in \overline{B}_{C^{1,\alpha}(\overline{D})}(0, R)$ imply

$$|v(x) - \sigma_{1/a}(i_\ell \varepsilon_1)| < A\varepsilon_1 + Rh, \tag{F.28}$$

for all $x \in [(\ell - 1)h, \ell h]$.

Let $\varepsilon_2 = \varepsilon_0/A$. When we first choose $\ell_0$ to be large enough (so that $h = 1/\ell_0$ is small) and then $\varepsilon_1$ to be small enough, we may assume that

$$\max(2A\frac{\varepsilon_1}{h} + Rh^\alpha, A\varepsilon_1 + Rh) < \frac{1}{8}\varepsilon_2. \tag{F.29}$$

Then for any two functions $g, g' \in \mathcal{Y} \cap Z(i_1, i_2, \ldots, i_{\ell_0}) \subset \sigma_a(\overline{B}_{C^{1,\alpha}(\overline{D})}(0, R))$ of the form $g = \sigma_a \circ v, g' = \sigma_a \circ v'$ the inequalities (F.27) and (F.28) imply

$$\left|\frac{dv}{dx}(x) - \frac{dv'}{dx}(x)\right| < \frac{1}{4}\varepsilon_2, \tag{F.30}$$

$$|v(x) - v'(x)| < \frac{1}{4}\varepsilon_2,$$

for all $x \in D$. As $v, v' \in \overline{B}_{C^{1,\alpha}(\overline{D})}(0, R)$, this implies

$$\|v - v'\|_{C^1(\overline{D})} < \frac{1}{2}\varepsilon_2,$$

As the embedding $C^1(\overline{D}) \to H^1(D)$ is continuous and has norm less than 2 on the interval $D = [0, 1]$, we see that

$$\|v - v'\|_{H^1(\overline{D})} < \varepsilon_2,$$

and thus

$$\|g - g'\|_{H^1(\overline{D})} < A\varepsilon_2 = \varepsilon_0. \tag{F.31}$$

To prove (F.23), we assume that $(Z(i_1, i_2, \ldots, i_{\ell_0}) \cap \mathcal{Y}) \cap B_{H^1(D)}(g_j, \varepsilon_0) \neq \emptyset$, and $g \in Z(i_1, i_2, \ldots, i_{\ell_0}) \cap \mathcal{Y}$. By the assumption, there exists $g^* \in (Z(i_1, i_2, \ldots, i_{\ell_0}) \cap \mathcal{Y}) \cap B_{H^1(D)}(g_j, \varepsilon_0)$. Using (F.31), we have

$$\|g - g_j\|_{H^1(D)} \leq \|g - g^*\|_{H^1(D)} + \|g^* - g_j\|_{H^1(D)} \leq 2\epsilon_0.$$

Thus, $g \in B_{H^1(D)}(g_j, 2\varepsilon_0)$, which implies that the property (F.23) follows.

We next consider the case when the parameter $a$ of the leaky relu function $\sigma_a$ is zero. Again, we assume that $D = [0,1]$ and $y_\ell = h\ell$, where $h = 1/\ell_0$ and $\ell = 0, 1, \dots, \ell_0$. We consider $g \in \mathcal{Y} \cap Z(i_1, i_2, \dots, i_{\ell_0}) \subset \sigma_a(\overline{B}_{C^{1,\alpha}(\overline{D})}(0, R))$ of the form $g = \sigma_a(v)$ and an interval $[\ell_1 h, (\ell_1+1)h] \subset D$, where $1 \le \ell_1 \le \ell_0 - 2$. We will consider four cases. First, if $g$ does not obtain the value zero on the interval $[\ell_1 h, (\ell_1 + 1)h]$ the mean value theorem implies that there is $x' \in [\ell_1 h, (\ell_1 + 1)h]$ such that $\frac{dg}{dx}(x') = \frac{dv}{dx}(x')$ is equal to $d = (g(\ell_1 h) - g([(\ell_1 - 1)h))/h$. Second, if $g$ does not obtain the value zero on either of the intervals $[(\ell_1 - 1)h, \ell_1 h]$ or $[(\ell_1 + 1)h, (\ell_1 + 2)h]$, we can use the mean value theorem to estimate the derivatives of $g$ and $v$ at some point of these intervals similarly to the first case. Third, if $g$ does not vanish identically on the interval $[\ell_1 h, (\ell_1+1)h]$ but it obtains the value zero on the both intervals $[(\ell_1 - 1)h, \ell_1 h]$ and $[(\ell_1 + 1)h, (\ell_1 + 2)h]$, the function $v$ has two zeros on the interval $[(\ell_1 - 1)h, (\ell_1 + 2)h]$ and the mean value theorem implies that there is $x' \in [(\ell_1 - 1)h, (\ell_1 + 2)h]$ such that $\frac{dv}{dx}(x') = 0$. Fourth, if none of the above cases are valid, $g$ vanishes identically on the interval $[\ell_1 h, (\ell_1 + 1)h]$. In all these cases the fact that $\|v\|_{C^{1,\alpha}(\overline{D})} \le R$ implies that the derivative of $g$ can be estimated on the whole interval $[\ell_1 h, (\ell_1 + 1)h]$ within a small error. Using these observations we see for any $\varepsilon_2, \varepsilon_3 > 0$ that if $y_\ell \in D = [d_1, d_2] \subset \mathbb{R}$, $\ell = 1, 2, \dots, \ell_0$ are a sufficiently dense grid in $D$ and $\varepsilon_1$ to be small enough, then the derivatives of any two functions $g, g' \in \mathcal{Y} \cap Z(i_1, i_2, \dots, i_{\ell_0}) \subset \sigma_a(\overline{B}_{C^{1,\alpha}(\overline{D})}(0, R))$ of the form $g = \sigma_a(v), g' = \sigma_a(v')$ satisfy $\|g - g'\|_{H^1([d_1 + \varepsilon_3, d_2 - \varepsilon_3])} < \varepsilon_2$. As the embedding $C^1([d_1 + \varepsilon_3, d_2 - \varepsilon_3]) \to H^1([d_1 + \varepsilon_3, d_2 - \varepsilon_3])$ is continuous,

$$\|\sigma_a(v)\|_{H^1([d_1, d_1 + \varepsilon_3])} \le c_a \|v\|_{C^{1,\alpha}(\overline{D})} \sqrt{\varepsilon_3},$$
$$\|\sigma_a(v)\|_{H^1([d_2 - \varepsilon_3, d_2])} \le c_a \|v\|_{C^{1,\alpha}(\overline{D})} \sqrt{\varepsilon_3},$$

and $\varepsilon_2$ and $\varepsilon_3$ can be chosen to be arbitrarily small, we see that the property (F.23) follows. Thus the property (F.23) is shown in all cases.

By our assumptions $\mathcal{Y} \subset \sigma_a(B_{C^{1,\alpha}(\overline{D})}(0, R))$ and hence $g \in \mathcal{Y}$ implies that $\|g\|_{C(\overline{D})} \le AR$. This implies that $\mathcal{Y} \cap Z(i_1, i_2, \dots, i_{\ell_0})$ is empty if there is $\ell$ such that $|i_\ell| > 2AR/\varepsilon_1 + 1$. Thus, there is a finite set $\mathcal{I} \subset \mathbb{Z}^{\ell_0}$ such that

$$\mathcal{Y} \subset \bigcup_{\vec{i} \in \mathcal{I}} Z(\vec{i}), \tag{F.32}$$

$$Z(\vec{i}) \cap \mathcal{Y} \neq \emptyset, \quad \text{for all } \vec{i} \in \mathcal{I}, \tag{F.33}$$

where we use notation $\vec{i} = (i_1, i_2, \dots, i_{\ell_0}) \in \mathbb{Z}^{\ell_0}$. On the other hand, we have chosen $g_j \in \mathcal{Y}$ such that $B_{H^1(D)}(g_j, \varepsilon_0)$, $j = 1, \dots, J$ cover $\mathcal{Y}$. This implies that for all $\vec{i} \in \mathcal{I}$ there is $j = j(\vec{i}) \in \{1, 2, \dots, j\}$ such that there exists $g \in Z(\vec{i}) \cap B_{H^1(D)}(g_j, \varepsilon_0)$. By (F.23), this implies that

$$Z(\vec{i}) \subset B_{H^1(D)}(g_{j(\vec{i})}, 2\varepsilon_0). \tag{F.34}$$

Thus, we see that $Z(\vec{i}), \vec{i} \in \mathcal{I}$ is a disjoint covering of $\mathcal{Y}$, and by (F.34), in each set $Z(\vec{i}) \cap \mathcal{Y}, \vec{i} \in \mathcal{I}$ the map $g \to \mathcal{R}_j(g)$ we have constructed a right inverse of the map $F_1$.

Below, we denote $s(\vec{i}, \ell) = i_\ell \varepsilon_1$. Next we construct a partition of unity in $\mathcal{Y}$ using maps

$$F_{z,s,h}(v, w)(x) = \int_D k_{z,s,h}(x, y, v(x), w(y)) dy,$$

where

$$k_{z,s,h}(x, y, v(x), w(y)) = v(x) \mathbf{1}_{[s - \frac{1}{2}h, s + \frac{1}{2}h)}(w(y)) \delta(y - z).$$

Then,

$$F_{z,s,h}(v, w)(x) = \begin{cases} v(x), & \text{if } -\frac{1}{2}h \le w(z) - s < \frac{1}{2}h, \\ 0, & \text{otherwise.} \end{cases}$$

Next, for all $\vec{i} \in \mathcal{I}$ we define the operator $\Phi_{\vec{i}} : H^1(D) \times \mathcal{Y} \to H^1(D)$,

$$\Phi_{\vec{i}}(v, w) = \pi_1 \circ \phi_{\vec{i}, 1} \circ \phi_{\vec{i}, 2} \circ \cdots \circ \phi_{\vec{i}, \ell_0}(v, w),$$

where $\phi_{\vec{i},\ell} : H^1(D) \times \mathcal{Y} \to H^1(D) \times \mathcal{Y}$ are the maps

$$\phi_{\vec{i},\ell}(v,w) = (F_{y_\ell, s(\vec{i},\ell), \varepsilon_1}(v,w), w),$$

and $\pi_1(v,w) = v$ maps a pair $(v,w)$ to the first function $v$. It satisfies

$$\Phi_{\vec{i}}(v,w) = \begin{cases} v, & \text{if } -\frac{1}{2}\varepsilon_1 \leq w(y_\ell) - s(\vec{i},\ell) < \frac{1}{2}\varepsilon_1 \text{ for all } \ell, \\ 0, & \text{otherwise.} \end{cases}$$

Observe that here $s(\vec{i},\ell) = i_\ell \varepsilon_1$ is close to the value $g_{j(\vec{i})}(y_\ell)$. Now we can write for $g \in Y$

$$F_1^{-1}(g) = \sum_{\vec{i} \in \mathcal{I}} \Phi_{\vec{i}}(\mathcal{R}_{j(\vec{i})}(g), g),$$

with suitably chosen $j(\vec{i}) \in \{1, 2, \ldots, J\}$.

Let us finally consider $A_{u_0}^{-1}$ where $u_0 \in C(\overline{D})$. Let us denote

$$\widetilde{K}_{u_0} w = \int_D u_0(y) \frac{\partial k}{\partial u}(x, y, u_0(y)) w(y) dy,$$

and $J_{u_0} = K_{u_0} + \widetilde{K}_{u_0}$ be the integral operator with kernel

$$j_{u_0}(x, y) = k(x, y, u_0(y)) + u_0(y) \frac{\partial k}{\partial u}(x, y, u_0(y)).$$

We have

$$(I + J_{u_0})^{-1} = I - J_{u_0} + J_{u_0}(I + J_{u_0})^{-1} J_{u_0},$$

so that when we write the linear bounded operator

$$A_{u_0}^{-1} = (I + J_{u_0})^{-1} : H^1(D) \to H^1(D),$$

as an integral operator

$$(I + J_{u_0})^{-1} v(x) = v + \int_D m_{u_0}(x, y) v(y) dy,$$

we have

$$\begin{aligned} &(I + J_{u_0})^{-1} v(x) \\ =\ & v(x) - J_{u_0} v(x) \\ &+ \int_D \left( \int_D \left\{ j_{u_0}(x, y') j_{u_0}(y, y') + \left( \int_D j_{u_0}(x, y') m_{u_0}(y', x') j_{u_0}(x', y) dx' \right) \right\} dy' \right) v(y) dy \\ =\ & v(x) - \int_D \widetilde{j}_{u_0}(x, y) v(y) dy, \end{aligned}$$

where

$$\widetilde{j}_{u_0}(x, y) = -j_{u_0}(x, y) + \int_D (j_{u_0}(x, y') j_{u_0}(y, y')) dy' + \int_D \int_D j_{u_0}(x, y') m_{u_0}(y', x') j_{u_0}(x', y) dx' dy'.$$

This implies that the operator $A_{u_0}^{-1} = (I + J_{u_0})^{-1}$ is a neural operator, too. Observe that $\widetilde{j}_{u_0}(x, y), \partial_x \widetilde{j}_{u_0}(x, y) \in C(\overline{D} \times \overline{D})$.

This proves Theorem 3. $\qquad \square$

