# Appendix

 ## A   Proof of Proposition 1 in Section 2

 *Proof.* We use the notation $T|_{S(v,T+b)}(v) = (T_i v)_{i \in S(v,T+b)}$. Assume that $T + b$ has a DSS with
 respect to every $v \in L^2(D)^n$ in the sense of Definition 2, and that

$$\mathrm{ReLU}(Tv^{(1)} + b) = \mathrm{ReLU}(Tv^{(2)} + b) \quad \text{in } D, \tag{A.1}$$

 where $v^{(1)}, v^{(2)} \in L^2(D)^n$. Since $T+b$ has a DSS with respect to $v^{(1)}$, we have for $i \in S(v^{(1)}, T+b)$

$$0 < \mathrm{ReLU}(T_i v^{(1)} + b_i) = \mathrm{ReLU}(T_i v^{(2)} + b_i) \text{ in } D,$$

 which implies that

$$T_i v^{(1)} + b_i = T_i v^{(2)} + b_i \text{ in } D.$$

 Thus,

$$v^{(1)} - v^{(2)} \in \mathrm{Ker}\left(T|_{S(v^{(1)}, T+b)}\right). \tag{A.2}$$

 By assuming (A.1), we have for $i \notin S(v^{(1)}, T)$,

$$\{x \in D \mid T_i v^{(1)}(x) + b_i(x) \le 0\} = \{x \in D \mid T_i v^{(2)}(x) + b_i(x) \le 0\}.$$

 Then, we have

$$T_i(v^{(1)} - (v^{(1)} - v^{(2)}))(x) + b_i(x) = T_i v^{(2)}(x) + b_i(x) \le 0 \text{ if } T_i v^{(1)}(x) + b_i(x) \le 0,$$

 that is,

$$T_i v^{(1)}(x) + b_i(x) \le T_i\left(v^{(1)} - v^{(2)}\right)(x) \text{ if } T_i v^{(1)}(x) + b_i(x) \le 0.$$

 In addition,

$$T_i(v^{(1)} - v^{(2)})(x) = T_i v^{(1)}(x) + b_i(x) - \left(T v^{(2)}(x) + b_i(x)\right) = 0 \text{ if } T_i v^{(1)}(x) + b_i(x) > 0.$$

 Thus,

$$v^{(1)} - v^{(2)} \in X(v, T + b). \tag{A.3}$$

Combining (A.2) and (A.3), and (2.1) as $v = v^{(1)}$, we conclude that

$$v^{(1)} - v^{(2)} = 0.$$

 Conversely, assume that there exists a $v \in L^2(D)^n$ such that

$$\mathrm{Ker}\left(T|_{S(v,T+b)}\right) \cap X(v, T + b) \ne \{0\}.$$

 Then there is $u \ne 0$ such that

$$u \in \mathrm{Ker}\left(T|_{S(v,T+b)}\right) \cap X(v, T + b).$$

 For $i \in S(v, T + b)$, we have by $u \in \mathrm{Ker}(T_i)$,

$$\mathrm{ReLU}\left(T_i(v - u) + b_i(x)\right) = \mathrm{ReLU}\left(T_i v + b_i(x)\right).$$

 For $i \notin S(v, T + b)$, we have by $u \in X(v, T + b)$,

$$\mathrm{ReLU}\left(T_i(v - u)(x) + b_i(x)\right) = \begin{cases} 0 & \text{if } T_i v(x) + b_i(x) \le 0 \\ T_i v(x) + b_i(x) & \text{if } T_i v(x) + b_i(x) > 0 \end{cases}$$
$$= \mathrm{ReLU}\left(T_i v(x) + b_i(x)\right).$$

 Therefore, we conclude that

$$\mathrm{ReLU}\left(T(v - u) + b\right) = \mathrm{ReLU}\left(Tv + b\right),$$

 where $u \ne 0$, that is, $\mathrm{ReLU} \circ (T + b)$ is not injective. $\qquad\square$

# B  Details of Sections 3.1 and 3.2

## B.1  Proof of Lemma 1

466  *Proof.* The restriction operator, $\pi_\ell : L^2(D)^m \to L^2(D)^\ell$ ($\ell < m$), acts as follows,

$$\pi_\ell(a,b) := b, \quad (a,b) \in L^2(D)^{m-\ell} \times L^2(D)^\ell. \tag{B.1}$$

467  Since $L^2(D)$ is a separable Hilbert space, there exists an orthonormal basis $\{\varphi_k\}_{k\in\mathbb{N}}$ in $L^2(D)$. We
468  denote by

$$\varphi_{k,j}^0 := \left( 0, ..., 0, \underbrace{\varphi_k}_{j-th}, 0, ..., 0 \right) \in L^2(D)^m,$$

469  for $k \in \mathbb{N}$ and $j \in [m-\ell]$. Then, $\{\varphi_{k,j}^0\}_{k\in\mathbb{N}, j\in[m-\ell]}$ is an orthonormal sequence in $L^2(D)^m$, and

$$V_0 := L^2(D)^{m-\ell} \times \{0\}^\ell$$
$$= \operatorname{span}\left\{ \varphi_{k,j}^0 \mid k \in \mathbb{N}, \ j \in [m-\ell] \right\}.$$

470  We define, for $\alpha \in (0,1)$,

$$\varphi_{k,j}^\alpha := \left( 0, ..., 0, \underbrace{\sqrt{(1-\alpha)}\varphi_k}_{j-th}, 0, ..., 0, \sqrt{\alpha}\xi_{(k-1)(m-\ell)+j} \right) \in L^2(D)^m, \tag{B.2}$$

471  with $k \in \mathbb{N}$ and $j \in [m-\ell]$. We note that $\{\varphi_{k,j}^\alpha\}_{k\in\mathbb{N}, j\in[m-\ell]}$ is an orthonormal sequence in $L^2(D)^m$.
472  We set

$$V_\alpha := \operatorname{span}\left\{ \varphi_{k,j}^\alpha \mid k \in \mathbb{N}, j \in [m-\ell] \right\}. \tag{B.3}$$

473  It holds for $0 < \alpha < 1/2$ that

$$\left\| P_{V_\alpha^\perp} - P_{V_0^\perp} \right\|_{\mathrm{op}} < 1.$$

474  Indeed, for $u \in L^2(D)^m$ and $0 < \alpha < 1/2$,

$$\left\| P_{V_\alpha^\perp} u - P_{V_0^\perp} u \right\|_{L^2(D)^m}^2 = \| P_{V_\alpha} u - P_{V_0} u \|_{L^2(D)^m}^2$$

$$= \left\| \sum_{k\in\mathbb{N}, j\in[m-\ell]} (u, \varphi_{k,j}^\alpha)\varphi_{k,j}^\alpha - (u, \varphi_{k,j}^0)\varphi_{k,j}^0 \right\|_{L^2(D)^m}^2$$

$$= \left\| \sum_{k\in\mathbb{N}, j\in[m-\ell]} (1-\alpha)(u_j, \varphi_k)\varphi_k - (u_j, \varphi_k)\varphi_k \right\|_{L^2(D)}^2$$

$$+ \left\| \sum_{k\in\mathbb{N}, j\in[m-\ell]} \alpha(u_m, \xi_{(k-1)(m-\ell)+j})\xi_{(k-1)(m-\ell)+j} \right\|_{L^2(D)}^2$$

$$\leq \alpha^2 \sum_{j\in[m-\ell]} \sum_{k\in\mathbb{N}} |(u_j, \varphi_k)|^2 + \alpha^2 \sum_{k\in\mathbb{N}} |(u_m, \xi_k)|^2 \leq 4\alpha^2 \|u\|_{L^2(D)^m}^2,$$

475  which implies that $\left\| P_{V_\alpha^\perp} - P_{V_0^\perp} \right\|_{\mathrm{op}} \leq 2\alpha$.

476  We will show that the operator

$$P_{V_\alpha^\perp} \circ T : L^2(D)^n \to L^2(D)^m,$$

477  is injective. Assuming that for $a, b \in L^2(D)^n$,

$$P_{V_\alpha^\perp} \circ T(a) = P_{V_\alpha^\perp} \circ T(b),$$

478 is equivalent to
$$T(a) - T(b) = P_{V_\alpha}(T(a) - T(b)).$$

479 Denoting by $P_{V_\alpha}(T(a) - T(b)) = \sum_{k \in \mathbb{N}, j \in [m-\ell]} c_{k,j} \varphi_{k,j}^\alpha$,
$$\pi_1(T(a) - T(b)) = \sum_{k \in \mathbb{N}, j \in [m-\ell]} c_{k,j} \xi_{(k-1)(m-\ell)+j}.$$

480 From (3.1), we obtain that $c_{kj} = 0$ for all $k, j$. By injectivity of $T$, we finally get $a = b$.

481 We define $Q_\alpha : L^2(D)^m \to L^2(D)^m$ by
$$Q_\alpha := \left( P_{V_0^\perp} P_{V_\alpha^\perp} + (I - P_{V_0^\perp})(I - P_{V_\alpha^\perp}) \right) \left( I - (P_{V_0^\perp} - P_{V_\alpha^\perp})^2 \right)^{-1/2}.$$

482 By the same argument as in Section I.4.6 Kato [2013], we can show that $Q_\alpha$ is injective and
$$Q_\alpha P_{V_\alpha^\perp} = P_{V_0^\perp} Q_\alpha,$$

483 that is, $Q_\alpha$ maps from $\mathrm{Ran}(P_{V_\alpha^\perp})$ to
$$\mathrm{Ran}(P_{V_0^\perp}) \subset \{0\}^{m-\ell} \times L^2(D)^\ell.$$

484 It follows that
$$\pi_\ell \circ Q_\alpha \circ P_{V_\alpha^\perp} \circ T : L^2(D)^n \to L^2(D)^\ell$$

485 is injective. $\qquad \square$

## B.2 Remarks following Lemma 1

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

'} : \left( \mathrm{span}\{\varphi_k\}_{k \leq N} \right)^{d_{in}} \to \left( \mathrm{span}\{\varphi_k\}_{k \leq N} \right)^{d_{in}} \times \left( \mathrm{span}\{\varphi_k\}_{k \leq N'} \right)^{d_{out}},$$

656 is injective. Furthermore, by the same argument (ii) (construction of $H$) in the proof of Theorem 1,

$$H_{N'} \in NO_{L,N'}(\sigma; D, d_{in}, d_{out}),$$

657 because both of two-layer ReLU and Leaky ReLU neural networks can represent the identity map.
658 Note that above $K_{inj,N}$ is an orthogonal projection, so that $K_{inj,N} \circ \cdots \circ K_{inj,N} = K_{inj,N}$. However,
659 we write above $H_{N'}(a)_1$ as $K_{inj,N} \circ \cdots \circ K_{inj,N}(a)$ so that it can be considered as combination of
660 $(L+2)$ layers of neural networks.

661 We estimate that for $a \in L^2(D)^{d_{in}}$, $\|a\|_{L^2(D)^{d_{in}}} \leq R$,

$$\|H_{N'}(a)\|_{L^2(D)^{d_{in}+d_{out}}} \leq 1 + 4M + \|K_{inj}\|_{\mathrm{op}}^{L+2} R =: C_H.$$

662 Here, we repeat an argument similar to the one in the proof of Lemma 2: $H_{N'} : L^2(D)^{d_{in}} \to$
663 $L^2(D)^{d_{in}+d_{out}}$ has the form of

$$H_{N'}(a) = \left( \sum_{k \leq N} (H_{N'}(a)_1, \varphi_k) \varphi_k, \sum_{k \leq N'} (H_{N'}(a)_2, \varphi_k) \varphi_k \right).$$

664 where $(H_{N'}(a)_1, \varphi_k) \in \mathbb{R}^{d_{in}}$, $(H_{N'}(a)_2, \varphi_k) \in \mathbb{R}^{d_{out}}$. We define $\mathbf{H}_{N'} : \mathbb{R}^{N d_{in}} \to \mathbb{R}^{N d_{in} + N' d_{out}}$
665 by

$$\mathbf{H}_{N'}(\mathbf{a}) := \left[ ((H_{N'}(\mathbf{a})_1, \varphi_k))_{k \in [N]}, ((H_{N'}(\mathbf{a})_2, \varphi_k))_{k \in [N']} \right] \in \mathbb{R}^{N d_{in} + N' d_{out}}, \mathbf{a} \in \mathbb{R}^{N d_{

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

_0}$, $\epsilon_1 > 0$ and $y_\ell \in D$ ($\ell = 1, \ldots, \ell_0$), where $s(\vec{i}, \ell) := i_\ell \epsilon_1$. Here, $F_{z, s, h}(v, w)$ are integral neural operators with distributional kernels

$$F_{z, s, h}(v, w)(x) = \int_D k_{z, s, h}(x, y, v(x), w(y)) dy,$$

where $k_{z, s, h}(x, y, v(x), w(y)) = v(x) \mathbf{1}_{[s - \frac{1}{2}h, s + \frac{1}{2}h]}(w(y)) \delta(y - z)$, $\mathbf{1}_A$ is the indicator function of a set $A$ and $y \mapsto \delta(y - z)$ is the Dirac delta distribution at the point $z \in D$. Using these, we can write the inverse of $F_1$ at $g \in \mathcal{Y}$ as

$$F_1^{-1}(g) = \lim_{m \to \infty} \sum_{\vec{i} \in \mathcal{I}} \Phi_{\vec{i}} \mathcal{H}_{j(\vec{i})}^{\circ m} \begin{pmatrix} v_{j(\vec{i})} \\ g \end{pmatrix}, \qquad (E.6)$$

where $j(\vec{i}) \in \{1, 2, \ldots, J\}$ are suitably chosen and the limit is taken in the norm topology of $H^1(D)$. This result is summarized by the following theorem, a modified version of Theorem 3 where the inverse operator $F_1^{-1}$ in (E.6)