# OpenReview forum: "Globally injective and bijective neural operators"
_NeurIPS.cc/2023/Conference — NeurIPS 2023 poster_

### Official Review · Reviewer_GL1W · 2023-06-21

**Soundness:** 4 excellent
**Presentation:** 3 good
**Contribution:** 2 fair
**Rating:** 4
**Confidence:** 3

**Summary:**

This paper extends several known results on ReLU networks from finite dimensional domains to the more challenging infinite dimensional domains. Namely:
(a) Conditions for injectiviity of infinite dimensional ReLU networks are provided
(b) Universality of infinite dimensional ReLU networks was proven
(c) Extension to network with non-linear kernels are provided


**Strengths:**

Strengths: From a mathematical standpoint the paper proves some natural and  non-trivial theorems

**Weaknesses:**

I think this paper is not appropriate for this venue: it is not clear to me why the ML world should care that much about infinite dimensional ReLU operators, and the authors do not make an effort to explain this. There is certainly room for purely theoretical papers in Neurips and there are many such papers. However, I believe that, even if the proofs are above the paygrade of most Neurips attendants, it should be at least clear to said attendants why the questions are interesting. I do not feel this is the case here

**Questions:**

To convince me, mostly rebuttal should focus on importance of the problems discussed in the paper.

I note some minor comments and typos I saw while reading the paper:
Line 7 `the case the case'
Line 34 'on its face' not sure that is an expression in english
Line 43, 53 114 and throughout `the equivalent condition' should be `an equivalent condition'
Definition 1 should be more explicit IMHO: What are bias functions? What do the linear kernels do? Better to just right out the formula
Line 105 be *the* ReLU activation

**Limitations:**

Yes

---

> ### Author Rebuttal · Authors · 2023-08-09
>
> We would like to thank the reviewer for your comments and careful eye.
>
> _I think this paper is not appropriate for this venue: it is not clear to me why the ML world should care that much about infinite dimensional ReLU operators, and the authors do not make an effort to explain this. There is certainly room for purely theoretical papers in Neurips and there are many such papers. However, I believe that, even if the proofs are above the paygrade of most Neurips attendants, it should be at least clear to said attendants why the questions are interesting. I do not feel this is the case here._
>
> We could have done a better job of valorizing the results, and emphasizing that our analysis goes far beyond the ReLU case. The ReLU activation function is of interest only in section 2.1. We think that our results, properly valorized, are well-suited for NeurIPS not because the paygrade of the proofs, but because the results are practical and important. That said, it is clear from your comment and others, that our paper still needs more motivation. Therefore, we would like to include some more applications of injectivity \& invertibility. For more details, please see the global comment to all reviewers. We hope that including those comments will better motivate the focus on injectivity/invertibility, and address your criticisms.
>
> _I note some minor comments and typos I saw while reading the paper: Line 7 the case the case'_
>
> Thank you for pointing this out. It has been addressed.
>
> _Line 34 'on its face' not sure that is an expression in english Line 43_
>
> We have replaced the phrase 'on its face' with the phrase 'on first inspection.'
>
> _Line 43, 53 114 and throughout the equivalent condition' should be `an equivalent condition.'_
>
> All have been fixed, thank you.
>
> _Definition 1 should be more explicit IMHO: What are bias functions?_
>
> We have made Definition 1 more explicit by adding equation definition of $T_\ell$, .
>
> Bias functions are analogous to bias vectors. We believe that their meaning will become clear by defining $T_\ell$ more explicitly.
>
> _What do the linear kernels do?_
>
> A linear kernel is a kernel in the sense of a convolution. That is, they are the filter of a convolution. That is, an operator of the form $v \mapsto \int_\Omega k(x - y) v(y) dy$, where $x$ is the independent variable. A nonlinear kernel is an operator of the form $v \mapsto \int_\Omega k(x,y,v(x),v(y))v(y) dy$ where again $x$ is independent. Note that this map is no longer linear in $v$.
>
> _Better to just right out the formula Line 105 be the ReLU activation._
>
> You are right, that is more clear. This change has been made.

---

> > ### Comment · Reviewer_GL1W · 2023-08-13
> >
> > Thanks for your answers. I still think Neurips is not the right venue for this paper. I would sent it to e.g., Acta Numerica... I read the motivation you provided in the main remark and the concepts discussed there are still very vague and abstract to me.
> > I believe 99% of Neurips audience will feel the same way.

---

### Official Review · Reviewer_ZQ9v · 2023-07-04

**Soundness:** 3 good
**Presentation:** 1 poor
**Contribution:** 3 good
**Rating:** 5
**Confidence:** 4

**Summary:**

The paper considers the question of when neural operators, which have infinite-dimensional inputs and outputs, are injective and bijective. This question is answered in quite some generality in different settings and under different assumptions.

**Strengths:**

--- A careful and at times, deep, analysis for the questions under consideration is provided.

**Weaknesses:**

1. Relevance and Scope: While appreciating the depth of the functional analysis that is presented, this reviewer is left a bit perplexed about the rationale behind all this heavy machinery. The authors do not really motivate why one needs bijective neural operators in the first place. There is some boilerplate on generative models in the discussion but this does not occupy any centerstage. If the fundamental question is itself not posed properly, the interest of the subsequent analysis becomes rather limited. The authors should clearly motivate why they consider these questions in the first place and explain it well to the reader.

2. Clarity: The paper has excessively abstract notation and presentation. It needs to be reorganized to bring out the main contributions well. Here are some specific questions about the clarity:


a) Why introduce Directed spanning sets in Definition 1 (which is almost impossible to follow as no intuition is provided into what it is) and Proposition 1, when immediately afterwards you have Proposition 2 which provides the answer for a bijective activation function. Who cares about ReLU when you already have a result on Leaky ReLU which is heavily used to begin with.

  b) Similarly what is the use of the local layer wise analysis when you anyway consider the global end-to-end neural operator in Section 3 ?

  c) In Lemma 1, what does condition 3.1 even mean in practice ? How can one check it ?

  d) The authors have to concede that having a universal approximation theorem does not mean much, see Lanthaler et al and Kovachki et al 2021a where this issue of universal approximation is critiqued as the system size can grow (even double) exponentially. So having a universal approximation theorem does not imply any kind of efficient approximation but it is a necessary condition at best.

  e) Does section 3.3 imply that FNO and WNO are injective and bijective neural operators ? If not, why not ? If yes, under what conditions ?

  f) Can you find a non-trivial example for section 4.1, apart from Example 1 which appears to be trivial. Has anyone ever implemented example 1? If so, what is its empirical performance.

  g) What is the rationale for section 4.2, why would I want to explicitly compute the inverse of the neural operator -- in what situations is that useful ? Can you give an example ?

All the questions clearly indicate that the paper is not well-written and needs a substantial rewrite.

3. Concreteness: The main limitation in my view is that lack of exemplification of the results, except for example 1 -- the authors should give many more examples of practical utility, with conditions on the weight matrices, kernels as well as activation functions so that either available or de novo neural operators fall into their theoretical framework. This will enable the readers to better understand the significance of this work.

4. Novelty: In many places in the paper, the authors refer to Puthawala et al 2022 -- how does the current paper differ from this reference ? A thorough elaboration of these differences is essential for judging the novelty of this paper.

5. Finite-dimensional representations and Aliasing: The authors always assume that one has access to functions as both inputs and outputs. This is far from the case and in practice, one has to work with finite-dimensional representations of the functions, see for instance Fanaskov and Oseledets, Spectral neural operators, 2022 for a discussion on this issue. This use of finite-dimensional representation can lead to what are called aliasing errors. For instance, FNO has aliasing errors -- how do these errors affect your theoretical considerations -- in particular, does aliasing destroy the bijection property of the neural operator. Addressing this issue is crucial for any relevance for the theory, particularly in the construction of inverse neural operators

**Questions:**

Quite a few questions were already asked in the section on Weaknesses. The authors should address them.

**Limitations:**

--- No solid rationale is provided for the whole premise of the paper.

--- A lot of abstract theory with few examples, if any. Practical utility of the
concept is totally unclear from the current version.

--- Unclear if the framework survives contact with reality in the form of finite-dimensional
representation of functions (for instance sampling on a uniform grid) that can lead to
aliasing errors.

---

> ### Author Rebuttal · Authors · 2023-08-09
>
> We would like to thank the reviewer for your comments, careful eye, and fair criticisms.
>
> _Relevance and Scope_
>
> We should have motivated the application of injectivity and invertibility more. We will, please see the global comment.
>
> _Why introduce...begin with._
>
> You make a good point. The for these two cases back-to-back is to use the ReLU activation to 'build a bridge' from the Euclidean case to infinite dimensions, and draw contrasts between them. We did not do a good enough job making this parallel clear, and so we will include a sentence at line 92 saying the following.
>
> "Deriving a condition for layer-wise injectivity with bijective activation functions is trivial in a finite dimensional setting. With a ReLU activation function it requires the so-called Directed Spanning Set (DSS) condition. This condition is not automatic, but will hold with high probability for random weight matrices if they are expansive enough.
>
> In this section we derive a generalization of the DSS condition and show that it is much more restrictive than the infinite dimensional setting. We then present a less restrictive condition that is met when the activation function is bijective, e.g. a leaky-ReLU activation is used."
>
> _Similarly what...Section 3?_
>
> Although it would appear that the end-to-end condition supersedes the layer-wise one, it is not the case. The layer-wise case is less restrictive and has different applications & implications then global result does not. We didn't do enough to make this apparent in the manuscript. Therefore, we would like to add the following sentences just after Line 97.
>
> "Although it may appear that the end-to-end result is strictly stronger than the layerwise result, this is not the case. The layerwise result is an exact characterization, whereas the end-to-end result is sufficient for injectivity, but not necessary. The layerwise analysis is also constructive, and so gives a rough guide for the construction of injective networks, whereas the global analysis is less so. Finally, the layerwise condition has different applications, such as network of stochastic depth, see e.g. [Huang et al. 2016] or [Benitez et al. 2023]. End-to-end injectivity by enforcing layerwise injectivity is straight forward, whereas deriving a sufficient condition for any depth is more daunting."
>
> _In Lemma...check it ?_
>
> To make condition 3.1 clearer, and to provide a 'hands on' example, we would like to include the following  Condition 3.1 has a straight-forward interpretation. We think that this example provides a reprieve from the abstraction, and gives a simpler takeaway.
>
> "Lemma 1 and Eqn. 3.1 may be interpreted as saying that if _some_ orthonormal sequence $\\{\xi_k\\}\_{k \in \mathbb{N}}$ exists that doesn't overlap the range of $T$, then $T$ may be embedded in a small space without losing injectivity. As an example of such a $\\{\xi_k\\}\_{k \in \mathbb{N}}$, if the range $\mathrm{Ran}(T)$ of $T$ is included in the continuous function space (for instance, a finite rank neural operator with continuous basis like FNOs), then we may choose $\\{\xi_k\\}\_{k \in \mathbb{N}}$ to be a discontinuous basis and the condition is automatically met."
>
> _The authors...at best._
>
> Indeed, having a universal approximation theorem is only a starting point for showing efficient approximation but, we argue, even such a starting point wasn't achieved until recently in the finite-dimensional case. The proof of our universal approximation result is compatible with future efficient (quantitative) approximation results that may arise for neural operators. To make this point clear, we would like to include the following sentences just after Line 185.
>
> "The proof for universal approximation theorem is constructive. If, in the future, efficient approximation bounds for neural operators are given, such bounds can likely be used directly in our universality proof to generate corresponding efficient approximation bounds for injective neural operators."
>
> _Does section...what conditions?_
>
> In short, it does give such conditions. We have (abstract) conditions that apply perfectly well to FNO or WNO. It all depends on the choice of basis. It is clear that we did not do a good enough job of drawing attention to this in the main text. Therefore we would like to modify lines 218 - 221 to say the following.
>
> "We show that Propositions 1, 2 (characterization of layerwise injectivity/bijectivity), and Lemma 1 (global injectivity) all have natural analogues for finite rank operator $K_{\ell,N}$ in Proposition 6 and Lemma 3 in Appendix C. These conditions applies out-of-the-box to both Fourier Neural Operators (FNO) and Wavelet Neural Operators (WNO). We also show the universal approximation in the case of finite rank approximation."
>
> _Can you...empirical performance._
>
> Motivated by your comment, we derived another four examples that illustrate non-triviality of the results in Section 4.1. The entire derivation and proof is worked out, but is too long to include in this rebuttal. Please see the global comment for the setup of these examples.
>
> _What is...an example?_
>
> There are two main rationales for Section 4.2, some abstract involving the algebra of groups and other involves justifying the discussion of operator encoder-type networks. We feel we didn't do a good enough job of bring them to the fore. Therefore, we would like to add the following sentences just after Line 294.
>
> "The proof that neural operators may be inverted with other neural operators provides a theoretical justification for Integral Auto Encoder networks (IAE-nets, Ong et al. 2022) where an encoder/decoder pair that parallel the roles of the finite-dimensional VAEs (Kingma & Welling 2023). This section proves that the decoder half of a IAE-net is provably able to inverse the encoder half. Our analysis also shows that injective differential operators (as arise in PDE) and integral operator encoders form a formal algebra under operator composition."

---

> > ### Comment · Reviewer_ZQ9v · 2023-08-17
> > **Reply to the authors' rebuttal**
> >
> > I start by thanking the authors for their rebuttal and apologize for the delay in responding. The authors have attempted to address several of my comments yet many concerns still remain. I outline them below:
> >
> > 1. Now, the authors motivate the rationale in terms of pseudo differential operators (PDOs) and claim that that their construction is an analogue of this very important concept to neural operators. I respectfully disagree. PDOs are a very general theory for linear PDEs whereas your construction presumably is limited to (possibly) nonlinear neural operators. I don't see the connection about why PDOs are a motivation and why they were not introduced in the first place. In particular, are you claiming that your invertible NOs are a natural framework for solving linear PDEs -- if so, where is the evidence ? Again, i don't see a direct relation with Bayesian inverse problems (BIP). Are you saying that your construction will directly solve BIPs ? I presume that any solution of BIPs will involve some form of sampling (either MCMC or normalizing flows or diffusion models) and all that is needed is a surrogate of the forward operator -- this surrogate need not be invertible at all -- so I am still not convinced that there is any practical justification behind your construction.
> >
> > 2. The rewriting that you promise for a camera ready version might improve the overall presentation.
> >
> > 3. I am still not able to understand how your results apply to FNO and VNO *out of the box* ? Does it mean that FNO is invertible under all conditions -- this is impossible to believe as FNO works very well for problems such as diffusion equation where there is no invertibility whatsoever. Please clarify my genuine concern here. Also, what do you mean by a finite-rank operator here -- do you claim that FNO is finite-rank which it is patently not -- I am confused here.
> >
> > 4. Your 4 examples A-D leave me confused. Does A mean that this is the class for which you have infectivity ? B is just a linear operator -- having subjectivity is of little utility for it. C and D are so succinct that it is impossible for me to evaluate them.
> >
> > 5. You did not answer my question about how exactly does this paper differ from and what is its novelty vis a vis Puthawala et al 2022 ?
> >
> > 6. You did not answer my question about what happens to your constructions about injective and bijective operators when the data itself is only available as point samples on a grid, as it is in practice ?
> >
> > 7. What exactly do you justify about IAEnets ? If its decoder is exactly able to invert the encoder -- so what ? The universal approximation theorem (as well as quantitative error bounds) should hold even if they decoder only approximately inverts the encoder -- see constructions in the paper of Lanthaler, Mishra and Karniadakis on DeepONets where such approximation suffices.
> >
> > Summary: The authors have not convinced this reviewer of the concrete utility of their construction for any practical applications of operator learning. I am looking forward to your replies and apologize again for the delay in responding to your rebuttal.

---

> > > ### Author Response · Authors · 2023-08-19
> > >
> > > Thank you for taking time to read and consider our rebuttal. It is clear that you care deeply about improving our work, and we appreciate this. We only wish we had more than 5000 characters to answer your questions.
> > >
> > > Our paper is foundational, theoretical, and we feel it provides useful results for futuer more applied work. We believe that NeurIPS is the right venue for these kinds of papers. Consider the additional literature suggested by reviewer kfSM. We hope that, like those suggested works, you can agree that our work is rigorous, provides deep insights on an important architecture, and so will likely be well-received at NeurIPS.
> > >
> > > 1. We are talking about non-linear operators and mappings, and our prime applications are in solving inverse problems. That said, the framework also applied to solving differential equations. To make the connection more concrete, we have an example worked out.
> > > $$
> > > -D_x^2u(x)+su(x)+p(u(x))=f(x)
> > > $$
> > > where $x\in [0,1]$, $u(0)=u(1)=0$, $p(u)$ is a non-linear term, $f$ is a source, and $s>0$. There is a linear integral operator $G:L^2(0,1)\to L^2(0,1)$,
> > > $$
> > > Gf(x)=\int_0^1 k(x,y)f(y)dy,
> > > $$
> > > for which $v=Gf$ solves the linear equation
> > > $$
> > > -D_x^2v(x)+sv(x)=f(x)
> > > $$
> > > with $u(0)=u(1)=0$. Here, $k$ is Green's function. The equation for $u$ may be rewritten as
> > > $$
> > > u=N(u)
> > > $$
> > > where
> > > $$
> > > N(u)=N_{s,p,f}(u)=-G(p(u)+f).
> > > $$
> > > When $s$ is large enough, this non-linear integral operator form can be solved using a fixed point iteration $u_k=N(u_{k-1})$, $u_0=f$. By unrolling the iteration, the result of the $k$:th iteration, $N^k(f)$, can be approximated by a NO. Above, the integral operator $G$ is a pseudodifferential operator which solves the linear equation for $v$ whereas $N^k$ is a NO which gives an approximate solution of the non-linear equation for $u$.
> > > We are also interested in inverse problems for PDE of determining the coefficient functions when one is given observations of the (boundary values of) solutions $u_i$ corresponding to various sources $f_i$, $i\le m$. There, the operation
> > > $$
> > > p\to(N_{s,p,f_1})^k(f_1),...,(N_{s,p,f_m})^k(f_m)
> > > $$
> > > is a NO. If this NO is injective, the non-linear function $p$ can be determined when the solutions $u_i$ are observed.
> > >
> > > 2. We think so too.
> > >
> > > 3.
> > > - "Does it...all conditions." If each of an FNO satisfies the conditions in Prop. 5, then yes.
> > > - "this is...invertibility whatsoever." Theorems 1 and 2 are universal approximation theorems that *any* continuous operator (not only injective ones) can be approximated by an injective neural operator. This means that even diffusion problems, those without invertibility, may be approximated arbitrarily well.
> > > - "what do...patently not." A FNO is not a finite rank operator. What we mean by finite-rank neural operator, that a neural operator whose (non-local) integral operator for each layer is represented by finite truncated basis expansion. If it is a Fourier basis, it is an actual FNO.
> > >
> > > 4. We apologize, the rebuttal had tight space constraints. Our machinery shows that problems of the form of A are surjective. For B, it is not linear, note that the term $K$ is nonlinear in $u$. For C, the point is that coercivity is a broadly useful property. See, e.g. Li, Schwab, Antholzer et Haltmeier Inverse Problems 36, 2020. In this setting, a regularization operator is learned that must be coercive, see condition 2.2.c.. For D, the details are involved but briefly, in quantum mechanics, some phenomena may be modeled as nonlinear (or non-physical in the case of negative energy) perturbations to physical systems. Our coercivity analysis `does not care' about such perturbations.
> > >
> > > 5. The results of Puthawala et al. 2022 only apply in the finite-dimensional Euclidian setting and that there. Further, surjectivity becomes nontrivial in the infinite-dimensional case. Notions like bi/injectivity, or closedness under inverses don't transfer from the Sobolev to Euclidian spaces. The current work also considers the non-Relu activation case much more.
> > >
> > > 6. Please see our response to reviewer KyUH, Line 215. Bijectivity and/or injectivity holds when the function spaces are suitably approximated by finite dimensional spaces using a finite set of basis vectors. One can apply the similar techniques that are used in Finite Element Method (FEM) analysis where the weak formulation of a PDE becomes a matrix equation. The infinite dimensional theory advice in choosing a suitable basis and gives error estimates for such approximations.
> > >
> > > 7. Please allow us to draw a parallel between a similar work. There is great interest in applying DL to fluid dynamics, including divergence-free fluid flow, and a bevy of DL models that are universal approximators. It is necessary to design a neural network that provably models divergence-free flow? No. But, designing such a network is natural https://arxiv.org/abs/2210.01741. Encoding injectivity directly into a neural network is natural in applications when encoding a signal, or in modeling invertible processes.

---

> > > > ### Comment · Reviewer_ZQ9v · 2023-08-19
> > > > **Reply to the authors' response**
> > > >
> > > > I thank the authors for replying to my comments. At the outset, in my understanding, you can use multiple threads to reply to comments -- so in practice, there is no 5000 character limit. That being said, your reply has assuaged some of my concerns but a few still remain and I outline them below:
> > > >
> > > > 1. Thanks for your worked out example -- unfortunately it is not convincing in your context as certainly the nonlinear operator N need not be bijective for the fixed point argument to work. Any NO (as long as it is a universal approximator will be fine here). However, I fully buy the argument that your ideas are important for inverse problems -- please use them as your main motivation and examples in a CRV.
> > > >
> > > > 3. So, the bijectivity of FNO boils down to proposition 5 in the appendix. This is the crux of my issue with your approach. Let me explain why: as it is written, how proposition 5 can be verified in practice for a concrete architecture, say FNO. Please note that FNO cannot be written in terms of underlying Fourier basis in general as the nonlinearity (and residual terms) give it a proper nonlinear structure (See  Lanthaler et al,
> > > > Nonlinear Reconstruction for Operator Learning of PDEs with Discontinuities.  In 11th International Conference on Learning Representations (ICLR) (2023)). Nevertheless, how exactly does proposition 5 translate to conditions on the weights and biases of FNO, both at initialization and how does one maintain it during training -- this is the core of my criticism about your abstract results -- how does one realize them in practice ?
> > > >
> > > > 6. I am not convinced about your explanation here. First, you cannot reduce Neural operators to matrix equations due to nonlinearities. Second, successful NOs like FNO are precisely non-linear in the sense of Lanthaler et al,
> > > > Nonlinear Reconstruction for Operator Learning of PDEs with Discontinuities.  In 11th International Conference on Learning Representations (ICLR) (2023) and it is their nonlinearity which explains their success in many examples. So what exactly would be a suitable basis for FNO ? Finally, data is given in terms of point wise samples, which implies automatically a sinc-basis as a natural one. I am curious about how theory deals with this case.
> > > >
> > > > Summarizing, I certainly find your ideas interesting and the math solid and deep but I am not yet convinced that a case has been made about its practical utility. I am happy to revisit my assessment once you have clarified my remaining concerns.

---

> > > > > ### Author Response · Authors · 2023-08-20
> > > > > **Reply Part 1**
> > > > >
> > > > > 1. Inverse problems are indeed a critical application, and we will articulate this further in the CRV. Nevertheless, think there are other important applications of injectivity and surjectivity in parallel as well. Revisiting the example from our previous comment, if I - N is non-surjective then nonlinear ODE may not have a solution. This may happen even in the linear case. If $s$ is eigenvalue of $-D_x^2$ with Dirichlet boundary conditions, then $G$ if is neither injective nor surjective and if $f$ is not orthogonal to the corresponding eigenfunction, then the problem has no solutions. Similar phenomena appear for non-linear equations. In this way, injectivity & surjectivity of PDO (and by extension NOs) are of key practical interest.
> > > > >
> > > > > 2. We apologize, we made a typo in our previous comment. The entire FNO need not satisfy prop 5. If each __layer__ of the FNO satisfies the conditions in Prop. 5, then injectivity of the entire network follows. We agree, FNO are nonlinear, and this fact is of great consequence as shown by the Lanthaler et al. paper. Using Eqn. 2.6 from that paper, if we write the $W_\ell \cdot v^\ell(x)$ term as an integral transform using a Dirac function, the terms $W_\ell \cdot v^\ell(x) + K_\ell v^\ell$ can be combined into an integral transform with one kernel. Without truncation the Fourier basis, injectivity of the FNO follows when Proposition 1 (from the current manuscript) is met. If any truncation is done, the $\mathcal N^{\textrm{FNO}}$ will not be injective. This is a consequence of Corollary 6 from Puthawala et al. 2022 Globally Injective ReLU Networks, Journal of Machine Learning Research 23 (2022) 1-55..
> > > > >
> > > > > In general, for the FNO network and beyond, we've worked out three approaches to realize these conditions in practice.
> > > > >
> > > > > - If the activation function is bijective, e.g., a leaky relu, then, roughly, the condition relaxes to requiring that the weight matrix $C_N$, defined in Eqn. C.3 is injective. This is the content of Proposition 5.ii. The weight matrices may be regularized to avoid having small singular values, for example. Square matrices are generically full-rank, and so any square weight matrix (used in training or in a fully trained network) likely meets the condition of Proposition 5.ii without any intervention, or else may be perturbed by an arbitrarily small amount to do so.
> > > > >
> > > > > - If one is committed to using an activation function like relu, is to follow the 'recipe' laid out in Theorem 2 and Lemma 2. See that the proof of Theorem 2 is constructive, and its construction may be followed to make an arbitrarily good injective approximator. We plan on including a remark in the CMV to this effect just after the statement of Theorem 1, per reviewer GKcz's suggestion. See our rebuttal to that reviewer for more details.
> > > > >
> > > > > - If the activation function is bijection, we have prepared another detailed example below. Consider an integral operator of the form $I+K\\colon L^2(D)\\to L^2(D)$ where $Ku(x)\\coloneqq\\int\_D k(x,y)u(y)dy$. If $I + K^{(j)}$ has kernel $k\_j(x,y)$, then
> > > > > $$
> > > > > A = (I + K^{(m)})\\dots(I + K^{(1)})\\colon L^2(D)\\to L^2(D)
> > > > > $$
> > > > > is an integral operator of the form $I+K$. The question is when we may approximate operators $K$ and $K^{(j)}$ using finite dimensional basis as in the prior comment using the projection $P\_n$ where $K\_n \\coloneqq I + P\_n K^{(j)}P\_n \\colon \\mathrm{Ran}(P\_n) \\to \\mathrm{Ran}(P\_n)$ admits a matrix representation. From standard results [Example 7.3, Lee 2013, Introduction to Smooth Manifold] the set of invertible matrices $GL(n,\\mathbb R) \\subset \\mathbb R^{n \\times n}$ is an open set with two path connected components, matrices with positive and negative determinants. Let us notate these two components as $GL^+(n, \\mathbb R)$ and $GL^-(n, \\mathbb R)$ respectively. The strategy is to write an operator $I + K\_n \\in GL^+(n, \\mathbb R)$ as the product of perturbations of the identity that are each invertible. By path connectedness, given such a $I + K\_n$, there are $K^{(j)}\_n \\in GL^+(n,\\mathbb R)$ for $j = 1,\\dots,J$ such that for all $\\|K^{(j)}\_n\\|< \\epsilon\\leq\\frac 1{n^2}$ and so that
> > > > > $$
> > > > > I+K\_n=\\prod^J\_{j = 1}(I+K\_n^{(j)}).
> > > > > $$
> > > > > The point here is that by requiring that $K\_n^{(j)}$ is small enough during training time (by checking, e.g., the Frobenius norm), $I+K\_n^{(j)}$ is guaranteed to be invertible, and so too must be $I + K\_n$. Hence all discretized neural operators of the form $F \\colon \\mathbb R^n \\to \\mathbb R^n$ $F(u) = \\sigma ((I+K\_n)u)$ may be written as
> > > > > $$
> > > > > F(u)=\\sigma(\\prod^J\_{j = 1}(I+K\_n^{(j)})u)
> > > > > $$
> > > > > where $\\|K\_n^{(j)}\\| < \\epsilon$. We may, of course, do a similar construction to represent matrices in $GL^-(n, \\mathbb R)$ likewise as the product of perturbations from the identity. This argument may also be generalized to operators of the form $F(u) = \\sigma((W + K\_n)u)$ for $u \\in \\mathbb R^n$ where $W$ is an invertible matrix.

---

> > > > > > ### Author Response · Authors · 2023-08-20
> > > > > > **Reply Part 2**
> > > > > >
> > > > > > 3. Our explanation was unclear. We do not argue that the entire FNO is a linear operator or may be profitably modeled by a linear operator. We intended to make a different point with our comment, about what basis functions should be chosen for the end member (first and last) layers of the FNO. To be clear, we interpret this question as not one about FNO specifically, but rather a general question about realizing pointwise data as coefficients of a partial basis expansion in some (unknown) basis. In FEM, given a weak formulation and choice of cells (i.e. the mesh) on which the solution is to be obtained, you may choose a set of basis functions (commonly polynomial piecewise functions with compact support) so that the evaluation of the sum of the basis at a data point is its value. This lets you transition from Sobolev space (where you apply the weak formulation) to Euclidian space (where numerical algorithms are run) and vice versa. A sinc basis would be a natural choice for an FNO where the data is regularly sampled. With irregular data a wavelet basis with compact (space) support would be a natural choice. Choosing a wavelet basis is one the key ideas behind Wavelet Neural Operators (WNO), see e.g. Tripura et Chakraborty, Wavelet neural operator for solving parametric partial differential equations in computational mechanics problems 2023.
> > > > > >
> > > > > > The question of what the best basis to choose in terms of efficiency of representation beyond the question of injectivity/surjectivity is quite deep, even when considering just one layer of a neural operator. If $I+K\\colon L^2(d) \\to L^2(D)$ (one layer of a neural operator) is an injective Fredholm operator with index zero, then $(I+K)^{-1} \\colon L^2(D) \\to L^2(D)$ is bounded. If $P_n \\colon L^2(D) \\to L^2(D)$ is a finite-dimensional projector to the span of $(v_1,v_2,\\dots,v_n)$ where $(v_j)$ is an orthogonal basis of $L^2(D)$ then $I + P_nKP_n$ is invertible for an $n$ large enough, and $\\|P_nKP_n-K\\|\_{L^2(D)\\to L^2(D)}\\to 0$ as $n \\to \\infty$. How large $n$ needs to be has to do with how 'good' a basis is for representing $I + K$, specifically how quickly $\\|P_nKP_n-K\\|\_{L^2(D)\\to L^2(D)}\\to 0$. If the kernel $K$ is singular only at $x = y$ and $K$ is a pseudodifferential operator, then wavelets are an efficient choice of basis. See the work Dahmen, W., Prössdorf, S., and Schneider, R.. "Wavelet approximation methods for pseudodifferential equations: I. Stability and convergence..". In the context of ML, $K$ is, of course, being learned. Hence some modifications must be made to allow $K$ to be learned simultaneously with the basis expansion.

---

> > > > > > > ### Comment · Reviewer_ZQ9v · 2023-08-21
> > > > > > > **Reply to the authors**
> > > > > > >
> > > > > > > At the outset, I would like to thank the authors for putting in the extra effort to address my concerns. Here are my follow-up comments.
> > > > > > >
> > > > > > > 1. Thanks for the clarification about FNO -- I am still not convinced. The weight matrices in FNO are not necessarily square matrices -- in fact, key ingredients are the lifting and projection operators (first and last layers) which embed the inputs into a much higher-dimensional latent space and project back the hidden output to the required output space. Given the fact that these matrices are potentially very ill-conditioned and far from being square matrices, your arguments don't seem plausible. it would be very interesting to see some experimental evidence in a CRV about a concrete (numerical) example with FNO where the authors investigate how the initial and trained FNOs satisfy their conditions -- perhaps, gradient descent training automatically leads to better conditioning in this regard and it would be nice to see some evidence of it.
> > > > > > >
> > > > > > > 2. I think the authors missed my point about the link between infinite dimensions and finite-dimensional representations of neural operators. Please see Bartolucci et al Arxiv:2305.19913v1 which highlights the issue of aliasing to claim that this transition between infinite and finite dimensions for neural operators can suffer from aliasing errors generically, both FNO and WNO have this issue, where traditional numerical methods such as FEM don't. So you cannot apply the infinite-dimensional arguments in toto and should discuss this as a possible limitation of your method on how the aliasing error will affect your theorems.
> > > > > > >
> > > > > > > Summary: Given that the discussion period is drawing to a close, this reviewer maintains the overall impression of the article that the underlying math is correct and deep but the link to practical applications is tenuous and the presentation is poor. The authors are well-advised to motivate their rationale in terms of inverse problems, rewrite the CRV to highlight their main results better and include worked-out examples and also discuss limitations of the method. Given these considerations, I raise my score to acceptance of the paper with the above caveats and thank the authors for engaging in this detailed discussion.

---

### Official Review · Reviewer_KyUH · 2023-07-06

**Soundness:** 3 good
**Presentation:** 3 good
**Contribution:** 4 excellent
**Rating:** 8
**Confidence:** 3

**Summary:**

The authors present theoretical results in the field of operator learning, specifically dealing with operators are injective and surjective. They build on existing finite-dimensional work and consider the infinite-dimensional case of learning mappings between infinite-rank Sobolev spaces. The paper contributes several theoretical results in this context, including characterizing the conditions for layer-wise injectivity and bijectivity given certain activation functions, universal approximation results for injective linear neural operators, and sufficient conditions for surjective/bijective nonlinear integral operators. The paper lays the groundwork for analysis of learning injective and bijective operators in the infinite-dimensional setting.

**Strengths:**

The paper has several significant theoretical results and answers many natural questions one would have about the theory of injectivity and bijectivity in neural operators. Examples and practical implementation results in finite-rank cases are also described in the paper and appendix, which is helpful and grounds the theoretical results in practice. The writing is very clear and the proofs are clean. Overall, it is a very thorough paper.

**Weaknesses:**

Just a note on writing, too many proofs and examples are black-boxed and put in the appendix. At least a description of various proof techniques in the main text would be helpful. The paper would also benefit from more motivation of injective/bijective models in downstream tasks/generative modeling, describing any particular successes or failures different activation functions have had, etc. More discussion on how this theory should guide practice would also improve the significance of the paper.

**Questions:**

Line 107: Is it possible to give a more intuitive description of Definition 2 and why it is called a directed spanning set?

Line 234: Not sure why it is a natural conclusion from using the sup norm that the approximation does not smooth out non-smooth operators. Can you clarify this comment?

Line 215: This is a very interesting remark, can you imagine other orthonormal bases and discuss whether they would be worth formulating as a neural operator architecture?

Line 249: What does it mean for integral transform with kernel the attention mechanism in a transformer to have be injective, practically/in a particular problem?

**Limitations:**

This theoretical work naturally pertains to tasks where injectivity and bijectivity are desirable, however neural operators are of course used in much broader contexts. As the authors discuss, their contribution is largely theoretical and an important next step for this work would be explicit constructions of injective neural operators/inverses that are good approximators.

---

> ### Author Rebuttal · Authors · 2023-08-09
>
> Thank you for your careful review, suggestions, and strong endorsement. Please find answers to your questions and feedback below.
>
> _Just a note on writing, too many proofs and examples are black-boxed and put in the appendix. At least a description of various proof techniques in the main text would be helpful._
>
> We find it difficult to balance including sufficient 'teasers' for the proofs of the main results against other page-length considerations. We realized, in light of your comment, that the proof of Theorem 2 lacked intuition. We would therefore add a few sentences giving some intuition about the main proof ideas just after line 231.
>
> _The paper would also benefit from more motivation of injective/bijective models in downstream tasks/generative modeling, describing any particular successes or failures different activation functions have had, etc. More discussion on how this theory should guide practice would also improve the significance of the paper._
>
> This point is well taken. We could have done a better job of more motivation and application discussion. In our global comment at the top, for some more discussion that we'd like to include.
>
> _Line 107: Is it possible to give a more intuitive description of Definition 2 and why it is called a directed spanning set?_
>
> Yes, it is possible. To make the definition more digestible, and explain the notes, we would like to add the following text just after line 110, between Def. 2 and Prop. 2. This addition would give some intuition for the $\ker(T|_{S(v,T+b)})$ term of Eqn. 2.1.
>
> "The name directed spanning set arises from the kernel term of Eqn. (2.1). The indices of $S(v,T + b)$ are those that are directed (positive) in the direction of $v$. If $T$ restricted to these indices together span $L^2(D)^n$, then the kernel term is $\{0\}$, and the condition automatically satisfied. Hence, The DSS conditions measures the extend to which the set of indices which are directed w.r.t. form a span of the input space of L."
>
> _Line 234: Not sure why it is a natural conclusion from using the sup norm that the approximation does not smooth out non-smooth operators. Can you clarify this comment?_
>
> The issue of smoothening out nonsmooth operators was a feature of some prior work that is not present in our work, but this point wasn't clearly made in Remark 1. We would like to modify the content of Remark 1 to make this point more clear. Please see our proposed new text for Remark 1 below.
>
> "Observe that in our finite-rank approximation result, we only require that the target function G+ is continuous and bounded, but not smooth. This differs from prior work that requires smoothness of the function to be approximated."
>
> _Line 215: This is a very interesting remark, can you imagine other orthonormal bases and discuss whether they would be worth formulating as a neural operator architecture?_
>
> We're glad that you found it interesting! Your follow up question is interesting as well. We can't think of any other NO architecture that can be formed by a particular choice of basis. We would, however, like to include the following remark just after line 217 to make this point more clear.
>
> "Lemma 2 and Remark 3 in the appendix C.3 give a 'recipe' to construct the projection $B$ such that the composition $B \circ T$ (interpreted as augmenting finite rank neural operator $T$ with one layer $B$) is injective.
> The projection $B$ is constructed by using an orthogonal sequence $\{\xi_k\}_k$ subject to the condition (3.1), which that does not over-leap the range of $T$.
> This condition is automatically satisfied for any orthogonal base $\{\varphi_k\}_k$.
> That is, if we find an orthogonal sequence $\{\xi_k\}_k$ that is 'easy' to compute, then, we can construct 'easy' projection $B$.
> This could yield practical implications in guiding the choice of the orthogonal basis ${\varphi_k}_k$ for the neural operator's design."
>
> _Line 249: What does it mean for integral transform with kernel the attention mechanism in a transformer to have be injective, practically/in a particular problem?_
>
> Appealing to the analogy that the attention mechanism allows a layer to 'focus' on particular pieces of a signal, an injective attention mechanism could be considered one that `allows focus, but not blindspots.' Here, a 'blindspot' means a (complete) loss of information.

---

### Official Review · Reviewer_GKdz · 2023-07-07

**Soundness:** 4 excellent
**Presentation:** 3 good
**Contribution:** 3 good
**Rating:** 6
**Confidence:** 2

**Summary:**

This paper provides a theoretical analysis of the injectivity and surjectivity of neural operators.

Section 2 discusses the injectivity of a single layer of neural operators. For the ReLU activation, the injectivity of the layer is characterized by the directed spanning set. On the other hand, if the activation is bijective, the injectivity of the layer is equivalent to the injectivity of its constituent.

Section 3 of the paper examines how injectivity can be preserved when composing layers. The paper shows that integral neural operators with L^2-integral kernels possess a universal approximation property for continuous operators. A truncated series expansion-based approximation, implementable for this family, is demonstrated to maintain the universal approximation property.

Section 4 considers nonlinear integral operators and presents a sufficient condition for a layer in this class to be bijective. Additionally, this document proposes a method for constructing the inverse of a nonlinear integral neural operator layer. The inverse is expressed as a limit of integral neural operators.

**Strengths:**

[originality]

- As far as I understand, this paper is the first to provide a rigorous framework for analyzing the injectivity and bijectivity of neural operators.

[quality]

- The paper is written with mathematical rigor, and there are no noticeable flaws in the logical development. However, I have not been able to check the proofs in detail.

[clarity]

- The sections have clear purposes, with concise explanations of the goals in each section.

[significance]

- This paper presents universality results and identifies conditions under which injectivity or bijectivity can be guaranteed. These theoretical foundations can improve the models and facilitate their application by increasing their reliability and enhancing understanding.

**Weaknesses:**

[originality]

- None in particular

[quality]

- The paper contains a noticeable number of grammatical and typographical errors. It is recommended that an up-to-date automatic checker is used to revise the paper and correct these issues.
- The model classes, including layers, can be assigned specific symbols to make them stand out in the paper.
- Line 33 mentions "infinite-rank Sobolev spaces," but I could not find the definition of the notion of rank of a Sobolev space in the paper or elsewhere on the internet.

[clarity]

- Throughout the introduction, it remains unclear what is meant by "finite-rank case" and "infinite-rank case." This issue may be resolved if the definition of the rank of Sobolev spaces is provided or if it is made clear that it refers to the rank of the target operator.

[significance]

- The theoretical results presented in the paper do not appear to have any immediate tangible implications for the models or learning algorithms of neural operators.
- The motivation of the paper needs to be elaborated as it is currently unclear why it is important to understand the conditions under which the neural operators have injectivity or bijectivity.

**Questions:**

Please kindly correct me if any of my understandings in the above comments are incorrect.

- Line 32: Should "infinite-dimension setting" be "infinite-rank setting"?
- Line 33: What does "infinite-rank Sobolev space" mean? I have not been able to find a definition of the rank of Sobolev spaces either in this paper or elsewhere.

[minor suggestions]

- Line 27 “a operators” → “an operator”
- Line 21 “Bayesian UQ” → “Bayesian uncertainty quantification” (if UQ stands for that)
- Line 23 “existence and uniqueness” → “existence and uniqueness of the solutions”
- Line 47 “and their implementation” → “and that their implementation”
- Line 48 “universality approximation” → “universal approximation”

**Limitations:**

None in particular.

---

> ### Author Rebuttal · Authors · 2023-08-09
>
> Thank you for your careful review, suggestions, and endorsement. We have address your feedback below
>
> _The paper contains a noticeable number of grammatical and typographical errors. It is recommended that an up-to-date automatic checker is used to revise the paper and correct these issues._
>
> We apologize for any and all grammatical and typographical errors. We reviewed all of the text and typography and fixed and removed all errors.
>
> _The model classes, including layers, can be assigned specific symbols to make them stand out in the paper._
>
> We appreciate any suggestions that improve clarity, and want to make sure that we understand the suggestion exactly, and hoped to get more clarification on this point. Is the suggestion to use a specific symbol to represent an injective versus non-injective layer? For example, $\mathcal L^{\textrm{inj}}$ for the injective case and $\mathcal L^{\textrm{bijec}}$ for the bijective case?
>
> _Line 33 mentions "infinite-rank Sobolev spaces," but I could not find the definition of the notion of rank of a Sobolev space in the paper or elsewhere on the internet._
>
>  The prefix infinite-rank was meant purely for emphasis, but we agree that it is confusing and the emphasis isn't needed. We have removed it.
>
> _Throughout the introduction, it remains unclear what is meant by "finite-rank case" and "infinite-rank case." This issue may be resolved if the definition of the rank of Sobolev spaces is provided or if it is made clear that it refers to the rank of the target operator._
>
> In order to make this distinction clearer, we have added an additional sentence at the start of the introduction that frames and defined the finite-rank and infinite-rank cases, and draws attention to their essential differences. We think this has made the distinction clearer, and makes the work more accessible.
>
> _The theoretical results presented in the paper do not appear to have any immediate tangible implications for the models or learning algorithms of neural operators._
>
> It is true that our work is principally theoretical and does not make concrete recommendations for e.g. learning algorithms, but we think it has useful application and takeaways. Mainly by giving an algebraic perspective on neural operators, and by laying the ground work for rigorous application of Bayesian inversion via neural operators. We elaborate on these points more in the global comment.
>
> Additionally, the proof of Theorem 1, that establishes universality of injective operators, proceed by `giving a recipe' for construction an arbitrarily good injective approximator. This recipe can be followed in application, and so may be a guide for injective approximation. We have included statements to this effect just after the statement of Theorem 1.
>
> _The motivation of the paper needs to be elaborated as it is currently unclear why it is important to understand the conditions under which the neural operators have injectivity or bijectivity._
>
> We realized that our motivation could be made even stronger. We have done just this in the global comment above.
>
> _minor suggestions_
>
> Thank you for the suggestions. All of the suggested changes have been made.

---

> > ### Comment · Reviewer_GKdz · 2023-08-11
> >
> > Thank you for your response.
> >
> > > We appreciate any suggestions that improve clarity, and want to make sure that we understand the suggestion exactly, and hoped to get more clarification on this point. Is the suggestion to use a specific symbol to represent an injective versus non-injective layer? For example, for the injective case and for the bijective case?
> >
> > Please consider this as just a very small suggestion. What I meant in the review was assigning dedicated symbols to some sets of functions/operators, e.g., denoting (say) $\mathrm{NO}^{injective}$ to refer to the subset of $\mathrm{NO}_L(\sigma; D, d{in}, d{out})$ consisting only of injective elements. If a symbol is given to such a set, it may be visually faster to see that, e.g., Theorem 1 describes the approximation ability of such a subset.

---

> > > ### Author Response · Authors · 2023-08-11
> > > **Thank you for the clarification**
> > >
> > > Thank you for the clarification.
> > >
> > > Now that we understand, we agree that making such a modification would increase readability, draw attention to the pertinent content of the theorem (injectivity, rather than dimension bookkeeping), and make the theorem \& subsequent discussion more digestible overall.
> > >
> > > We will certainly implement this notation.

---

### Official Review · Reviewer_kfSm · 2023-07-19

**Soundness:** 4 excellent
**Presentation:** 3 good
**Contribution:** 4 excellent
**Rating:** 7
**Confidence:** 3

**Summary:**

This study investigates the injectivity and bijectivity of neural operators (NOs) in the infinite-rank setting which is less investigated than their finite-rank counter, such as invertible flow networks. This study is based on the finite-rank analysis by Puthawala et al. (2022a). As a previous work, Alberti et al. (2022) have shown a global injectivity of an infinite-rank NN based on wavelet expansion. The NO in consideration is formulated as a composite of hidden layers of the form $(\mathcal{L}v)(x) := \sigma( T(v)(x) + b(x) )$. In Section 2, the iff conditions for linear NOs are stated. Particularly Prop 2 (DSS condition for ReLU) is shown by extending the finite-rank results by Puthawala et al. (2022a), and Prop 3 (for bijective activation) is shown based on Fredholm theory. In Section 3, $cc$-universalities of (Theorem 1) injective linear NOs, and (Theorem 2) injective finite-rank NOs are shown. It is remarkable that different from finite-rank results, Theorem 1 does not require any assumption between input and output dimensions. In Section 4, a nonlinear NO with each layer being of the form $\sigma \circ F_1(u) = \sigma(Wu + K(u))$, which covers attention mechanisms, is considered, (Props 3 and 4) sufficient conditions of the surjectivity and bijectivity are stated using the Leray-Schauder fix point theorem, and (Theorem 3) the inverses of bijective nonlinear NOs are constructed.

**Strengths:**

- Modern deep learning tasks tend to be formulated in the *infinite-dimensional* setup, and the authors established a mathematical foundation for a *general class of NOs*.


**Weaknesses:**

(Minor comments)
- I believe that the NOs in consideration cover a wide range of practical examples, but it would broaden their potential readers if the authors could explicitly showcase such examples (eg., not just ones mentioned in l.252 and Example 1, but also subnetworks, operator transformers, and integral autoencoders, etc...)
- Definition 1 would be clearer if the description in ll.77-80 by sentence “and $T_\ell$ : … are sums of …” is associated with an expression such as $T_\ell = …$.
- Some citations in ll.339-341 are duplicated.
- Additional literature (not mandatory as these are in finite-dimensional settings
  - https://proceedings.neurips.cc/paper/2020/hash/2290a7385ed77cc5592dc2153229f082-Abstract.html
  - https://arxiv.org/abs/2204.07415


**Questions:**

please refer to comments in the Weaknesses section.

---

> ### Author Rebuttal · Authors · 2023-08-09
>
> We appreciate your valuable comments, constructive feedback, and endorsement. We have addressed your feedback below
>
>
> _I believe that the NOs in consideration cover a wide range of practical examples, but it would broaden their potential readers if the authors could explicitly showcase such examples (eg., not just ones mentioned in l.252 and Example 1, but also subnetworks, operator transformers, and integral autoencoders, etc...)._
>
> This is a great suggestion. Just after l.39, we have included more examples of model applications including graph neural operators, IAE-nets, Operator transformers, and Factorized Fourier Neural operators. We have cited applications where neural operators are used as well, especially in the context of inverse problems where injectivity is a critical property.
>
> _Definition 1 would be clearer if the description in ll.77-80 by sentence “and_  $T_\ell$ : … are sums of …” _is associated with an expression such as_ $T_\ell =$.
>
> We agree that this change will make the description clearer. We have included such a change in the new version.
>
> _Some citations in ll.339-341 are duplicated._
>
> The duplicated reference have been removed.
>
> _Additional literature (not mandatory as these are in finite-dimensional settings)._
>
> We intended to cite these papers to help fill out the story in finite dimensions but, in an oversight, did not. We have included reference to them in the related work section, and referenced them again the section on universality.

---

### Author Rebuttal · Authors · 2023-08-09

We appreciate all the reviewers' valuable comments, and close attention.
We have a few things that we would like to say globally, in response to points brought up by multiple reviewers. We have also replies to individual reviewers below.

Several reviewers remarked that the question of injectivity for neural operators could be better motivated. We agree. To make the motivation of the work stronger, we would like to add the two remarks to the paper. The two remarks will in the introduction just after Line 39, and will then be briefly restated in the conclusion.


(1) The first remark forms a connection between the algebra of injective/invertable neural operators, and pseudodifferential operators. Pseudodifferential operators revolutionized the theory of linear PDE and so we believe that making such a connection is valuable, and lays the groundwork for an algebraic study of neural operators.

``Our work draws parallels between neural operators and pseudodifferential operators [Taylor, Princeton Mathematical Series 1981], a class that contains many inverses of linear partial differential operators and integral operators. The connection to pseudodifferential operators provided an algebraic perspective to linear PDE [Kohn \& Nirenberg, Communications on Pure and Applied Mathematics 1965, Shubin, Springer 1987]. An important fact in the analysis  of pseudodifferential operators, is that the inverses of certain operators, e.g. elliptic pseudodifferential operator, are themselves pseudodifferential operator. By proving an analogous result in section 4.2, that the inverse of invertible NO are themselves given by NO, we draw an important and profound connection between (non)linear partial differential equations and NO.
    ''

(2) The second described the application of injective neural operators to Bayesian inverse problems. In short, there are a lot of ways to incorrectly discretize inverse problems that makes Bayesian methods discretization dependent. Injective and invertible neural operators are natural ways to do this correctly.

"There are significant benefits to applying Bayesian solution methods to inverse and imaging problems in infinite dimensions. This, for example, allows one to study functions of continuous space \& time variables. These infinite dimensional models can then be approximated by a finite dimensional model without losing discretization invariance, see [Stuart, Acta Numerica 2010].  Crucially, discretization must be done `at the last possible moment,' or else performance degrades as a discretization becomes finer, see [Lassas-Siltanen, Inverse Problems 2004] and also [Lassas-Saksman-Siltanen, Inverse Problems in Imaging 2009]. By formulating machine learning problems in infinite dimensional function spaces and then approximating these methods using finite dimensional subspaces, we avoid bespoke ad-hoc methods and instead obtain methods that apply to any discretization.

One example is the following. Let $\mathcal M$ be the submanifold of $X=L^2([0,1]^2)$ or $X=L^2([0,1]^3)$ corresponding to natural images or 3D medical model. Let $K\subset \mathbb{R}^D$ be a manifold with the same topology as $\mathcal M$, $\iota:\mathbb{R}^D\to X$ be an embedding, and define Let $K_1=\iota(K)\subset \mathcal M$. Given $\mu$, a measure supported on $M$, the task is to find a neural operator $f_\theta\colon X\to X$ that maps (pushes forward) the uniform distribution on the model space $K_1$ to $\mu$ and so thus maps $K_1$ to $\mathcal M$. If $f_\theta\colon X\to X$ is bijective, computing likelihood functions in statistical analysis is made easier via the change of variables formula. Further, we may interpret $f_\theta^{-1}$ as encoder and $f_\theta$ the corresponding decoder, which parameterized elements of $\mathcal M$. As everything is formulated in infinite dimension function space $X$, we obtain discretization invariant methods.''

Motivated by the remarks of Reviewer ZQ9v, we would also like to include the following examples as other functions for which the machinery developed in Section 4.1 applies. We think that these addition flesh out the applications of that section.

The below four points are all other functions, for which we may prove

(A) We can obtain surjectivity (by illustrating coercivity) of neural operators of the form $F(U) = Wu + \sigma_2(K (u))$  where $\sigma_1$ is bijective, $\sigma_2$ bounded.

(B) We can obtain surjectivity (by illustrating coercivity) of neural operators of the form $F(U) = \alpha u + K (u)$ where $K(u) = \int_D a(x,y,u(x),u(y))dy$ where $a(x,y,s_1,s_2)$ is continuous and is such that $\exists R > 0$, $c_1 < \alpha$ so that for all $|(s_1,s_2)| > R$, $\mathrm{sign}(s_1)\mathrm{sign}(s_2)a(x,y,s_1,s_2) \geq -c_1$.

(C) In imaging applications layer wise coercivity of neural networks have been a useful property, see for example [Li et al., Inverse Problems, 2022].

(D) We may also show that coercivity is preserved by perturbations in a bounded domain. This makes it possible to study non-linear and non-positive perturbations of physical models. For example, in quantum mechanics when a non-negative energy potential $|\phi|^4$ is replaced by a Mexican hat potential $-C|\phi|^2 + |\phi|^4$, as occurs in the study of magnetization, superconductors and the Higgs field.

---

### Decision · Program_Chairs · 2023-09-21

**Decision:**

Accept (poster)

**Comment:**

The paper analyzes the injectivity and bijectivity of neural operators. Injectivity is important when input has to be recovered from outputs such as encoding and decoding in autoencoders.  The main contribution is characterizing sufficient conditions under which the neural operator is bijective. This is an extension of Puthawala et al to infinite-dimensional settings. Reviewers agree that this paper is theoretically sound. While there were concerns about the lack of motivation and examples, the rebuttal provides more details and examples. Concerns regarding aliasing and weight matrices can be included in discussions as limitations. For the camera-ready version, the writing has to be improved based on reviews. In particular, the following changes are required:
- A solid introduction that better motivates readers to follow theoretical findings. A short intro to neural operators will make the paper more accessible. Furthermore, the authors need to more elaborate on connections to Puthawala's paper. There are interesting arguments in rebuttal that can be added to the introduction.
- Motivation behind injectivity requires more elaboration. Authors need to give specific applications that rely on injectivity and bijectivity with more reasoning.
- Sections 4.1 and 4.2 need further discussions. In particular, it is important to address the concerns of reviewer ZQ9v
- Include limitations such as aliasing and squared weight matrices in discussions